# Quantifying Error Propagation and Model Collapse in Diffusion Models

**Naïl B. Khelifa** [1]  **Richard E. Turner** [1]  **Ramji Venkataramanan** [1]

## Abstract

Machine learning models are increasingly trained or fine-tuned on synthetic data. Recursively training on such data has been observed to significantly degrade performance in a wide range of tasks, often characterized by a progressive drift away from the target distribution. In this work, we theoretically analyze this phenomenon in the setting of score-based diffusion models. For a realistic pipeline where each training round uses a combination of synthetic data and fresh samples from the target distribution, we obtain upper and lower bounds on the accumulated divergence between the generated and target distributions. Notably, to the best of our knowledge, this is the first lower bound on the divergence between the learned and target distributions, even for standard diffusion models. Our results allow us to characterize different regimes of drift, depending on the score estimation error and the proportion of fresh data used in each generation. In a certain regime, the accumulated divergence after several retraining rounds can be expressed as a discounted sum of score estimation errors made at each generation. We also provide empirical results on synthetic data and images to illustrate the theory.

## 1. Introduction

As generative AI becomes ubiquitous, synthetic data is increasingly being used for training (Zelikman et al., 2022; Gulcehre et al., 2023). However, it has been observed that the performance of a model can degrade dramatically when trained predominantly on its own output (Briesch et al., 2024; Alemohammad et al., 2024a), a phenomenon often termed *model collapse* (Shumailov et al., 2024; Gerstgrasser et al., 2024). For example, with recursive self-training a

generative model may lose the tails of the target distribution or exhibit a marked decrease in diversity (Dohmatob et al., 2024b; Shi et al., 2025). In general, model collapse manifests as a progressive drift away from the target distribution with each training round. This work characterizes the regimes of collapse and quantifies error accumulation in the setting of score-based diffusion models (Sohl-Dickstein et al., 2015; Ho et al., 2020; Song et al., 2021), which achieve state-of-the-art performance for text-to-image, text-to-audio, and video generation (Saharia et al., 2022; Liu et al., 2023; Ho et al., 2022) as well as in a range of scientific applications (Watson et al., 2023; Corso et al., 2023).

There is a growing body of theoretical work studying the effects of recursively training machine learning models with a combination of real and synthetic data. However, this has largely been in the context of either regression models (Dohmatob et al., 2024a; 2025; Dey et al., 2025; Vu et al., 2026; Garg et al., 2026) or maximum-likelihood type estimators for learning parametric distributions (Bertrand et al., 2024; Suresh et al., 2025; Barzilai & Shamir, 2025; Kanabar & Gastpar, 2025). Various authors have studied model collapse in diffusion and flow models empirically and proposed techniques to mitigate it (Alemohammad et al., 2024b; Shi et al., 2025; Yoon et al., 2024; Zhu et al., 2025). On the theoretical side, (Fu et al., 2024; Cui et al., 2026) propose architecture-specific upper bounds on the error accumulation; both papers consider two-layer score networks. Our approach is complementary: working at a population level and in an architecture-agnostic setting, we characterize error accumulation in recursive training by establishing a lower bound on one-generation distribution discrepancy and propagate this bound across generations.

**Setting** We study a recursive training procedure in which the training data in each round is a combination of fresh samples from the true data distribution and synthetic samples generated by the current diffusion model. Let $p_{\text{data}}$ denote the (unknown) true data distribution on $\mathbb{R}^d$. We refer to each training round as a generation and denote the model distribution at the end of generation $i$ by $\hat{p}^i$ for $i \geq 0$, with $\hat{p}^0 = p_{\text{data}}$. At each generation $i \geq 1$, the training data consists of:

- a proportion $\alpha$ of fresh samples drawn from $p_{\text{data}}$, and
- a proportion $(1 - \alpha)$ of synthetic samples drawn from the

[1]University of Cambridge, Cambridge, United Kingdom. Correspondence to: Naïl B. Khelifa <nbk24@cam.ac.uk>, Richard E. Turner <ret26@cam.ac.uk>, Ramji Venkataramanan <rv285@cam.ac.uk>.

*Proceedings of the 43rd International Conference on Machine Learning*, Seoul, South Korea. PMLR 306, 2026. Copyright 2026 by the author(s).

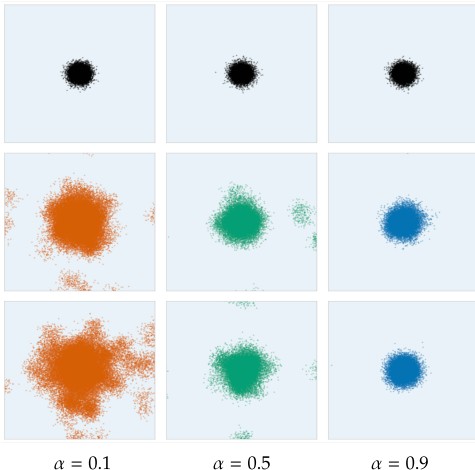

$\alpha = 0.1$      $\alpha = 0.5$      $\alpha = 0.9$

*Figure 1.* Samples generated by a recursively trained model, projected onto the first two principal components, with $p_{\text{data}}$ chosen to be a 10-dimensional Gaussian mixture. Columns correspond to fresh data fractions $\alpha \in \{0.1, 0.5, 0.9\}$ (left to right); rows show generations 0 (start), 10 (middle), and 20 (end). At $\alpha = 0.1$ (left column), the distribution progressively disperses as errors accumulate without sufficient fresh data. At $\alpha = 0.5$ (center), moderate degradation occurs with visible spreading but preserved structure. At $\alpha = 0.9$ (right column), the distribution remains stable throughout, demonstrating that high fresh data fractions effectively prevent collapse. Experiment details in Appendix G.1.

current model $\hat{p}^i$.

We emphasize that the samples are not labeled as fresh or synthetic. This captures a realistic setting where the training is agnostic to how the samples were generated. Therefore, at the population level, the effective training distribution for generation $i \geq 1$ is the mixture

$$q_i := \alpha \, p_{\text{data}} + (1 - \alpha) \, \hat{p}^i. \tag{1}$$

This framework matches that of existing works and may be generalized to a setting where each past generation $k \leq i$ is sampled from in proportion $\alpha_k$ (Fu et al., 2024). A score network is trained on samples from $q_i$, which is then used to produce the next synthetic distribution $\hat{p}^{i+1}$. This defines the recursion

$$\hat{p}^i \xrightarrow{\text{ mix with } p_{\text{data}} } q_i \xrightarrow{\text{ train model }} \hat{p}^{i+1}. \tag{2}$$

The goal of this work is to answer the following question:

> *In this recursive setting, how do errors accumulate over generations and how does the divergence between $\hat{p}^i$ and $p_{\text{data}}$ evolve?*

Figure 1 illustrates the effect of different mixing proportions on a recursively trained diffusion model, for $p_{\text{data}}$ a 10-dimensional mixture of Gaussians. Figure 8 in Appendix G.2 shows similar behavior for the Fashion-MNIST

dataset (Xiao et al., 2017), i.e., $p_{\text{data}}$ is the distribution that uniformly samples images from the original dataset.

**Learning Target** At a high level, training a diffusion model is equivalent to learning the score functions of Gaussian-smoothed versions of $p_{\text{data}}$ at a range of noise levels (Song et al., 2021; Chen et al., 2023c). Although the intention is to learn $p_{\text{data}}$, the score network at generation $i$ is trained on samples drawn from the mixture $q_i$. At the population level, the identifiable target of training is therefore the score of $q_i$, not the score of $p_{\text{data}}$. This mismatch between the intended target and the learned target is a fundamental source of error under recursive training. Furthermore, the score network does not learn the score of $q_i$ perfectly (due to finite-sample and function approximation errors). This score-matching error propagates through the stochastic differential equation (8) defining the reverse process, causing the learned distribution $\hat{p}^{i+1}$ to differ from $q_i$. Characterizing this error propagation is crucial for understanding the divergence of $\hat{p}^{i+1}$ from $p_{\text{data}}$ as $i$ grows.

We track the evolution of the recursion using two quantities based on the $\chi^2$-divergence (defined in (5)):

- *Accumulated divergence*, measuring the divergence between the $i$-th generation model and the true data distribution:

$$D_i := \chi^2(\hat{p}^i \,\|\, p_{\text{data}}). \tag{3}$$

- The *intra-generation* divergence, measuring the error introduced in one training round:

$$I_i := \chi^2(\hat{p}^{i+1} \,\|\, q_i). \tag{4}$$

We work primarily with the $\chi^2$-divergence because, in the perturbative regime relevant for our analysis, it is equivalent to the KL divergence up to universal constants (see empirical evidence in Figure 4, where our bounds hold for both divergences), while yielding a particularly convenient algebra under the refresh step $q_i = \alpha p_{\text{data}} + (1 - \alpha)\hat{p}^i$ that makes the cross-generation recursion explicit.

**Main contributions** The recursion induced by training on real-synthetic mixtures involves two competing effects: error *mitigation* from the fresh data used in each step, and error *accumulation* from imperfect score learning and sampling. Our contributions quantify both terms and their interaction across a growing number of generations.

1. *Lower bound for intra-generation divergence*: We provide a lower bound (Proposition 3.3) for $I_i$ that, for small score errors, is the product of two diffusion-specific terms: (i) *observability* of errors at the endpoint of the diffusion path (captured by a coefficient $\eta_i$), and (ii) the energy of the

path-wise score error.

$$\underbrace{\chi^2(\hat{p}^{i+1}\|q_i)}_{\substack{\text{Intra-Generational}\\\text{Divergence}}} \gtrsim \underbrace{\eta_i}_{\substack{\text{Observability}\\\text{of Errors}}} \cdot \underbrace{\varepsilon^2_{\star,i}}_{\substack{\text{Score Error}\\\text{Energy}}}$$

To our knowledge, this is the first lower bound for diffusion models (even without recursive training) quantifying the discrepancy between the learned distribution and the target.

2. *Intra-generation Divergence Control via Score Error:* Combining the above lower bound with a standard upper bound (Proposition 3.1), and proving an equivalence between the score error energies on the ideal-path and the learned-path (for small errors), we show in Theorem 3.5 that in this regime $I_i$ is equivalent to the score error energy (up to constants): $\chi^2(\hat{p}^{i+1}\|q_i) \asymp \varepsilon^2_{\star,i}$.

3. *Long-horizon regimes and a discounted accumulation law.* We analyze the accumulated divergence $D_N$ for small per-generation score errors, and identify a dichotomy as the generation $N$ grows (Proposition 4.1 and Theorem 4.2): (i) if the score errors persist across generations (energies bounded below by a positive constant), then the accumulated divergence $D_N$ cannot vanish; (ii) if the score error energies are summable, i.e. $\sum_k \varepsilon^2_{\star,k} < \infty$, then $D_N$ remains stable (uniformly bounded) with growing $N$. Moreover, the accumulated divergence after a finite number of generations $N$ admits an interpretable geometrically-discounted decomposition up to a bias term:

$$\chi^2(\hat{p}^N\|p_{\text{data}}) + C_{\text{bias}} \asymp \sum_{k=0}^{N-1} \underbrace{(1-\alpha)^{2(N-1-k)}}_{\substack{\text{Forgetting}\\\text{(real data)}}} \cdot \underbrace{\varepsilon^2_{\star,k}}_{\substack{\text{Accumulation}\\\text{(synthetic data)}}},$$

which shows that errors from $m$ generations in the past are suppressed by a factor $(1-\alpha)^{2m}$, quantifying the effect of $\alpha$, the proportion of fresh data used in each training round. Qualitatively, this is consistent with empirical and theoretical work in a variety of settings showing that incorporating fresh data in each training round can mitigate model collapse, e.g. (Gerstgrasser et al., 2024; Zhu et al., 2025; Bertrand et al., 2024; Dey et al., 2025).

We also provide experiments on both synthetic and real-image datasets to illustrate the validity of our bounds. Our results are relevant in the regime where the energy of the path-wise score error (defined in (10)) is small for all but a few generations. This regime is of interest since we seek to quantify how the intra-generation and accumulated divergences evolve when the score is estimated with high accuracy in each generation. However, we note that the number of samples for the score error to be accurately estimated may grow exponentially in the dimension $d$ (see (12)).

**Notation** Bold symbols (e.g. $\mathbf{X}_t$, $\mathbf{Y}_t$) denote $\mathbb{R}^d$-valued random variables or processes, and calligraphic symbols denote measurable sets. The space of continuous functions from $[0, T]$ to $\mathbb{R}^d$ is denoted by $C([0, T], \mathbb{R}^d)$. For a stochastic process $(\mathbf{Z}_t)_{t\in[0,T]}$ defined on $C([0, T], \mathbb{R}^d)$, we write $\text{Law}((\mathbf{Z}_t)_t)$ for its law on path space and $\text{Law}(\mathbf{Z}_t)$ for the law of its marginal at time $t$. For two smooth probability densities $p, q$ on $\mathbb{R}^d$ and a convex function $f$ with $f(1) = 0$, the associated $f$-divergence is defined as

$$D_f(p\|q) = \int_{\mathbb{R}^d} f\left(\frac{p(\mathbf{x})}{q(\mathbf{x})}\right) q(\mathbf{x}) \, \mathrm{d}\mathbf{x}. \tag{5}$$

In particular, $f(u) = u \log u$ yields the Kullback–Leibler divergence and $f(u) = (u - 1)^2$ yields the $\chi^2$-divergence. For two nonnegative functions (or sequences) $f$ and $g$, we write $f(\mathbf{x}) \asymp g(\mathbf{x})$ if there exist constants $0 < c_1 \leq c_2 < \infty$ such that $c_1 \, g(\mathbf{x}) \leq f(\mathbf{x}) \leq c_2 \, g(\mathbf{x})$.

## 2. Background and Model Assumptions

**Forward diffusion** To define the score-matching objective and the subsequent sampling procedure, we associate to each $q_i$ a forward diffusion process on a truncated time interval $[t_0, T]$, with $0 < t_0 < T$. We consider the following variance-preserving type Ornstein-Uhlenbeck process (Karatzas & Shreve, 2014; Song et al., 2021):

$$\mathrm{d}\mathbf{X}^i_t = -\tfrac{1}{2}\mathbf{X}^i_t \, \mathrm{d}t + \mathrm{d}\mathbf{B}_t, \tag{6}$$

where $(\mathbf{B}_t)_{t\in[0,T]}$ is a standard $d$-dimensional Brownian motion. The process is initialized with $\mathbf{X}^i_0$ drawn from a suitable distribution $q_{i,0}$. Denote by $q_{i,t} := \text{Law}(\mathbf{X}^i_t)$ the time-$t$ marginal of the forward process, with population score $\mathbf{s}^\star_i(\mathbf{x}, t) := \nabla_{\mathbf{x}} \log q_{i,t}(\mathbf{x})$.

**Reverse-time generation and learning** Under standard regularity conditions (Anderson, 1982; Haussmann & Pardoux, 1986), the time reversal of (6) solves the reverse-time SDE:

$$\mathrm{d}\mathbf{Y}^{i,\star}_s = \left[-\tfrac{1}{2}\mathbf{Y}^{i,\star}_s - \mathbf{s}^\star_i(\mathbf{Y}^{i,\star}_s, s)\right] \mathrm{d}s + \mathrm{d}\bar{\mathbf{B}}_s, \tag{7}$$

integrated backward from $s = T$ to $s = 0$. Here $(\bar{\mathbf{B}}_s)_{s\in[0,T]}$ is another Brownian motion on the same probability space. If we run the reverse SDE starting from $\mathbf{Y}^{i,\star}_T \sim q_{i,T}$, then $\mathbf{Y}^{i,\star}_0 \sim q_{i,0}$. In practice, the score $\mathbf{s}^\star_i$ is unknown and is approximated by a learned network $\mathbf{s}_{\theta_i}$ trained on samples from $q_i$ in (1). Moreover, since the score function is often ill-behaved and challenging to estimate as $t \to 0$ (Song & Ermon, 2020; Kim et al., 2022), as done in practice we truncate the learned process at $s = t_0 > 0$. We choose $q_{i,0}$ such that $q_{i,t_0} = q_i$. Plugging the learned score $\mathbf{s}_{\theta_i}$ into the reverse dynamics yields the learned reverse process:

$$\mathrm{d}\hat{\mathbf{Y}}^i_s = \left[-\tfrac{1}{2}\hat{\mathbf{Y}}^i_s - \mathbf{s}_{\theta_i}(\hat{\mathbf{Y}}^i_s, s)\right]\mathrm{d}s + \mathrm{d}\bar{\mathbf{B}}_s, \tag{8}$$

with initialization $\hat{\mathbf{Y}}_T^i \sim \mathcal{N}(0, \mathbf{I}_d)$. The next generation model is thus denoted $\hat{p}^{i+1} := \mathrm{Law}(\hat{\mathbf{Y}}_{t_0}^i)$. Finally, denote the corresponding path laws on the path-space $C([t_0, T], \mathbb{R}^d)$ by

$$\mathbb{P}_i^\star := \mathrm{Law}\left((\mathbf{Y}_s^{i,\star})_{s \in [t_0, T]}\right), \quad \hat{\mathbb{P}}_i := \mathrm{Law}\left((\hat{\mathbf{Y}}_s^i)_{s \in [t_0, T]}\right),$$

so that the time-$t_0$ marginals of the ideal and the learned reverse processes satisfy

$$\mathbf{Y}_{t_0} \sim q_i \text{ under } \mathbb{P}_i^\star, \text{ and } \hat{\mathbf{Y}}_{t_0}^i \sim \hat{p}^{i+1} \text{ under } \hat{\mathbb{P}}_i. \quad (9)$$

With some abuse of notation, we use $q_i, \hat{p}^{i+1}$ for both the laws and the densities with respect to the Lebesgue measure.

**Score error and energies** We define the score error for each $t \in [t_0, T]$ as the vector field,

$$\mathbf{e}_{i,t} := \mathbf{s}_{\theta_i}(\cdot, t) - \mathbf{s}_i^\star(\cdot, t),$$

and its path-wise energies

$$\varepsilon_{\star,i}^2 := \mathbb{E}_{(\mathbf{Y}_s)_{s \in [t_0, T]} \sim \mathbb{P}_i^\star}\left[\int_{t_0}^T \|\mathbf{e}_{i,s}(\mathbf{Y}_s)\|_2^2 \, \mathrm{d}s\right], \quad (10)$$

$$\hat{\varepsilon}_i^2 := \mathbb{E}_{(\mathbf{Y}_s)_{s \in [t_0, T]} \sim \hat{\mathbb{P}}_i}\left[\int_{t_0}^T \|\mathbf{e}_{i,s}(\mathbf{Y}_s)\|_2^2 \, \mathrm{d}s\right]. \quad (11)$$

These quantities control the intra-generation divergence $I_i$ defined in (4). We note that (10) is the score-matching loss for generation $i$ (Hyvärinen, 2005; Vincent, 2011; Song & Ermon, 2019; Ho et al., 2020). At a finite sample level, for a training sample $\mathcal{D}_i \overset{\text{i.i.d}}{\sim} q_i$ of size $n_i$, under appropriate assumptions, the minimax-optimal score estimation error satisfies (Zhang et al., 2024):

$$\varepsilon_{\star,i}^2 \lesssim \mathrm{polylog}(n_i) \, n_i^{-1} \, (1/t_0)^{d/2}. \quad (12)$$

Similar bounds, relying on different assumptions, have been derived in various works (Oko et al., 2023; Dou et al., 2024; Wibisono et al., 2024; Lewis et al., 2025). Our results are relevant in the regime where $\varepsilon_{\star,i}^2 < 1$. From (12), this requires a training sample of size $n_i \sim (1/t_0)^{d/2}$. However, when the distribution $p_{\text{data}}$ is supported on a low dimensional manifold (as is the case for images), under suitable assumptions we expect the optimal score estimation error to depend only on the intrinsic dimension $d^\star \ll d$ (Chen et al., 2023b).

Our main goal is to quantify how the score estimation error (10) propagates within and across generations. We make a few simplifying assumptions to keep the analysis tractable and reduce bookkeeping.

**Model Assumptions** The proportion $\alpha$ of fresh samples used for training is strictly positive. We will assume $p_{\text{data}}$ admits a density with respect to Lebesgue measure and the model distribution $\hat{p}^i$ is absolutely continuous with respect to $p_{\text{data}}$, for each generation $i$. This is a standard assumption in analyses of diffusion models (Chen et al., 2023c; Benton et al., 2024). In addition to the score estimation error, there are two other sources of error in practice. One is due to the learned reverse process (8) being initialized with $\mathbf{Y}_T^{i,\star} \sim \mathcal{N}(0, \mathbf{I}_d)$ rather than $\mathbf{Y}_T^{i,\star} \sim q_{i,T}$ used for the ideal process in (7). The KL divergence between $q_T$ and $\mathcal{N}(0, \mathbf{I}_d)$ converges to 0 exponentially fast in $T$ (Bakry et al., 2014), so we ignore this error. The second source of error is the discretization error when the reverse process is implemented in discrete-time. We do not quantify these errors as our main goal is to obtain lower bounds on the intra-generation and accumulated errors. Obtaining error propagation bounds taking these additional sources of error into account is a direction for future work.

## 3. Intra-generation divergence bounds

We now analyze the *intra-generational* mechanism by which score estimation error in generation $i$ translates into mismatch between the sampling output $\hat{p}^{i+1}$ and the training distribution $q_i$ in (1). Our goal is to relate the one-step divergence $I_i$ defined in (4) to the pathwise score-error energies defined by (10)-(11).

**Key Technical Ideas and Girsanov Quantities** To quantify how local score errors propagate to the final samples, we use a change-of-measure argument based on Girsanov's theorem (Girsanov, 1960; Le Gall, 2018). This is a key result in the theory of diffusion models that shows how the drift mismatch between the ideal and learned reverse-time processes (score error in our setting) controls the Radon-Nikodym derivative of the path-space measure $\hat{\mathbb{P}}_i$ with respect to $\mathbb{P}_i^\star$. Define the time-integrated random error,

$$M_t^i := -\int_{t_0}^t \mathbf{e}_{i,s} \cdot \mathrm{d}\bar{\mathbf{B}}_s, \quad (13)$$

and its quadratic variation

$$\langle M^i \rangle_t = \int_{t_0}^t \|\mathbf{e}_{i,s}\|_2^2 \, \mathrm{d}s. \quad (14)$$

We also define $Z_t^i := M_t^i - \frac{1}{2}\langle M^i \rangle_t$. We note that both $M_t^i$ and $\langle M^i \rangle_t$ are functions of a random trajectory $(\mathbf{Y}_s)_{s \in [t_0, t]}$, distributed according to a path-space measure, e.g. $\mathbb{P}_i^\star$ or $\hat{\mathbb{P}}_i$. Under mild and standard assumptions (A1 and A2 below), Girsanov's theorem states that $\hat{\mathbb{P}}_i \ll \mathbb{P}_i^\star$ and $\frac{\mathrm{d}\hat{\mathbb{P}}_i}{\mathrm{d}\mathbb{P}_i^\star} = \exp(Z_T^i)$. Crucially, the likelihood ratio of the terminal marginals $R_i := \hat{p}^{i+1}/q_i$ is obtained by projecting this

path density onto the terminal state (Proposition B.1):

$$R_i(\mathbf{x}) := \frac{\hat{p}^{i+1}(\mathbf{x})}{q_i(\mathbf{x})} = \mathbb{E}_{(\mathbf{Y}_s)_{s \in [t_0,T]} \sim \mathbb{P}_i^\star}\left[e^{Z_T^i} \mid \mathbf{Y}_{t_0} = \mathbf{x}\right]. \tag{15}$$

This relation is central to our analysis: it shows how pathwise score errors, encoded through $Z_T^i$, are projected onto the terminal marginal and thereby induce distributional drift. A key challenge in our analysis is to obtain a good lower bound on the conditional expectation in (15). We begin with an upper bound that follows from a standard application of Girsanov's theorem.

### 3.1. Upper Bound via Change of Measure

Assume there exists $i_0 \geq 1$ such that for all generations $i \geq i_0$, the following conditions hold.

**(A1) Finite energy along learned paths**, i.e. $\hat{\varepsilon}_i^2 < \infty$.

**(A2) Martingale property of the Girsanov density.** The exponential $e^{Z_T^i}$ associated with the drift difference between (7) and (8) defines a true martingale on $[t_0, T]$ under $\mathbb{P}_i^\star$ so that the Girsanov change of measure holds. (A more precise version is stated in Appendix B.1.)

Assumptions A1-A2 impose no restriction on the form of the score error beyond finite quadratic energy and are standard in diffusion theory. They yield the following well-known upper bound relating sampling error to the pathwise score-estimation energy (Chen et al., 2023a;c; Benton et al., 2024).

**Proposition 3.1** (Intra-generational upper bound). *Under A1–A2, $\hat{\mathbb{P}}_i \ll \mathbb{P}_i^\star$ and, by data processing inequality (marginalization to time $t_0$),*

$$\mathrm{KL}(\hat{p}^{i+1}\|q_i) \leq \mathrm{KL}(\hat{\mathbb{P}}_i\|\mathbb{P}_i^\star) = \tfrac{1}{2}\hat{\varepsilon}_i^2. \tag{16}$$

For completeness, we give the proof in Appendix B.1.

### 3.2. Lower Bound Via Error Observability

The upper bound of Proposition 3.1 measures the entire path-space score error. Obtaining a divergence lower bound is more challenging as we have to work with the marginal likelihood ratio given by (15).

First Challenge: Observability of Errors

It is possible that the path-space score error $\hat{\varepsilon}_i^2 > 0$, but $\mathrm{KL}(\hat{p}^{i+1}\|q_i) = 0$, i.e., a non-zero path-wise error may leave no trace at the end point $t_0$ if it is orthogonal (in $L^2$) to all observables at $t_0$. This phenomenon is intrinsic, so we need an observability condition linking path-wise perturbations to the endpoint. We quantify the visibility of score errors at $t_0$ through the following coefficient.

**Definition 3.2** (Observability of Errors Coefficient). Recalling the definition of $M_T^i$ in (13),

$$\eta_i := \frac{\mathrm{Var}_{\mathbb{P}_i^\star}\big(\mathbb{E}[M_T^i \mid \mathbf{Y}_{t_0}]\big)}{\varepsilon_{\star,i}^2} \in [0,1], \tag{17}$$

with the convention $\eta_i := 0$ if $\varepsilon_{\star,i}^2 = 0$. The denominator in (17) equals $\mathrm{Var}_{\mathbb{P}_i^\star}(M_T^i)$, by Itô isometry (Le Gall, 2018).

The leading term in our divergence lower bound (Proposition 3.3) is proportional to $\eta_i$. Hence the bound is informative whenever $\eta_i > 0$.

*When is $\eta_i > 0$ in practice?* By definition,

$$\eta_i = 0 \iff \mathbb{E}_{\mathbb{P}_i^\star}\big[M_T^i \mid \mathbf{Y}_{t_0}\big] = 0 \ \mathbb{P}_i^\star\text{-a.s.},$$

i.e., $\eta_i = 0$ when the drift-mismatch martingale $M_T^i$ is *conditionally orthogonal* to all terminal observables. This cancellation is highly non-generic: in common parametric score models (e.g. neural networks with smooth activations (e.g. SiLU, GeLU)), due to random initialization and stochastic optimization noise we expect $\eta_i > 0$. The experiments discussed in Appendix G suggest that in realistic data-driven setups, $\eta_i > 0$ holds across generations; see Figure 7 (10-dimensional Gaussian Mixture) and Figure 9 (CIFAR-10 dataset). More generally, a simple and practically relevant sufficient condition is *state dependence* of the score perturbation function $\mathbf{e}_{i,t}(\mathbf{x})$: time-only (or purely path-orthogonal) perturbations can be averaged out by the conditional expectation, whereas perturbations coupled to the sample state $\mathbf{x}$ typically leave a detectable imprint on the endpoint. In particular, any nontrivial state-dependent affine component yields observability: if $\mathbf{e}_{i,t}(\mathbf{x}) = \mathbf{w}\,\mathbf{x} + \xi(t)$ with $\mathbf{w} \neq 0$ and $\xi$ independent of $\mathbf{x}$, then $\eta_i > 0$. This dichotomy is illustrated empirically in Figure 2, where state-dependent perturbations exhibit higher observability than time-only perturbations.

For clarity, we provide an additional toy example of a basic linear-Gaussian model with state-dependent score error, where the observability coefficient $\eta_i \geq \eta > 0$. Consider the linear-Gaussian reverse diffusion $\mathrm{d}\overline{\mathbf{Y}}_{i,s}^\star = -\beta\mathbf{Y}_{i,s}^\star\mathrm{d}s + \mathrm{d}\bar{\mathbf{B}}_s$ and a linear state-dependent score error of the form $\mathbf{e}_{i,s}(\mathbf{x}) = a_i(s)C_i\mathbf{x}$ where $C_i = C_i^\top \succ 0$, and $a_i(s) \geq 0$ is not identically zero. In this setting, one has that $M_T^i = -\int_{t_0}^T a_i(s)(C_i\mathbf{Y}_{i,s}^\star) \cdot \mathbf{B}_s$. Using Ito's formula on the quadratic form $(\mathbf{Y}_{i,s}^\star)^\top C_i\mathbf{Y}_{i,s}^\star$, one has that, $\mathbb{E}_{\mathbb{P}_i^\star}[M_T^i \mid \mathbf{Y}_s = \mathbf{y}] = \mathbf{y}^\top K_i\mathbf{y} + \kappa_i$ for some matrix $K_i \succ 0$, and a constant $\kappa_i$. Hence, $\mathrm{Var}(\mathbb{E}_{\mathbb{P}_i^\star}[M_T^i \mid \mathbf{Y}_s]) > 0$, which implies that $\eta_i > 0$. Intuitively, the conditions $C_i = C_i^\top \succ 0$ and $a_i(s) \geq 0$ rule out sign-changing cancellations of the state-dependent score error along the reverse dynamics.

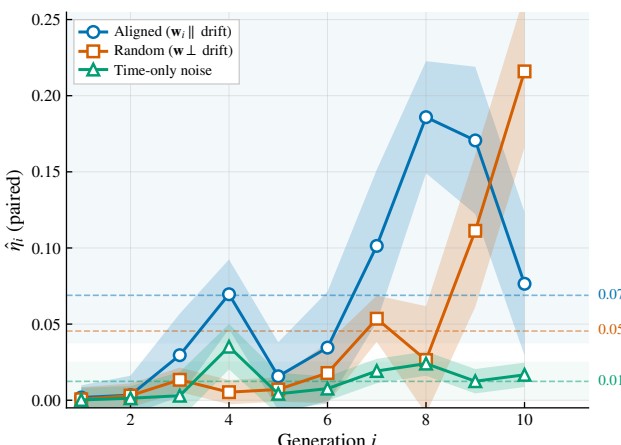

*Figure 2.* **Observability of three classes of score perturbations in CIFAR-10 ([Krizhevsky, 2009](#)) diffusion sampling.** We inject controlled perturbations $\mathbf{e}_i(\mathbf{x}, t)$ and estimate the observability coefficient $\hat{\eta}_i = \text{Var}(\widehat{\mathbb{E}}[M_i \mid \mathbf{Y}_{t_0}])/\text{Var}(M_i)$ using paired reverse trajectories with shared diffusion noise. **Aligned**: $\mathbf{e}_{i,t}(\mathbf{x}) = \mathbf{w}_i\mathbf{x}$ with $\mathbf{w}_i$ chosen so that the error points in a direction correlated with the drift (i.e., errors push trajectories further along where they are already going). **Random**: $\mathbf{e}_{i,t}(\mathbf{x}) = \mathbf{w}\mathbf{x}$ with random $\mathbf{w}$ independent of the drift direction **Time-only**: $\mathbf{e}_{i,t}(\mathbf{x}) = \xi(t)$, independent of state. State-dependent perturbations yield higher $\hat{\eta}_i$, confirming that errors coupled to the sample state are more visible at the terminal distribution.

SECOND CHALLENGE: RATIO OF MARGINALS

The ratio of marginals, given by (15), depends on an exponential functional of the score error along the path, projected to the endpoint. Thus, controlling marginals requires uniform integrability and moment bounds to prevent rare trajectories from dominating $e^{Z_T^i}$ and to relate its fluctuations to the scale $\varepsilon_{\star,i}^2$. This motivates the following regularity assumptions: moment bounds on the Girsanov ratio and the quadratic variation in (14).

**(A3)** $L^{1+\delta}$-**integrability of the Girsanov density.** There exist $\delta > 1$ and $C_\delta < \infty$ such that the Radon–Nikodym derivative $\frac{d\hat{\mathbb{P}}_i}{d\mathbb{P}_i^\star} = e^{Z_T^i}$ satisfies

$$\sup_{i \geq i_0} \mathbb{E}_{\mathbb{P}_i^\star}\left[e^{Z_T^i(1+\delta)}\right] \leq C_\delta. \tag{18}$$

**(A4) Quadratic Variation Moment.** There exists $K_\gamma < \infty$ and $\gamma > \max\{2, \frac{4}{\delta-1}\}$ with $\delta > 1$ defined in A3 such that

$$\sup_{i \geq i_0} \frac{\mathbb{E}_{\mathbb{P}_i^\star}[\langle M^i \rangle_T^{2+\gamma}]}{\varepsilon_{\star,i}^{2(2+\gamma)}} \leq K_\gamma. \tag{19}$$

**Discussion on assumptions**    We discuss settings in which assumptions A3 and A4 are likely to hold. For A3, a sufficient condition is when score errors are uniformly bounded,

i.e. there exist a constant $c > 0$ such that $\|\mathbf{e}_i(\mathbf{x}, t)\|_2 \leq c$ for all $(\mathbf{x}, t) \in \mathbb{R}^d \times [t_0, T]$. In that case, the quadratic variation $\langle M^i \rangle_T$ is deterministically bounded by $c^2(T - t_0)$. Since $Z_T^i = M_T^i - \frac{1}{2}\langle M^i \rangle_T$, standard exponential-martingale estimates imply A3. A4 is a complementary concentration assumption on the quadratic variation $\langle M^i \rangle_T$ relative to its mean $\varepsilon_{\star,i}^2$. It is not implied by boundedness of errors alone and rules out situations in which the pathwise score-error energy is carried by rare trajectories with unusually large spikes.

Assumptions A3-A4 are required to hold for generations after some $i_0 > 0$, which avoids artifacts due to arbitrary or poor initializations. Figure 3 highlights the importance of this transient regime: the misalignment at $i = 1$ does not contradict the theory, but rather indicates that this transient lasts roughly one step in our setup; after that, the bounds are consistent with theory. Moreover, assumptions A3-A4 are stated with free parameters to retain flexibility; for concreteness, one may for instance fix $(\delta, \gamma) = (3, 3)$.

We now state the main lower bound result.

**Proposition 3.3** (Intra-generational Lower Bound). *Under Assumptions A1–A4, the following holds for generations $i \geq i_0$. There exist constants $c > 0$ and $C < \infty$ (depending only on $\eta_i$, $K_\gamma$, $C_\delta$, and $\delta$) such that if the total score error satisfies the perturbative condition $\varepsilon_{\star,i}^2 \leq 1$, then:*

$$\chi^2(\hat{p}^{i+1} \| q_i) \geq \frac{1}{4} \cdot \eta_i \cdot \varepsilon_{\star,i}^2 - C \cdot \varepsilon_{\star,i}^4. \tag{20}$$

*In particular, when $\varepsilon_{\star,i}^2 \leq \min\{1, \frac{\eta_i}{8C}\}$, we obtain the clean lower bound: $\chi^2(\hat{p}^{i+1} \| q_i) \geq \frac{1}{8} \cdot \eta_i \cdot \varepsilon_{\star,i}^2$.*

The proof is given in Appendix D. The upper and lower bounds of Propositions 3.1 and 3.3 are illustrated in Figure 3 for a 10-dimensional Gaussian mixture, and compared with the empirically computed divergences.

*Remark* 3.4. The lower bound of Proposition 3.3 is not restricted to the setting of recursive training, and applies to any standard diffusion model defined via (8). Formally, for any target distribution $p_{\text{data}}$ and any learned distribution $\hat{p}$ obtained by training a diffusion model, one has under the conditions of Proposition 3.3 (dropping the generation $i$),

$$\chi^2(\hat{p} \| p_{\text{data}}) \geq \frac{1}{4}\eta\varepsilon_\star^2 - C\varepsilon_\star^4,$$

where, $\eta$ and $\varepsilon_\star^2$ are defined similarly to $\eta_i$ (Definition 3.2) and $\varepsilon_{\star,i}^2$ (equation (10)) without the notion of generation $i$.

We note that, since the forward process runs over a finite horizon $T$ and we initialize the reverse process from $\mathcal{N}(0, \mathbf{I}_d)$ rather than from $q_{i,T}$, there is an additional bias term. However, for the OU forward diffusion, this initialization mismatch decays exponentially fast in $T$ ([Bakry & Émery, 1985](#)). By contrast, the score-error energy $\varepsilon_{\star,i}^2$

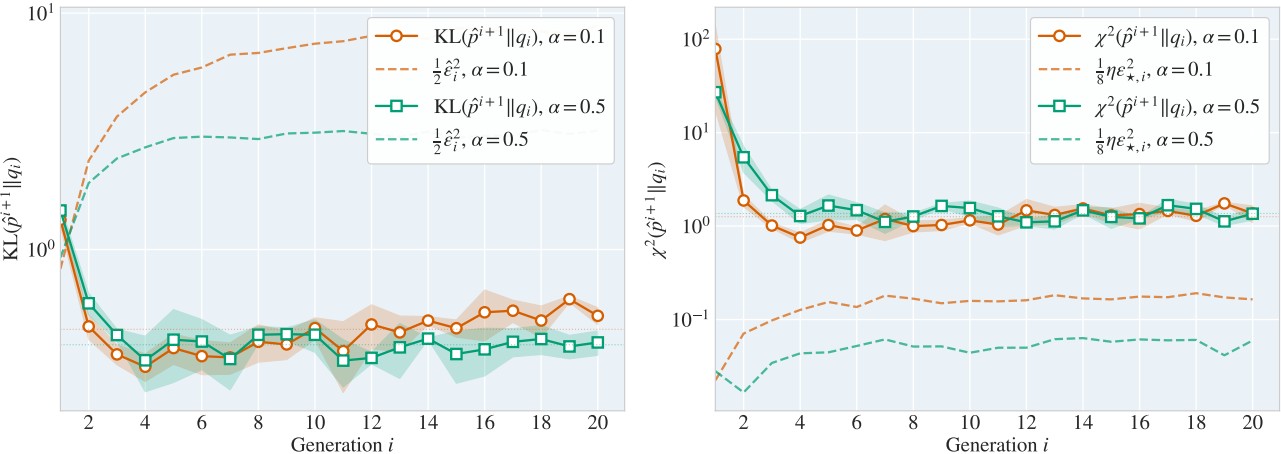

*Figure 3.* **Empirical verification of intra-generational error bounds for a Gaussian mixture.** $p_{\text{data}} \sim \frac{1}{5}\sum_{k=1}^{5}\mathcal{N}(\boldsymbol{\mu}_k, \sigma^2\mathbf{I}_{10}) \in \mathbb{R}^{10}$. **Left:** Upper bound verification (Proposition 3.1). The KL divergence $\text{KL}(\hat{p}^{i+1}\|q_i)$ between the learned distribution and the training mixture remains bounded by $\frac{1}{2}\hat{\varepsilon}_i^2$. The bound is tight at early generations and becomes conservative as the system equilibrates. **Right:** Lower bound verification (Proposition 3.3). The $\chi^2$ divergence $I_i = \chi^2(\hat{p}^{i+1}\|q_i)$ is bounded below by $\frac{1}{8}\hat{\eta}_i\varepsilon_{\star,i}^2$, where $\hat{\eta}_i$ is the estimated observability coefficient. Shaded regions indicate $\pm 1$ standard deviation across runs. Both bounds hold consistently across $\alpha \in \{0.1, 0.5\}$, validating the theoretical framework. Experiment details are given in Appendix G.1.

entering our lower bound is not expected to grow exponentially with $T$; under a mild uniform-in-time second-moment bound on the score error, it grows at most linearly in $T$. Hence, for sufficiently large $T$, this initialization mismatch is negligible compared with the score-error term that drives the lower-bound mechanism.

### 3.3. Two-sided equivalence

Proposition 3.1 controls the intra-generation divergence from above in terms of the learned-path energy $\hat{\varepsilon}_i^2$, whereas Proposition 3.3 provides a matching lower bound in terms of the ideal-path energy $\varepsilon_{\star,i}^2$, up to the observability factor $\eta_i$ and higher-order terms. To turn these complementary statements into a single equivalence $I_i = \chi^2(\hat{p}^{i+1}\|q_i) \asymp \varepsilon_{\star,i}^2$, the two energies $\varepsilon_{\star,i}^2$ (under $\mathbb{P}_i^\star$) and $\hat{\varepsilon}_i^2$ (under $\hat{\mathbb{P}}_i$) as well as the two divergences KL (upper bound) and $\chi^2$ (lower bound) need to be related. Under Assumptions A3 and A4, we show that these energies and divergences are equivalent. Combining these equivalence results with Propositions 3.1 and 3.3 yields the following two-sided control of the intra-generational divergence.

**Theorem 3.5** (Two-Sided bounds for intra-generational divergence)**.** *Under Assumptions A1- A4, the following holds for generations $i \geq i_0$. There exist constants $c > 0$ and $C < \infty$ (depending only on $\delta, C_\delta, \gamma, K_\gamma$) such that, when the perturbative condition $\varepsilon_{\star,i}^2 \leq \min\{1, \frac{\eta_i}{8C}\}$ holds, we have:*

$$\frac{1}{4}\,\eta_i\,\varepsilon_{\star,i}^2 \;-\; C\cdot\varepsilon_{\star,i}^4 \;\leq\; \chi^2(\hat{p}^{i+1}\|q_i) \;\leq\; 4\,\varepsilon_{\star,i}^2 \;+\; c\varepsilon_{\star,i}^4.$$

*In particular, for sufficiently small $\varepsilon_{\star,i}^2$ and $\eta_i \geq \underline{\eta} > 0$, we have*

$$\chi^2(\hat{p}^{i+1}\|q_i) \asymp \varepsilon_{\star,i}^2. \tag{21}$$

The proof is given in Appendix E. This theorem shows that the intra-generational divergence is controlled by the pathwise score-error energy and that the two are equivalent (up to constants), up to observability and tail effects. Figure 4 indicates these bounds hold in a fully data-driven experiment for both the $\chi^2$ and KL divergences.

## 4. Error Accumulation Across Generations

Theorem 3.5 shows that when the score error is observable (i.e., $\eta_i > 0$), the intra-generational divergence $I_i = \chi^2(\hat{p}^{i+1}\|q_i)$ is equivalent (up to constants) to $\varepsilon_{\star,i}^2$ in the perturbative regime. We now study how this divergence propagates through the recursion (2) over many generations, recalling our assumption that the refresh rate $\alpha > 0$.

A key observation is that the refresh step contracts the accumulated divergence. Indeed, a simple calculation (Lemma F.1) shows that $\chi^2(q_i \| p_{\text{data}}) = (1-\alpha)^2\chi^2(\hat{p}^i \| p_{\text{data}})$. In contrast, the score error in the next round of training increases the accumulated divergence. These two competing effects determine the evolution of $D_i = \chi^2(\hat{p}_i \| p_{\text{data}})$ as $i$ grows. We distinguish two cases, assuming small score errors in each generation: if $\sum_{i\geq 0}\varepsilon_{\star,i}^2 = \infty$, then we show that $(D_i)_{i\geq 0}$ are non-summable. On the other hand, if $\sum_{i\geq 0}\varepsilon_{\star,i}^2 < \infty$, we show that (up to constants) $D_i$ is given by a discounted sum of score error energies from past generations.

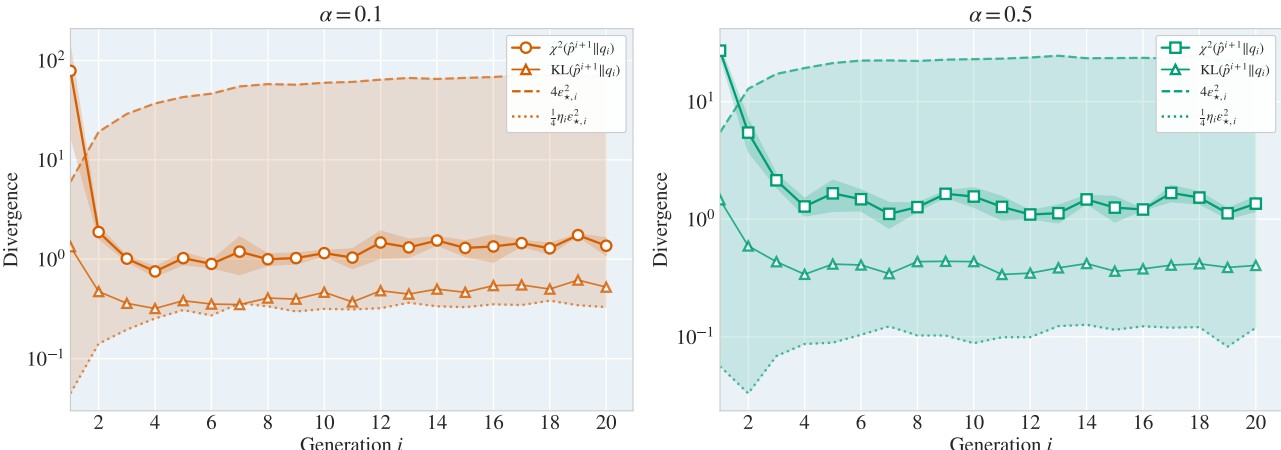

*Figure 4.* **Two-sided control of intra-generation divergence.** $p_{\text{data}} \sim \frac{1}{5}\sum_{k=1}^{5} \mathcal{N}(\boldsymbol{\mu}_k, \sigma^2 \mathbf{I}_{10}) \in \mathbb{R}^{10}$. We track the evolution of the divergences $I_i = \chi^2(\hat{p}^{i+1}\|q_i)$ and $I_i^{\text{KL}} = \text{KL}(\hat{p}^{i+1}\|q_i)$ (solid lines with markers) across 20 generations for low ($\alpha = 0.1$, left) and high ($\alpha = 0.5$, right) fresh data ratios. Consistent with Theorem 3.5, the intra-generational divergences are effectively bounded below and above by the pathwise score error energy (dashed upper bound $= 4\varepsilon_{\star,i}^2$) and the observability-weighted lower bound (dotted line $= \frac{1}{4}\hat{\eta}_i \varepsilon_{\star,i}^2$, where $\hat{\eta}_i$ is the estimated observability of errors, as detailed in Appendix G.1). Shaded regions indicate $\pm 1$ standard deviation across runs.

### 4.1. Persistent Errors

**Proposition 4.1** (Persistent Errors)**.** *Assume A1-A4, and that for $i \geq i_0$, $\eta_i \geq \underline{\eta} > 0$ and $\varepsilon_{\star,i}^2 \leq \min\{1, \frac{\eta_i}{8C}\}$. Then:*

*(i) Non-summable score error implies non-summable global drift. If $\sum_{i \geq 0} \varepsilon_{\star,i}^2 = +\infty$, then $\sum_{i \geq 0} D_i = +\infty$.*

*(ii) A score-error floor implies persistent accumulated divergence. If $\varepsilon_{\star,i}^2 \geq \underline{\varepsilon}^2 > 0$ for all sufficiently large $i$, then*

$$\limsup_{i \to \infty} D_i \geq \frac{\alpha}{16(1 + (1-\alpha)^2)} \cdot \underline{\eta}\,\underline{\varepsilon}^2,$$

*In particular, the sequence $(D_i)$ exhibits infinitely many macroscopic deviations from $p_{\text{data}}$.*

### 4.2. Controlled Errors

We now bound the accumulated divergence $D_i$ when the score errors $\varepsilon_{\star,i}^2$ are summable, under a technical assumption defined in terms of an adaptive tail set.

**An adaptive tail assumption** For a constant $\zeta_i \in (0,1)$, define the adaptive "good" set

$$\mathcal{G}_i := \{\mathbf{x} : \left| \hat{p}^i(\mathbf{x})/p_{\text{data}}(\mathbf{x}) - 1 \right| \leq \sqrt{D_i/\zeta_i}\}. \quad (22)$$

Since $\mathbb{E}_{p_{\text{data}}}[|\hat{p}^i(x)/p_{\text{data}} - 1|^2] = D_i$, Chebyshev's inequality implies $p_{\text{data}}(\mathcal{G}_i^c) \leq \zeta_i$. As $D_i$ grows, the set $\mathcal{G}_i$ expands and $\mathcal{G}_i^c$ consists of the regions where $\hat{p}^i/p_{\text{data}}$ is increasingly atypical. We then assume that the training distribution $q_i$ does not over-emphasize this tail region in the following sense.

**(A5)** There exist a generation $i_0 \geq 0$ such that, for the $\delta$ in Assumption A3 and $p' := \frac{\delta+1}{\delta-1}$, there exists a constant $C_\zeta > 0$ such that,

$$\sup_{i \geq i_0} \left\{ \zeta_i^{-1} \mathbb{E}_{q_i}\left[\left(\frac{q_i}{p_{\text{data}}}\right)^{p'} \mathbf{1}_{\mathcal{G}_i^c}\right] \right\} \leq C_\zeta.$$

From the definition of $q_i$ in (1), we know that $q_i/p_{\text{data}} = \alpha + (1-\alpha)\hat{p}^i/p_{\text{data}}$, which can blow up on $\mathcal{G}_i^c$ only if $\hat{p}^i/p_{\text{data}}$ is large there. Therefore, (A5) is a local condition on a vanishing tail set, which rules out spurious heavy tails that may cause the synthetic model to place large mass in regions where $p_{\text{data}}$ is tiny.

**Theorem 4.2** (Discounted accumulation of errors)**.** *Assume A1-A5, and that for all $i \geq i_0$, we have $\eta_i \geq \underline{\eta} > 0$ and $\varepsilon_{\star,i}^2 \leq \min\left\{1, \frac{\eta_i}{8C}\right\}$. Also assume that*

$$\sum_{i=0}^{\infty} \varepsilon_{\star,i}^2 < \infty.$$

*Then, there exists a constant $C_{\text{bias}} > 0$ defined in Equation (115) such that for each generation $N \geq i_0$,*

$$\begin{aligned} D_{N+1} &+ C_{\text{bias}} \\ &\asymp \sum_{i=i_0}^{N} (1-\alpha)^{2(N-i)} \varepsilon_{\star,i}^2 + (1-\alpha)^{2(N+1-i_0)} D_{i_0}. \end{aligned} \quad (23)$$

The proof is given in Appendix F.3, where we derive upper and lower bounds for $D_{N+1}$ with explicit constants. Theorem 4.2 shows that in the regime where the score errors

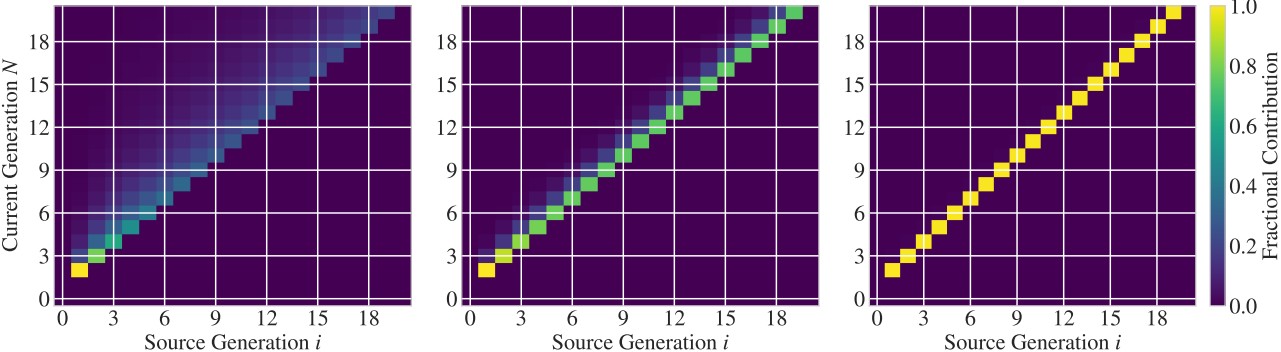

*Figure 5.* Geometrically-discounted decomposition of accumulated divergence in (23), for $p_{\text{data}} = \frac{1}{5}\sum_{k=1}^{5}\mathcal{N}(\boldsymbol{\mu}_k, \sigma^2\mathbf{I}_{10}) \in \mathbb{R}^{10}$. Each panel shows contributions of $\varepsilon_{\star,i}^2$ to the current global divergence $D_N = \chi^2(\hat{p}^N \| p_{\text{data}})$, for $i \leq N$. **Left** ($\alpha = 0.1$): Errors persist across many generations, creating a wide band of contributions. **Center** ($\alpha = 0.5$): Intermediate regime with moderate memory decay. **Right** ($\alpha = 0.9$): Short memory—only the most recent generation dominates, yielding a sharp diagonal structure. The plots are consistent with Theorem 4.2, confirming that higher fractions of fresh data accelerate the forgetting of past errors. Experiment details in Appendix G.1.

are small and summable, the accumulated divergence is a geometrically-discounted sum of square errors. In particular, errors from $m$ generations in the past are suppressed by a factor $(1 - \alpha)^{2m}$, and the larger the proportion of fresh training samples in each round, the higher the rate of "forgetting" score errors. Qualitatively, this is consistent with the findings in many different settings that incorporating fresh data in each training round can mitigate model collapse (Gerstgrasser et al., 2024; Zhu et al., 2025; Bertrand et al., 2024; Dey et al., 2025).

Figure 5 illustrates the decomposition of Theorem 4.2 for a 10-dimensional Gaussian mixture, tracking the contribution to $D_{N+1}$ of each term in the sum in (23). Figure 10 in Appendix G.3 shows the same behavior on the Fashion-MNIST dataset (Xiao et al., 2017)). Figure 6 in Appendix G.1 illustrates that the functional dependence predicted by Theorem 4.2 accurately captures the observed growth across generations in the 10-dimensional Gaussian mixture setting.

## 5. Conclusion and Future Work

We analyzed recursive training of diffusion models with a fixed proportion $\alpha > 0$ of fresh data in each training round. We derived lower and upper bounds on the intra-generation and accumulated divergences, in terms of the path-wise energy of the score error in each generation. An essential component of our lower bound is the observability coefficient in (17), which quantifies the fraction of the path-wise error energy that manifests in the learned distribution. The paper focuses on the setting where the score error energy in each generation is small. An important direction for future work is to establish bounds in the large error regime. Obtaining a lower bound in this regime, even for one generation, is challenging because the ratio between the learned

and ideal densities (given by the projection of the path-wise ratio in (15)) cannot be linearized with reasonable control of the remainder term. Another direction for further work is to take into account the discretization error due to the diffusion being implemented in discrete-time. Finally, a key open question is: is there a limiting distribution to which the model converges when recursively trained, and, if so, how it depends on $\alpha$ and $p_{\text{data}}$?

## Impact Statement

This paper presents work whose goal is to advance the field of Machine Learning. There are many potential societal consequences of our work, none which we feel must be specifically highlighted here.

## Acknowledgements

NBK is supported by a G-Research Trinity College Studentship, and RET is supported in part by the EPSRC Probabilistic AI Hub (EP/Y028783/1). RV was supported in part by an EPSRC Mathematical Sciences Small Grant.

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

# A. Preliminaries and notation for Appendices

We consider reverse-time diffusions on the interval $[t_0, T]$ (integrated from $T$ down to $t_0$). Let $\bar{\mathbf{B}} = (\bar{\mathbf{B}}_s)_{s \in [t_0, T]}$ denote a standard Brownian motion in reverse time on a complete filtered probability space $(\Omega, \mathcal{F}, (\mathcal{F}_s)_{s \in [t_0, T]}, \mathbb{P})$.

**Indexing** The superscript $i$ denotes the *generation* (self-training iteration). Within each generation we compare: (i) an *ideal* reverse process (driven by the exact score) and (ii) a *learned* reverse process (driven by the estimated score).

**Reverse-time SDEs and path measures** For $s \in [t_0, T]$, the ideal and learned reverse-time processes solve

$$\mathrm{d}\mathbf{Y}_s^{i,\star} = \left[ -\tfrac{1}{2}\mathbf{Y}_s^{i,\star} - \mathbf{s}_i^{\star}(\mathbf{Y}_s^{i,\star}, s) \right]\mathrm{d}s + \mathrm{d}\bar{\mathbf{B}}_s, \qquad \mathrm{d}\hat{\mathbf{Y}}_s^{i} = \left[ -\tfrac{1}{2}\hat{\mathbf{Y}}_s^{i} - \mathbf{s}_{\theta_i}(\hat{\mathbf{Y}}_s^{i}, s) \right]\mathrm{d}s + \mathrm{d}\bar{\mathbf{B}}_s,$$

with independent initializations $\mathbf{Y}_T^{i,\star}, \hat{\mathbf{Y}}_T^{i} \sim \mathcal{N}(0, I_d)$ (since we ignore the initialization mismatch, i.e. assume $q_{i,T} = \mathcal{N}(0, \mathbf{I}_d)$). We write $\mathbb{P}_i^{\star}$ (resp. $\hat{\mathbb{P}}_i$) for the law on path space $C([t_0, T], \mathbb{R}^d)$ induced by $\mathbf{Y}^{i,\star}$ (resp. $\hat{\mathbf{Y}}^{i}$). We write for any $s \in [t_0, T]$ $q_{i,s} = \mathrm{Law}(\mathbf{Y}_s^{i,\star})$ and $\hat{q}_{i,s} = \mathrm{Law}(\hat{\mathbf{Y}}_s^{i})$.

**Marginal likelihood ratios** For any $\mathbf{x} \in \mathbb{R}^d$, we denote the marginal density ratios by

$$R_i(\mathbf{x}) := \frac{\hat{p}^{\,i+1}(\mathbf{x})}{q_i(\mathbf{x})}, \qquad T_i(\mathbf{x}) := \frac{q_i(\mathbf{x})}{p_{\mathrm{data}}(\mathbf{x})}, \qquad \text{where } q_i := \alpha p_{\mathrm{data}} + (1 - \alpha)\hat{p}^{\,i}.$$

**Divergences** We denote, for a generation $i \geq 0$ and $\alpha \in (0, 1)$.

$$I_i := \chi^2(\hat{p}^{\,i+1} \| q_i) = \mathbb{E}_{q_i}[(R_i - 1)^2], \qquad D_i := \chi^2(\hat{p}^{\,i} \| p_{\mathrm{data}}).$$

**Norms** Let $\mu$ denote a probability distribution on $\mathbb{R}^d$, assumed absolutely continuous with respect to the Lebesegue measure on $\mathbb{R}^d$, and with Radon-Nikodym derivative (w.r.t the Lebesgue measure on $\mathbb{R}^d$) denoted $\mu$ by abuse of notations. We denote the $L^2(\mu)$ the space of vector fields of finite second moment w.r.t. $\mu$. It is an Hilbert space when equipped with the inner product,

$$\langle \mathbf{v}, \mathbf{u} \rangle_{L^2(\mu)} = \int_{\mathbb{R}^d} \mathbf{v}(\mathbf{x}) \cdot \mathbf{u}(\mathbf{x}) \, \mu(\mathbf{x})\mathrm{d}\mathbf{x}.$$

This allows to define the $L^2(\mu)$ norm, for any $\mathbf{v} \in L^2(\mu)$ as,

$$\|\mathbf{v}\|_{L^2(\mu)} = \left( \int_{\mathbb{R}^d} \|\mathbf{v}(\mathbf{x})\|_2^2 \, \mu(\mathbf{x})\mathrm{d}\mathbf{x} \right)^{1/2}.$$

More generally, for any $r \neq 0$, $L^r(\mu)$ is a Banach space with associated norm defined for any $\mathbf{v} \in L^r(\mu)$,

$$\|\mathbf{v}\|_{L^r(\mu)} = \left( \int \|\mathbf{v}(\mathbf{x})\|_2^r \, \mu(\mathbf{x})\mathrm{d}\mathbf{x} \right)^{1/r}.$$

**Score error and energy** The score (drift) discrepancy is defined for all $\mathbf{x} \in \mathbb{R}^d$ and all times $t \in [t_0, T]$ as,

$$\mathbf{e}_{i,t}(\mathbf{x}) := \mathbf{s}_{\theta_i}(\mathbf{x}, t) - \mathbf{s}_i^{\star}(\mathbf{x}, t).$$

We recall the two score error energies introduced in (10)-(11)):

$$\varepsilon_{\star,i}^2 := \int_{t_0}^{T} \|\mathbf{e}_{i,s}\|_{L^2(q_{i,s})}^2 \, \mathrm{d}s, \qquad \hat{\varepsilon}_i^2 := \int_{t_0}^{T} \|\mathbf{e}_{i,s}\|_{L^2(\hat{q}_{i,s})}^2 \, \mathrm{d}s.$$

**Perturbative Regime** The perturbative regime refers to the setting where there exists a generation $i_0 \geq 0$ such that,

$$\varepsilon_{\star,i}^2 \leq \min\left\{ 1, \frac{\eta_i}{8C} \right\}, \qquad \forall i \geq i_0,$$

where the constant $C := \frac{1 + K_\gamma}{8} + \frac{1}{4}\left( C_{\mathrm{BDG}}(4p') K_\gamma^{\frac{2p'}{2+\gamma}} \right)^{1/p'} C_\delta^{1/p}$ is defined in Proposition D.1.

**Girsanov Quantitites** For all generation $i \geq 0$, we define the time-integrated random error,

$$M_t^i := -\int_{t_0}^t \mathbf{e}_{i,s} \cdot \mathrm{d}\bar{\mathbf{B}}_s, \tag{24}$$

and its quadratic variation

$$\langle M^i \rangle_t = \int_{t_0}^t \|\mathbf{e}_{i,s}\|_2^2 \, \mathrm{d}s. \tag{25}$$

Importantly, we note that both $M_t^i$ and $\langle M^i \rangle_t$ are *functions of a random trajectory* $(\mathbf{Y}_s)_{s \in [t_0, t]}$, distributed according to a path-space measure, e.g. $\mathbb{P}_i^\star$ or $\hat{\mathbb{P}}_i$. We additionally define $Z_t^i := M_t^i - \frac{1}{2}\langle M^i \rangle_t$, which is also a function of random trajectories.

## B. Intra-generational likelihood ratios via Girsanov's theorem

### B.1. Proof of Upper Bound (Proposition 3.1)

We recall the definitions of the ideal and learned reverse-time processes on $s \in [t_0, T]$ (integrated backward from $T$ to $t_0$) from (7) and (8), both initialized at $\mathcal{N}(0, \mathbf{I}_d)$. Recall we defined two path-space probability measures $\mathbb{P}_i^\star$ and $\hat{\mathbb{P}}_i$ on the path space $C([t_0, T], \mathbb{R}^d)$:

- $\mathbb{P}_i^\star$: The law of the *ideal* reverse-time process (7). It is initialized at end time $T$ with Gaussian noise, and its dynamics are driven by the exact score $\mathbf{s}_i^\star$. Its terminal marginal at $t_0$ is the current mixing distribution $q_i$.

- $\hat{\mathbb{P}}_i$: The law of the *learned* reverse-time process (8). It is initialized at $T$ with Gaussian noise, and its dynamics are driven by the approximate score $\mathbf{s}_{\theta_i}$. Its terminal marginal at $t_0$ is the next-generation distribution $\hat{p}^{i+1}$.

Recall that we operate under the following assumptions:

**A1 Finite drift energy.** The learned path measure satisfies $\hat{\varepsilon}_i^2 < \infty$.

**A2 Martingale property.** For any $s \in [t_0, T]$, recalling the definitions of $M_s^i$ and $\langle M^i \rangle_s$ from (24) and (25), we define the Doléans–Dade exponential

$$U_s^i := \exp\left(M_s^i - \frac{1}{2}\langle M^i \rangle_s\right) = \exp(Z_s^i). \tag{26}$$

We assume $(U_s^i)_{s \in [t_0, T]}$ defines a true $\mathbb{P}_i^\star$-martingale on $[t_0, T]$.

**Discussion on Assumptions A1 and A2.** These assumptions ensure that the Girsanov transformation yields a valid probability measure (i.e., $\mathbb{E}_{\mathbb{P}_i^\star}[U_T] = 1$). While $U_s$ is guaranteed to be a local martingale by construction, different sufficient conditions can be found in the literature to guarantee the full martingale status. The most common ones are:

- *Novikov's condition* (Novikov, 1973):

$$\mathbb{E}_{(\mathbf{Y}_s)_{s \in [t_0, T]} \sim \mathbb{P}_i^\star}\left[\exp\left(\frac{1}{2}\int_{t_0}^T \|\mathbf{e}_{i,s}(\mathbf{Y}_s)\|_2^2 \, \mathrm{d}s\right)\right] < \infty.$$

- *Beneš Condition (Beneš, 1971)*: the drift discrepancy satisfies a linear growth bound

$$\|\mathbf{e}_{i,t}(\mathbf{x})\|_2 \leq C(1 + \|\mathbf{x}\|_2), \qquad C > 0.$$

In the context of diffusion models parameterized by neural networks, this holds if the network weights are finite and the activation functions are Lipschitz (e.g., ReLU, SiLU) or bounded (e.g., Tanh, Sigmoid), provided the domain is effectively bounded or the growth is controlled.

- *Stopping Time Condition.* Finally, another popular assumption relies on stopping times. Formally, the exponential process $(U_s)_{s\in[t_0,T]}$ is always a *local martingale*. This implies the existence of a non-decreasing sequence of stopping times $\{\tau_n\}_{n\geq 1}$ such that $\tau_n \uparrow T$ almost surely as $n \to \infty$, and for every $n$, the stopped process $(U_{s\wedge\tau_n})_{s\in[t_0,T]}$ is a true martingale. A canonical choice for these stopping times is defined by the accumulation of the drift energy:

$$\tau_n := \inf\left\{s \in [t_0, T] : \int_{t_0}^s \|\mathbf{e}_{i,u}\|_2^2 \, \mathrm{d}u \geq n\right\} \wedge T. \tag{27}$$

By definition, the accumulated energy up to time $\tau_n$ is bounded by $n$. Consequently, the conditional Novikov criterion is trivially satisfied for the stopped process:

$$\mathbb{E}_{(\mathbf{Y}_s)_{s\in[t_0,T]}\sim\mathbb{P}_i^\star}\left[\exp\left(\frac{1}{2}\int_{t_0}^{\tau_n}\|\mathbf{e}_{i,s}(\mathbf{Y}_s)\|_2^2\,\mathrm{d}u\right)\right] \leq e^{n/2} < \infty.$$

This guarantees that $\mathbb{E}[U_{\tau_n}^i] = 1$ for all $n$. Assumption A2 is therefore equivalent to the condition that the sequence is uniformly integrable, allowing the equality to hold in the limit: $\mathbb{E}[U_T^i] = \lim_{n\to\infty}\mathbb{E}[U_{\tau_n}^i] = 1$. This stopping time condition has been used in previous analyses of the convergence properties of diffusion models (Chen et al., 2023c; Benton et al., 2024).

*Proof of Proposition 3.1. Path-space Radon-Nikodym Derivative* The ideal process and learned reverse processes are defined on the filtered probability space $(\Omega, \mathcal{F}, (\mathcal{F}_s)_{s\in[t_0,T]}, \mathbb{P}_i^\star)$ equipped with the standard Brownian motion $\bar{\mathbf{B}}$. They share the same diffusion coefficient and their drifts terms differ by exactly $\Delta\mathbf{b}_s = -\mathbf{e}_{i,s}$. This observation allows the use of Girsanov's theorem (Cameron & Martin, 1944; Girsanov, 1960; Le Gall, 2018) to relate their path-space laws. The Doléans-dade exponential (Doléans-Dade, 1970) $(U_s^i)_{s\in[t_0,T]}$ defined in (26) by $U_s^i = \exp\left(M_s^i - \frac{1}{2}\langle M^i\rangle_s\right)$ is, by Itô's formula (Karatzas & Shreve, 2014; Le Gall, 2018), a nonnegative local martingale under $\mathbb{P}_i^\star$. By Assumption A2, $(U_s^i)_{s\in[t_0,T]}$ is in fact a (uniformly integrable) martingale, so in particular $\mathbb{E}_{\mathbb{P}_i^\star}[U_T^i] = U_{t_0}^i = 1$. Therefore, by Girsanov's theorem, the Radon-Nikodym derivative between $\hat{\mathbb{P}}_i$ and $\mathbb{P}_i^\star$ is given by,

$$\frac{\mathrm{d}\hat{\mathbb{P}}_i}{\mathrm{d}\mathbb{P}_i^\star}((\mathbf{Y}_s)_{s\in[t_0,T]}) = U_T^i = \exp\left(-\int_{t_0}^T \mathbf{e}_{i,s}(\mathbf{Y}_s)\cdot\mathrm{d}\bar{\mathbf{B}}_s - \frac{1}{2}\int_{t_0}^T\|\mathbf{e}_{i,s}(\mathbf{Y}_s)\|_2^2\,\mathrm{d}s\right).$$

Under this new measure $\hat{\mathbb{P}}_i$, the process $\hat{\mathbf{B}}_s := \bar{\mathbf{B}}_s + \int_{t_0}^s \mathbf{e}_{i,u}(\mathbf{Y}_u)\mathrm{d}u$ is a standard Brownian motion (by the Girsanov theorem), and the coordinate process $\mathbf{Y}$ satisfies the SDE of the learned reverse diffusion (8). Thus, under $\hat{\mathbb{P}}_i$, we can write $\mathrm{d}\bar{\mathbf{B}}_s = \mathrm{d}\hat{\mathbf{B}}_s - \mathbf{e}_{i,s}(\hat{\mathbf{Y}}_s)\mathrm{d}s$.

*KL Computations.* To compute the Kullback-Leibler divergence defined as $\mathrm{KL}(\hat{\mathbb{P}}_i\|\mathbb{P}_i^\star) = \mathbb{E}_{\hat{\mathbb{P}}_i}[\log\frac{\mathrm{d}\hat{\mathbb{P}}_i}{\mathrm{d}\mathbb{P}_i^\star}]$, we first rewrite the log-likelihood ratio,

$$\begin{aligned}
\log\frac{\mathrm{d}\hat{\mathbb{P}}_i}{\mathrm{d}\mathbb{P}_i^\star}((\mathbf{Y}_s)_{s\in[t_0,T]}) &= -\int_{t_0}^T \mathbf{e}_{i,s}(\mathbf{Y}_s)\cdot\mathrm{d}\bar{\mathbf{B}}_s - \frac{1}{2}\int_{t_0}^T\|\mathbf{e}_{i,s}(\mathbf{Y}_s)\|_2^2\,\mathrm{d}s \\
&= -\int_{t_0}^T \mathbf{e}_{i,s}\cdot(\mathrm{d}\hat{\mathbf{B}}_s - \mathbf{e}_{i,s}\mathrm{d}s) - \frac{1}{2}\int_{t_0}^T\|\mathbf{e}_{i,s}(\mathbf{Y}_s)\|_2^2\,\mathrm{d}s \\
&= -\int_{t_0}^T \mathbf{e}_{i,s}(\mathbf{Y}_s)\cdot\mathrm{d}\hat{\mathbf{B}}_s + \frac{1}{2}\int_{t_0}^T\|\mathbf{e}_{i,s}(\mathbf{Y}_s)\|_2^2\,\mathrm{d}s.
\end{aligned}$$

Taking expectation under $\hat{\mathbb{P}}_i$:

$$\mathbb{E}_{\hat{\mathbb{P}}_i}\left[\log\frac{\mathrm{d}\hat{\mathbb{P}}_i}{\mathrm{d}\mathbb{P}_i^\star}\right] = \mathbb{E}_{\hat{\mathbb{P}}_i}\left[-\int_{t_0}^T \mathbf{e}_{i,s}\cdot\mathrm{d}\hat{\mathbf{B}}_s\right] + \frac{1}{2}\mathbb{E}_{\hat{\mathbb{P}}_i}\left[\int_{t_0}^T\|\mathbf{e}_{i,s}(\mathbf{Y}_s)\|_2^2\,\mathrm{d}s\right].$$

The first term is the expectation of a stochastic integral. Under Assumption A1, the integrand is square-integrable, so the stochastic integral is a true martingale starting at 0, and its expectation vanishes. The second term is exactly $\frac{1}{2}\hat{\varepsilon}_i^2$.

*Marginalization (Data Processing).* Let $\pi_{t_0} : C([t_0, T]) \to \mathbb{R}^d$ be the projection map $\omega \mapsto \omega(t_0)$. The marginal distributions are push-forwards: $\hat{p}^{i+1} = (\pi_{t_0})_{\#}\hat{\mathbb{P}}_i$ and $q_i = (\pi_{t_0})_{\#}\mathbb{P}_i^\star$. By the data processing inequality (contraction of KL under push-forward):

$$\mathrm{KL}(\hat{p}^{i+1}\|q_i) \leq \mathrm{KL}(\hat{\mathbb{P}}_i\|\mathbb{P}_i^\star) = \frac{1}{2}\hat{\varepsilon}_i^2.$$

$\square$

## B.2. Relating Marginals via Girsanov's Theorem

**Proposition B.1** (Marginal Ratio Representation). *Under Assumptions A1 and A2, the Radon-Nikodym derivative between the path measures is given by $e^{Z_T^i}$, where $Z_T^i = M_T^i - \frac{1}{2}\langle M^i \rangle_T$. Consequently, the marginal likelihood ratio satisfies:*

$$R_i(\mathbf{x}) := \frac{\hat{p}^{i+1}(\mathbf{x})}{q_i(\mathbf{x})} = \mathbb{E}_{\mathbb{P}_i^\star}\left[ e^{Z_T^i} \mid \mathbf{Y}_{t_0} = \mathbf{x} \right], \quad q_i\text{-almost everywhere.} \tag{28}$$

*Proof.* As shown in Appendix B.1, we have $\frac{d\hat{\mathbb{P}}_i}{d\mathbb{P}_i^\star} = \exp\left(Z_T^i\right)$. Next, we restrict this relation to the marginals at time $t_0$. Let $f : \mathbb{R}^d \to \mathbb{R}$ be any bounded measurable test function acting on the data space. We compute the expectation of $f$ under the generated distribution $\hat{p}^{i+1}$:

$$\mathbb{E}_{\mathbf{x}\sim\hat{p}^{i+1}}[f(\mathbf{x})] = \mathbb{E}_{(\mathbf{Y}_s)_{s\in[t_0,T]}\sim\hat{\mathbb{P}}_i}[f(\mathbf{Y}_{t_0})] \qquad \text{(Definition of marginal)}$$

$$= \mathbb{E}_{(\mathbf{Y}_s)_{s\in[t_0,T]}\sim\mathbb{P}_i^\star}\left[ f(\mathbf{Y}_{t_0}) \frac{d\hat{\mathbb{P}}_i}{d\mathbb{P}_i^\star} \right] \qquad \text{(Change of measure)}$$

$$= \mathbb{E}_{(\mathbf{Y}_s)_{s\in[t_0,T]}\sim\mathbb{P}_i^\star}\left[ f(\mathbf{Y}_{t_0}) e^{Z_T^i} \right] \qquad \text{(Definition of } Z_T^i\text{)}$$

$$= \mathbb{E}_{(\mathbf{Y}_s)_{s\in[t_0,T]}\sim\mathbb{P}_i^\star}\left[ f(\mathbf{Y}_{t_0}) \mathbb{E}_{\mathbb{P}_i^\star}[e^{Z_T^i} \mid \mathbf{Y}_{t_0}] \right] \qquad \text{(Tower property).}$$

Comparing the first and last lines, we identify $\mathbb{E}_{\mathbb{P}_i^\star}[e^{Z_T^i} \mid \mathbf{Y}_{t_0}]$ as the density ratio $\frac{\hat{p}^{i+1}}{q_i}$. $\square$

## C. Moment Control and Observability Lemmas

Proposition B.1 expresses the density ratio as the conditional expectation of $\exp(Z_T^i)$ given $\mathbf{Y}_{t_0}$, where $Z_T^i = M_T^i - \frac{1}{2}\langle M^i \rangle_T$. The score error functional $Z_T^i$ is defined on the entire path, but only its projection onto the terminal state $\mathbf{Y}_{t_0}$ affects the marginal divergence. The first lemma in this section lower bounds the second moment of this projection.

**Lemma C.1** (Observability of Errors Transfer). *Assume A1 and A4. Assume $\varepsilon_{\star,i}^2 < \infty$. For any $\theta \in (0,1)$, the conditional expectation of the pathwise error satisfies the lower bound:*

$$\mathbb{E}_{\mathbb{P}_i^\star}\left[ \mathbb{E}_{\mathbb{P}_i^\star}[Z_T^i \mid \mathbf{Y}_{t_0}]^2 \right] \geq (1-\theta)\,\eta_i\,\varepsilon_{\star,i}^2 - \frac{1}{4}\left(\frac{1}{\theta} - 1\right)K_\gamma^{\frac{2}{2+\gamma}}\varepsilon_{\star,i}^4 \tag{29}$$

*Proof.* Recall the definition $Z_T^i = M_T^i - \frac{1}{2}\langle M^i \rangle_T$. We condition on the terminal state $\mathbf{Y}_{t_0}$ under the ideal measure $\mathbb{P}_i^\star$. Using Young's weighted inequality for any $\theta \in (0,1)$, $(a-b)^2 \geq (1-\theta)a^2 - \left(\frac{1}{\theta} - 1\right)b^2$, we have:

$$\left(\mathbb{E}_{\mathbb{P}_i^\star}[Z_T^i \mid \mathbf{Y}_{t_0}]\right)^2 \geq (1-\theta)\left(\mathbb{E}_{\mathbb{P}_i^\star}[M_T^i \mid \mathbf{Y}_{t_0}]\right)^2 - \frac{1}{4}\left(\frac{1}{\theta} - 1\right)\left(\mathbb{E}_{\mathbb{P}_i^\star}[\langle M^i \rangle_T \mid \mathbf{Y}_{t_0}]\right)^2.$$

Taking expectation with respect to $\mathbb{P}_i^\star$ yields:

$$\mathbb{E}_{\mathbb{P}_i^\star}\left[\left(\mathbb{E}_{\mathbb{P}_i^\star}[Z_T^i \mid \mathbf{Y}_{t_0}]\right)^2\right] \geq (1-\theta)\underbrace{\mathbb{E}_{\mathbb{P}_i^\star}\left[\left(\mathbb{E}_{\mathbb{P}_i^\star}[M_T^i \mid \mathbf{Y}_{t_0}]\right)^2\right]}_{\text{(Signal)}} - \frac{1}{4}\left(\frac{1}{\theta} - 1\right)\underbrace{\mathbb{E}_{\mathbb{P}_i^\star}\left[\left(\mathbb{E}_{\mathbb{P}_i^\star}[\langle M^i \rangle_T \mid \mathbf{Y}_{t_0}]\right)^2\right]}_{\text{(Noise)}}. \tag{30}$$

*Analysis of the Signal Term:* Since $M^i$ is an Itô integral with square integrable adapted integrand ($\varepsilon_{\star,i}^2 < \infty$ by assumption) it is a martingale with $M_{t_0}^i = 0$ hence $\mathbb{E}_{\mathbb{P}_i^\star}[M_T^i] = 0$. By the tower property of conditional expectation:

$$\mathbb{E}_{\mathbb{P}_i^\star}\left[\mathbb{E}_{\mathbb{P}_i^\star}[M_T^i \mid \mathbf{Y}_{t_0}]\right] = \mathbb{E}_{\mathbb{P}_i^\star}[M_T^i] = 0.$$

Thus, the second moment of the conditional expectation is equal to its variance:

$$\mathbb{E}_{\mathbb{P}_i^\star}\left[\left(\mathbb{E}_{\mathbb{P}_i^\star}[M_T^i \mid \mathbf{Y}_{t_0}]\right)^2\right] = \mathrm{Var}_{\mathbb{P}_i^\star}\left(\mathbb{E}_{\mathbb{P}_i^\star}[M_T^i \mid \mathbf{Y}_{t_0}]\right).$$

By Definition 3.2, the observability coefficient is $\eta_i := \mathrm{Var}(\mathbb{E}[M \mid Y])/\mathbb{E}[\langle M \rangle]$. Therefore:

$$\mathbb{E}_{\mathbb{P}_i^\star}\left[\left(\mathbb{E}_{\mathbb{P}_i^\star}[M_T^i \mid \mathbf{Y}_{t_0}]\right)^2\right] = \eta_i \cdot \mathbb{E}_{\mathbb{P}_i^\star}[\langle M^i \rangle_T] = \eta_i \varepsilon_{\star,i}^2.$$

*Analysis of Noise Term:* Applying Jensen's inequality for conditional expectation,

$$\left(\mathbb{E}_{\mathbb{P}_i^\star}[\langle M^i \rangle_T \mid \mathbf{Y}_{t_0}]\right)^2 \leq \mathbb{E}_{\mathbb{P}_i^\star}\left[\langle M^i \rangle_T^2 \mid \mathbf{Y}_{t_0}\right].$$

Taking the outer expectation $\mathbb{E}_{\mathbb{P}_i^\star}$ and using the tower property:

$$\begin{aligned}
\mathbb{E}_{\mathbb{P}_i^\star}\left[\left(\mathbb{E}_{\mathbb{P}_i^\star}[\langle M^i \rangle_T \mid \mathbf{Y}_{t_0}]\right)^2\right] &\leq \mathbb{E}_{\mathbb{P}_i^\star}\left[\mathbb{E}_{\mathbb{P}_i^\star}[\langle M^i \rangle_T^2 \mid \mathbf{Y}_{t_0}]\right] = \mathbb{E}_{\mathbb{P}_i^\star}[\langle M^i \rangle_T^2] \\
&\leq \mathbb{E}_{\mathbb{P}_i^\star}[\langle M^i \rangle_T^{2+\gamma}]^{\frac{2}{2+\gamma}} \\
&\leq K_\gamma^{\frac{2}{2+\gamma}} \varepsilon_{\star,i}^4, \quad \text{by } A4
\end{aligned}$$

Substituting the results for these two terms back into (30) gives the result. $\qquad\square$

The next lemma controls the second and fourth moments of $Z_T^i$.

**Lemma C.2** (Second and Fourth moment bound for the Girsanov log-density). *Assume A4. Then,*

$$\mathbb{E}_{\mathbb{P}_i^\star}[(Z_T^i)^2] \leq 2\varepsilon_{\star,i}^2 + \frac{1}{2}K_\gamma^{\frac{2}{2+\gamma}}\varepsilon_{\star,i}^4. \tag{31}$$

*Moreover, in the regime where $\varepsilon_{\star,i}^2 \leq 1$ there exists a constant $C_{Z4} < \infty$ such that, uniformly in $i \geq i_0$,*

$$\mathbb{E}_{\mathbb{P}_i^\star}\left[Z_T^{i\,4}\right] \leq C_{Z4}\,\varepsilon_{\star,i}^4. \tag{32}$$

*Proof. Second Moment Bound.* Using $(a-b)^2 \leq 2a^2 + 2b^2$ on $Z_T^i = M_T^i - \frac{1}{2}\langle M^i \rangle_T$, we have:

$$(Z_T^i)^2 \leq 2(M_T^i)^2 + \frac{1}{2}\langle M^i \rangle_T^2.$$

Taking expectations under $\mathbb{P}_i^\star$ and applying Itô's isometry ($\mathbb{E}[(M_T^i)^2] = \varepsilon_{\star,i}^2$) and Assumption A4:

$$\mathbb{E}_{\mathbb{P}_i^\star}[(Z_T^i)^2] \leq 2\varepsilon_{\star,i}^2 + \frac{1}{2}K_\gamma^{\frac{2}{2+\gamma}}\varepsilon_{\star,i}^4. \tag{33}$$

*Fourth Moment Bound.* Recall that $Z_T^i = M_T^i - \frac{1}{2}\langle M^i \rangle_T$. Using the elementary inequality $(a-b)^4 \leq 8a^4 + 8b^4$ we obtain

$$Z_T^{i\,4} \leq 8\,|M_T^i|^4 + 8\left(\frac{1}{2}\langle M^i \rangle_T\right)^4 = 8\,|M_T^i|^4 + \frac{1}{2}\,\langle M^i \rangle_T^4. \tag{34}$$

By the Burkholder–Davis–Gundy inequality (Burkholder et al., 1972; Revuz & Yor, 1999) for continuous martingales with exponent $p = 4$, there exists a constant $C_4$ such that

$$\mathbb{E}_{\mathbb{P}_i^\star}\left[|M_T^i|^4\right] \leq C_4\,\mathbb{E}_{\mathbb{P}_i^\star}\left[\langle M^i \rangle_T^2\right]$$

By Assumption A4 for $\gamma > \max\{2, \frac{4}{\delta-1}\}$,

$$\mathbb{E}_{\mathbb{P}_i^\star}\left[\langle M^i \rangle_T^2\right] \leq \mathbb{E}_{\mathbb{P}_i^\star}\left[\langle M^i \rangle_T^{2+\gamma}\right]^{\frac{2}{2+\gamma}} \leq K_\gamma^{\frac{2}{2+\gamma}}\varepsilon_{\star,i}^4,$$

hence

$$\mathbb{E}_{\mathbb{P}_i^\star}\big[|M_T^i|^4\big] \;\leq\; C\,\varepsilon_{\star,i}^4. \tag{35}$$

Similarly, by Assumption A4,

$$\mathbb{E}_{\mathbb{P}_i^\star}\big[\langle M^i\rangle_T^4\big] \;\leq\; \mathbb{E}_{\mathbb{P}_i^\star}\big[\langle M^i\rangle_T^{2+\gamma}\big]^{\frac{4}{2+\gamma}} \;\leq\; K_\gamma^{\frac{4}{2+\gamma}}\varepsilon_{\star,i}^8$$

Thus,

$$\mathbb{E}_{\mathbb{P}_i^\star}[(Z_T^i)^4] \leq 8K_\gamma^{\frac{2}{2+\gamma}}\,\varepsilon_{\star,i}^4 + \frac{K_\gamma^{\frac{4}{2+\gamma}}}{2}\varepsilon_{\star,i}^8$$

Observing that, whenever $\varepsilon_{\star,i}^2 \leq 1$, one has that $\varepsilon_{\star,i}^8 \leq \varepsilon_{\star,i}^4$, yields

$$\mathbb{E}_{\mathbb{P}_i^\star}\big[(Z_T^i)^4\big] \;\leq\; C\,\varepsilon_{\star,i}^4, \tag{36}$$

which proves the claim. $\qquad\square$

## D. Proof of the Intra-Generational Lower Bound (Proposition 3.3)

From Proposition B.1, recall that the ratio $\frac{\hat{p}^{i+1}}{q_i}$ is defined for all $\mathbf{x} \in \mathbb{R}^d$ by,

$$R_i(\mathbf{x}) \;:=\; \frac{\hat{p}^{i+1}(\mathbf{x})}{q_i(\mathbf{x})} \;=\; \mathbb{E}_{\mathbb{P}_i^\star}\Big[e^{Z_T^i} \,\Big|\, \mathbf{Y}_{t_0}\Big], \quad q_i\text{-almost everywhere.}$$

To obtain a lower bound, we write $e^{Z_T^i} = 1 + Z_T^i + \psi(Z_T^i)$ and control the remainder term $\psi(Z_T^i)$ under the assumptions of the proposition. To this end, we will use two lemmas proved in Appendix C. The first, Lemma C.1 controls the second moment of $\mathbb{E}_{\mathbb{P}_i^\star}\Big[Z_T^i \mid \mathbf{Y}_{t_0}\Big]$. The second, Lemma C.2, controls the second and fourth moments of $Z_T^i$.

We will prove the following version of Proposition 3.3 with explicit constants.

**Proposition D.1** (Intra-generational lower bound with explicit constants). *Assume A1–A4, with associated parameters $(\gamma, K_\gamma)$ and $(\delta, C_\delta)$. Let $C_{\mathrm{BDG}}(r)$ denote a Burkholder–Davis–Gundy (Burkholder et al., 1972; Revuz & Yor, 1999) constant such that for any continuous martingale $M$,*

$$\mathbb{E}\big[|M_T|^r\big] \leq C_{\mathrm{BDG}}(r)\,\mathbb{E}\big[\langle M\rangle_T^{r/2}\big]. \tag{37}$$

*Then, for every generation $i \geq i_0$ satisfying the perturbative condition $\varepsilon_{\star,i}^2 \leq 1$, one has*

$$\chi^2(\hat{p}^{i+1}\|q_i) \;\geq\; \frac{1}{4}\,\eta_i\,\varepsilon_{\star,i}^2 \;-\; C\,\varepsilon_{\star,i}^4, \tag{38}$$

*with the explicit choices*

$$C := \frac{K_\gamma^{\frac{2}{2+\gamma}}}{8} \;+\; \frac{C_{Z4}}{4} \;+\; \frac{1}{4}\Big(C_{\mathrm{BDG}}(4p')\,K_\gamma^{\frac{2p'}{2+\gamma}}\Big)^{1/p'}\,C_\delta^{1/p}, \tag{39}$$

*where $p = \frac{1+\delta}{2}$ and $p' = \frac{\delta+1}{\delta-1}$. In particular, defining the explicit perturbative threshold*

$$\kappa_i := \min\Big\{1,\; \frac{\eta_i}{8C}\Big\}, \tag{40}$$

*whenever $\varepsilon_{\star,i}^2 \leq \kappa_i$ we obtain the clean lower bound*

$$\chi^2(\hat{p}^{i+1}\|q_i) \;\geq\; \frac{c}{2}\,\eta_i\,\varepsilon_{\star,i}^2 = \frac{1}{8}\,\eta_i\,\varepsilon_{\star,i}^2. \tag{41}$$

*Proof.* The proof proceeds in six steps: we expand the density ratio using Taylor's theorem, apply Young's inequality to isolate the leading-order term, invoke the observability lemma for the signal, control the remainder using exponential integrability, and combine all bounds.

*1. Exponential expansion with controlled remainder.*

Define the remainder function $\psi : \mathbb{R} \to \mathbb{R}$ by

$$\psi(z) := e^z - 1 - z = \sum_{k=2}^{\infty} \frac{z^k}{k!}. \tag{42}$$

From Proposition B.1 (Marginal Ratio Representation), we have

$$R_i(\mathbf{x}) = \frac{\hat{p}^{i+1}(\mathbf{x})}{q_i(\mathbf{x})} = \mathbb{E}_{\mathbb{P}_i^\star}\left[e^{Z_T^i} \mid \mathbf{Y}_{t_0}\right]. \tag{43}$$

Applying the decomposition $e^{Z_T^i} = 1 + Z_T^i + \psi(Z_T^i)$ and taking conditional expectations:

$$R_i - 1 = \mathbb{E}_{\mathbb{P}_i^\star}[e^{Z_T^i} - 1 \mid \mathbf{Y}_{t_0}] = \mathbb{E}_{\mathbb{P}_i^\star}[Z_T^i \mid \mathbf{Y}_{t_0}] + \mathbb{E}_{\mathbb{P}_i^\star}[\psi(Z_T^i) \mid \mathbf{Y}_{t_0}].$$

We introduce the notation:

$$\bar{Z}_i := \mathbb{E}_{\mathbb{P}_i^\star}[Z_T^i \mid \mathbf{Y}_{t_0}] \quad \text{(conditional mean of log-likelihood ratio)}, \tag{44}$$

$$\Psi_i := \mathbb{E}_{\mathbb{P}_i^\star}[\psi(Z_T^i) \mid \mathbf{Y}_{t_0}] \quad \text{(conditional mean of remainder)}. \tag{45}$$

Thus we have the fundamental decomposition:

$$R_i - 1 = \bar{Z}_i + \Psi_i. \tag{46}$$

*2. Quadratic lower bound via Young's inequality.*

For any $\nu \in (0, 1)$, we apply the weighted Young's inequality in the form $(a + b)^2 \geq (1 - \nu)a^2 - (\frac{1}{\nu} - 1)b^2$:

$$(R_i - 1)^2 = (\bar{Z}_i + \Psi_i)^2 \geq (1 - \nu)\bar{Z}_i^2 - \left(\frac{1}{\nu} - 1\right)\Psi_i^2. \tag{47}$$

Taking expectations with respect to $q_i$ (equivalently, with respect to $\mathbf{Y}_{t_0}$ under $\mathbb{P}_i^\star$):

$$\chi^2(\hat{p}^{i+1}\|q_i) = \mathbb{E}_{q_i}[(R_i - 1)^2] \geq (1 - \nu)\underbrace{\mathbb{E}_{q_i}[\bar{Z}_i^2]}_{\text{Signal}} - \left(\frac{1}{\nu} - 1\right)\underbrace{\mathbb{E}_{q_i}[\Psi_i^2]}_{\text{Remainder}}. \tag{48}$$

We proceed to bound each term separately.

*3. The signal term—applying observability.* Since $\mathbf{Y}_{t_0} \sim q_i$ under $\mathbb{P}_i^\star$, we have

$$\mathbb{E}_{q_i}[\bar{Z}_i^2] = \mathbb{E}_{\mathbb{P}_i^\star}\left[\left(\mathbb{E}_{\mathbb{P}_i^\star}[Z_T^i \mid \mathbf{Y}_{t_0}]\right)^2\right]. \tag{49}$$

Invoking Lemma C.1, for any $\theta \in (0, 1)$:

$$\mathbb{E}_{\mathbb{P}_i^\star}\left[\left(\mathbb{E}_{\mathbb{P}_i^\star}[Z_T^i \mid \mathbf{Y}_{t_0}]\right)^2\right] \geq (1 - \theta)\,\eta_i\,\varepsilon_{\star,i}^2 - \frac{1}{4}\left(\frac{1}{\theta} - 1\right)K_\gamma^{\frac{2}{2+\gamma}}\varepsilon_{\star,i}^4. \tag{50}$$

For the noise term: by Jensen's inequality for conditional expectations,

$$\left(\mathbb{E}[\langle M^i\rangle_T \mid \mathbf{Y}_{t_0}]\right)^2 \leq \mathbb{E}[\langle M^i\rangle_T^2 \mid \mathbf{Y}_{t_0}]. \tag{51}$$

Taking outer expectations and applying the tower property:

$$\mathbb{E}\Big[\big(\mathbb{E}[\langle M^i\rangle_T \mid \mathbf{Y}_{t_0}]\big)^2\Big] \leq \mathbb{E}[\langle M^i\rangle_T^2] \leq K_\gamma^{\frac{2}{2+\gamma}} \varepsilon_{\star,i}^4, \tag{52}$$

where the final inequality uses Assumption A4.

*4. The remainder term—exponential moment control.*

We bound $\mathbb{E}_{q_i}[\Psi_i^2]$ using the exponential integrability condition (Assumption A3). By Jensen's inequality for conditional expectations (since $x \mapsto x^2$ is convex):

$$\Psi_i^2 = \big(\mathbb{E}[\psi(Z_T^i) \mid \mathbf{Y}_{t_0}]\big)^2 \leq \mathbb{E}[\psi(Z_T^i)^2 \mid \mathbf{Y}_{t_0}]. \tag{53}$$

Taking expectations over $\mathbf{Y}_{t_0} \sim q_i$ and applying the tower property:

$$\mathbb{E}_{q_i}[\Psi_i^2] \leq \mathbb{E}_{\mathbb{P}_i^\star}[\psi(Z_T^i)^2]. \tag{54}$$

Observe that, for any $z \in \mathbb{R}$, we have the bound

$$0 \leq \psi(z) = e^z - 1 - z \leq \frac{z^2}{2}e^{z_+}, \quad z_+ := \max(z, 0), \tag{55}$$

Using the bound (55):

$$\psi(Z_T^i)^2 \leq \frac{(Z_T^i)^4}{4}e^{2(Z_T^i)+} = \frac{(Z_T^i)^4}{4}\Big(\mathbf{1}_{Z_T^i \leq 0} + e^{2Z_T^i}\mathbf{1}_{Z_T^i \geq 0}\Big) \tag{56}$$

*5. Controlling the exponential factor (with only $2p' \leq 2 + \gamma$).* From (55), it remains to control the "positive" contribution

$$\mathbb{E}_{\mathbb{P}_i^\star}\Big[(Z_T^i)^4 e^{2Z_T^i}\mathbf{1}_{\{Z_T^i \geq 0\}}\Big].$$

Define the Hölder conjugates $p := \frac{1+\delta}{2}$ and $p' := \frac{\delta+1}{\delta-1}$ that are so that $\frac{1}{p} + \frac{1}{p'} = 1$. Hölder's inequality yields

$$\mathbb{E}_{\mathbb{P}_i^\star}\Big[(Z_T^i)^4 e^{2Z_T^i}\mathbf{1}_{\{Z_T^i \geq 0\}}\Big] \leq \mathbb{E}_{\mathbb{P}_i^\star}\Big[|Z_T^i|^{4p'}\mathbf{1}_{\{Z_T^i \geq 0\}}\Big]^{1/p'} \cdot \mathbb{E}_{\mathbb{P}_i^\star}\Big[e^{2pZ_T^i}\Big]^{1/p}. \tag{57}$$

Since $2p = 1 + \delta$, Assumption A3 gives

$$\sup_{i \geq i_0} \mathbb{E}_{\mathbb{P}_i^\star}\big[e^{2pZ_T^i}\big] = \sup_{i \geq i_0} \mathbb{E}_{\mathbb{P}_i^\star}\big[e^{(1+\delta)Z_T^i}\big] \leq C_\delta. \tag{58}$$

On the other hand, recall $Z_T^i = M_T^i - \frac{1}{2}\langle M^i\rangle_T$ so that, on the event $\{Z_T^i \geq 0\}$, we have $M_T^i \geq \frac{1}{2}\langle M^i\rangle_T$ i.e. $\langle M^i\rangle_T \leq 2M_T^i$ hence also

$$0 \leq Z_T^i = M_T^i - \frac{1}{2}\langle M^i\rangle_T \leq M_T^i, \qquad \text{on } \{Z_T^i \geq 0\}.$$

Therefore,

$$|Z_T^i|^{4p'}\mathbf{1}_{\{Z_T^i \geq 0\}} \leq |M_T^i|^{4p'}. \tag{59}$$

By the Burkholder–Davis–Gundy inequality (37) with $r = 4p'$,

$$\mathbb{E}_{\mathbb{P}_i^\star}\big[|M_T^i|^{4p'}\big] \leq C_{\text{BDG}}(4p') \mathbb{E}_{\mathbb{P}_i^\star}\big[\langle M^i\rangle_T^{2p'}\big].$$

Since, by definition of $\rho$ and $\gamma$ in Assumption A4, one has that $2p' \leq 2 + \gamma$, Lyapunov monotonicity yields

$$\mathbb{E}_{\mathbb{P}_i^\star}\big[\langle M^i\rangle_T^{2p'}\big] \leq \mathbb{E}_{\mathbb{P}_i^\star}\big[\langle M^i\rangle_T^{2+\gamma}\big]^{\frac{2p'}{2+\gamma}} \leq K_\gamma^{\frac{2p'}{2+\gamma}} \varepsilon_{\star,i}^{4p'}.$$

Combining with (59) gives

$$\mathbb{E}_{\mathbb{P}_i^\star}\Big[|Z_T^i|^{4p'}\mathbf{1}_{\{Z_T^i \geq 0\}}\Big] \leq C_{\text{BDG}}(4p') K_\gamma^{\frac{2p'}{2+\gamma}} \varepsilon_{\star,i}^{4p'}. \tag{60}$$

Plugging (58) and (60) into (57) yields

$$\mathbb{E}_{\mathbb{P}_i^\star}\Big[(Z_T^i)^4 e^{2Z_T^i}\mathbf{1}_{\{Z_T^i \geq 0\}}\Big] \leq \Big(C_{\mathrm{BDG}}(4p')\,K_\gamma^{\frac{2p'}{2+\gamma}}\Big)^{1/p'} C_\delta^{1/p}\,\varepsilon_{\star,i}^4.$$

Returning to (55) and using also $\mathbb{E}[(Z_T^i)^4]\mathbf{1}_{\{Z_T^i \leq 0\}} \leq \mathbb{E}[(Z_T^i)^4] \leq C_{Z4}\,\varepsilon_{\star,i}^4$ (Lemma C.2), we conclude that for some constant $C'$ depending only on $(\delta, C_\delta, \gamma, K_\gamma)$,

$$\mathbb{E}_{q_i}[\Psi_i^2] \leq \mathbb{E}_{\mathbb{P}_i^\star}[\psi(Z_T^i)^2] \leq C'\,\varepsilon_{\star,i}^4, \tag{61}$$

with an admissible explicit choice

$$C' := \frac{1}{4}\,C_{Z4} + \frac{1}{4}\Big(C_{\mathrm{BDG}}(4p')\,K_\gamma^{\frac{2}{2+\gamma}}\Big)^{1/p'} C_\delta^{1/p}.$$

*5. Combining all bounds.*

Substituting (50) and (61) into (48):

$$\chi^2(\hat{p}^{i+1}\|q_i) \geq (1-\nu)\left[(1-\theta)\eta_i\varepsilon_{\star,i}^2 - \frac{1}{4}\left(\frac{1}{\theta}-1\right)K_\gamma^{\frac{2}{2+\gamma}}\varepsilon_{\star,i}^4\right] - \left(\frac{1}{\nu}-1\right)C'\varepsilon_{\star,i}^4. \tag{62}$$

Expanding and collecting terms:

$$\chi^2(\hat{p}^{i+1}\|q_i) \geq (1-\nu)(1-\theta)\eta_i\varepsilon_{\star,i}^2 - \left[\frac{1}{4}(1-\nu)\left(\frac{1}{\theta}-1\right)K_\gamma^{\frac{2}{2+\gamma}} + \left(\frac{1}{\nu}-1\right)C'\right]\varepsilon_{\star,i}^4.$$

We choose $\nu = \theta = \frac{1}{2}$ to balance the terms:

- Leading coefficient: $(1-\frac{1}{2})(1-\frac{1}{2}) = \frac{1}{4}$

- First error coefficient: $\frac{1}{4}\cdot\frac{1}{2}\cdot(2-1)\cdot K_\gamma^{\frac{2}{2+\gamma}} = \frac{1}{8}K_\gamma^{\frac{2}{2+\gamma}}$

- Second error coefficient: $(2-1)\cdot C' = C'$

Thus:

$$\chi^2(\hat{p}^{i+1}\|q_i) \geq \frac{1}{4}\eta_i\varepsilon_{\star,i}^2 - \left[\frac{1}{8}K_\gamma^{\frac{2}{2+\gamma}} + C'\right]\varepsilon_{\star,i}^4. \tag{63}$$

Setting

$$C := \frac{1}{8}K_\gamma^{\frac{2}{2+\gamma}} + C', \tag{64}$$

we obtain the desired bound:

$$\chi^2(\hat{p}^{i+1}\|q_i) \geq \frac{1}{4}\cdot\eta_i\cdot\varepsilon_{\star,i}^2 - C\cdot\varepsilon_{\star,i}^4. \tag{65}$$

*6. Clean form in the perturbative regime.*

When the error is sufficiently small, specifically when

$$\varepsilon_{\star,i}^2 \leq \frac{c\cdot\eta_i}{2C} = \frac{\eta_i}{8C}, \tag{66}$$

the quartic error term satisfies $C\varepsilon_{\star,i}^4 \leq \frac{c}{2}\eta_i\varepsilon_{\star,i}^2$. Therefore:

$$\chi^2(\hat{p}^{i+1}\|q_i) \geq c\cdot\eta_i\cdot\varepsilon_{\star,i}^2 - \frac{c}{2}\cdot\eta_i\cdot\varepsilon_{\star,i}^2 = \frac{c}{2}\cdot\eta_i\cdot\varepsilon_{\star,i}^2. \tag{67}$$

This completes the proof. $\square$

*Remark* D.2 (Optimality of Constants). The constants $c = \frac{1}{4}$ and the specific form of $C$ arise from choosing $\nu = \theta = \frac{1}{2}$. These can be optimized by choosing $\nu$ and $\theta$ as functions of the ratio $\eta_i/K_\gamma$, but the qualitative scaling $\chi^2 \gtrsim \eta_i\varepsilon_{\star,i}^2$ is sharp.

# E. Proof of Theorem 3.5

We prove the following version of the result with explicit constants.

**Theorem E.1** (Two-Sided Control of Endogenous Error). *Under Assumptions A1- A4, the following holds for generations $i \geq i_0$. There exist constants $c > 0$ and $C < \infty$ (depending only on $\delta, C_\delta, \gamma, K_\gamma$) such that, whenever the perturbative condition $\varepsilon_{\star,i}^2 \leq 1$ holds, the sampling discrepancy is bounded by:*

$$\frac{1}{4}\,\eta_i\,\varepsilon_{\star,i}^2 \;-\; C_{\text{Low}} \cdot \varepsilon_{\star,i}^4 \;\;\leq\;\; \chi^2(\hat{p}^{\,i+1}\|q_i) \;\;\leq\;\; 4\,\varepsilon_{\star,i}^2 \;+\; [K_\gamma^{\frac{2}{2+\gamma}} + 2C']\varepsilon_{\star,i}^4. \tag{68}$$

*In particular, for sufficiently small $\varepsilon_{\star,i}^2 \to 0$, this yields*

$$\chi^2(\hat{p}^{\,i+1}\|q_i) \asymp \varepsilon_{\star,i}^2. \tag{69}$$

*Proof.* The proof relies on the decomposition established in Proposition D.1, which we first remind. Define the remainder function $\psi : \mathbb{R} \to \mathbb{R}$ by

$$\psi(z) := e^z - 1 - z = \sum_{k=2}^{\infty} \frac{z^k}{k!}. \tag{70}$$

Recall, from Proposition B.1 (Marginal Ratio Representation), we have

$$R_i(\mathbf{Y}_{t_0}) = \frac{\hat{p}^{i+1}(\mathbf{Y}_{t_0})}{q_i(\mathbf{Y}_{t_0})} = \mathbb{E}_{\mathbb{P}_i^\star}\left[e^{Z_T^i} \mid \mathbf{Y}_{t_0}\right]. \tag{71}$$

Applying the decomposition $e^{Z_T^i} = 1 + Z_T^i + \psi(Z_T^i)$ and taking conditional expectations:

$$R_i - 1 = \mathbb{E}_{\mathbb{P}_i^\star}[e^{Z_T^i} - 1 \mid \mathbf{Y}_{t_0}] = \mathbb{E}_{\mathbb{P}_i^\star}[Z_T^i \mid \mathbf{Y}_{t_0}] + \mathbb{E}_{\mathbb{P}_i^\star}[\psi(Z_T^i) \mid \mathbf{Y}_{t_0}].$$

We introduce the notation:

$$\bar{Z}_i := \mathbb{E}_{\mathbb{P}_i^\star}[Z_T^i \mid \mathbf{Y}_{t_0}] \quad \text{(conditional mean of log-likelihood ratio)}, \tag{72}$$

$$\Psi_i := \mathbb{E}_{\mathbb{P}_i^\star}[\psi(Z_T^i) \mid \mathbf{Y}_{t_0}] \quad \text{(conditional mean of remainder)}. \tag{73}$$

Thus we have the fundamental decomposition:

$$R_i - 1 = \bar{Z}_i + \Psi_i. \tag{74}$$

*Lower Bound.* This is exactly the statement of Proposition D.1 with $c = \frac{1}{4}$ (or $c = \frac{1}{8}$ in the clean regime).

*Upper Bound.* We use the elementary inequality $(a+b)^2 \leq 2a^2 + 2b^2$. Applied to the decomposition:

$$\chi^2(\hat{p}^{\,i+1}\|q_i) = \mathbb{E}_{q_i}[(\bar{Z}_i + \Psi_i)^2] \;\leq\; 2\mathbb{E}_{q_i}[\bar{Z}_i^2] \;+\; 2\mathbb{E}_{q_i}[\Psi_i^2]. \tag{75}$$

*1. The Signal Term.* By Jensen's inequality for conditional expectations, $\bar{Z}_i^2 = (\mathbb{E}[Z_T^i \mid \mathbf{Y}_{t_0}])^2 \leq \mathbb{E}[(Z_T^i)^2 \mid \mathbf{Y}_{t_0}]$. Taking expectation over $q_i$ and using the second moment bound of Lemma C.2:

$$\mathbb{E}_{q_i}[\bar{Z}_i^2] \leq \mathbb{E}_{\mathbb{P}_i^\star}[(Z_T^i)^2] \leq 2\varepsilon_{\star,i}^2 + \frac{1}{2}K_\gamma^{\frac{2}{2+\gamma}}\varepsilon_{\star,i}^4. \tag{76}$$

*2. The Remainder Term.* From the proof of Proposition D.1 (Eq. 61), we already established that $\mathbb{E}_{q_i}[\Psi_i^2] \leq C'\varepsilon_{\star,i}^4$ holds under Assumptions A3 and A4.

Substituting (33) and the remainder bound into (75):

$$\chi^2(\hat{p}^{\,i+1}\|q_i) \leq 2\left(2\varepsilon_{\star,i}^2 + \frac{1}{2}K_\gamma^{\frac{2}{2+\gamma}}\varepsilon_{\star,i}^4\right) + 2C'\varepsilon_{\star,i}^4 = 4\varepsilon_{\star,i}^2 + [K_\gamma^{\frac{2}{2+\gamma}} + 2C']\varepsilon_{\star,i}^4.$$

$\square$

# F. Proofs for Section 4

## F.1. Preliminary Results

We first quantify how much the discrepancy contracts purely due to mixing with fresh data (the "refresh" step).

**Lemma F.1** (Exact Contraction). *For any $\alpha \in (0, 1]$, the mixture $q_i$ satisfies:*

$$\chi^2(q_i \parallel p_{\text{data}}) = (1 - \alpha)^2 \chi^2(\hat{p}^i \parallel p_{\text{data}}) \tag{77}$$

*Proof.* Pointwise, the density ratio satisfies:

$$\frac{q_i}{p_{\text{data}}} - 1 = \alpha + (1 - \alpha)\frac{\hat{p}^i}{p_{\text{data}}} - 1 = (1 - \alpha)\left(\frac{\hat{p}^i}{p_{\text{data}}} - 1\right).$$

Squaring both sides and integrating with respect to $p_{\text{data}}$ yields the result. □

Next, we show that the innovation error $I_i$ cannot be completely hidden by the contraction.

**Lemma F.2** (Lower Bound Recursion).

$$D_{i+1} + (1 - \alpha)^2 D_i \geq \frac{\alpha}{2} I_i. \tag{78}$$

*Proof.* We decompose the likelihood ratio of the next generation $\hat{p}^{i+1}$ against the target $p_{\text{data}}$:

$$\frac{\hat{p}^{i+1}}{p_{\text{data}}} - 1 = \underbrace{\left(\frac{\hat{p}^{i+1}}{q_i} - 1\right)\frac{q_i}{p_{\text{data}}}}_{=:A} + \underbrace{\left(\frac{q_i}{p_{\text{data}}} - 1\right)}_{=:B}.$$

Then $D_{i+1} = \mathbb{E}_{p_{\text{data}}}[(A + B)^2]$. Using the algebraic inequality $(A + B)^2 \geq \frac{1}{2}A^2 - B^2$:

$$D_{i+1} \geq \frac{1}{2}\mathbb{E}_{p_{\text{data}}}[A^2] - \mathbb{E}_{p_{\text{data}}}[B^2].$$

1. *Term B:* By Lemma F.1, $\mathbb{E}_{p_{\text{data}}}[B^2] = \chi^2(q_i \parallel p_{\text{data}}) = (1 - \alpha)^2 D_i$.

2. *Term A:* We have:

$$\mathbb{E}_{p_{\text{data}}}[A^2] = \int \left(\frac{\hat{p}^{i+1}}{q_i} - 1\right)^2 \frac{q_i^2}{p_{\text{data}}^2} p_{\text{data}} = \int \left(\frac{\hat{p}^{i+1}}{q_i} - 1\right)^2 \frac{q_i}{p_{\text{data}}} q_i.$$

Since $q_i = \alpha p_{\text{data}} + (1 - \alpha)\hat{p}^i \geq \alpha p_{\text{data}}$, we have $q_i/p_{\text{data}} \geq \alpha$. Thus:

$$\mathbb{E}_{p_{\text{data}}}[A^2] \geq \alpha \int \left(\frac{\hat{p}^{i+1}}{q_i} - 1\right)^2 q_i = \alpha I_i.$$

Substituting these back yields $D_{i+1} \geq \frac{\alpha}{2}I_i - (1 - \alpha)^2 D_i$. Rearranging gives the result. □

**Lemma F.3** (Reverse Hölder bound for $R_i$). *Assume A3 with $\delta > 1$. Recall $q_i := \alpha p_{\text{data}} + (1 - \alpha)\hat{p}^i$, and $R_i := \frac{\hat{p}^{i+1}}{q_i}$. Then for all $i \geq i_0$,*

$$\mathbb{E}_{q_i}[R_i^{1+\delta}] \leq C_\delta, \tag{79}$$

*and consequently for any $r \in [1, 1 + \delta]$,*

$$\mathbb{E}_{q_i}[R_i^r] \leq C_\delta^{\frac{r-1}{\delta}}. \tag{80}$$

*Proof.* By the marginal projection identity (15),

$$R_i(\mathbf{Y}_{t_0}) = \mathbb{E}_{\mathbb{P}_i^\star}\big[e^{Z_T^i} \mid \mathbf{Y}_{t_0}\big], \qquad \mathbf{Y}_{t_0} \sim q_i \text{ under } \mathbb{P}_i^\star.$$

Since $x \mapsto x^{1+\delta}$ is convex, conditional Jensen yields

$$R_i(\mathbf{Y}_{t_0})^{1+\delta} \leq \mathbb{E}_{\mathbb{P}_i^\star}\big[e^{(1+\delta)Z_T^i} \mid \mathbf{Y}_{t_0}\big].$$

Taking expectation under $\mathbb{P}_i^\star$ and using (18) gives

$$\mathbb{E}_{q_i}[R_i^{1+\delta}] = \mathbb{E}_{\mathbb{P}_i^\star}[R_i(\mathbf{Y}_{t_0})^{1+\delta}] \leq \mathbb{E}_{\mathbb{P}_i^\star}[e^{(1+\delta)Z_T^i}] \leq C_\delta.$$

This proves (79). To get (80), take $r \in [1, 1+\delta]$ and fix $\theta = \frac{r-1}{\delta}$. Then, by interpolation with $L^1$,

$$\mathbb{E}_{q_i}[R_i^r] \leq (\mathbb{E}_{q_i}[R_i])^{1-\theta}(\mathbb{E}_{q_i}[R_i^{1+\delta}])^\theta = C_\delta^\theta, \qquad \theta = \frac{r-1}{\delta},$$

since, by definition of $R_i$, $\mathbb{E}_{q_i}[R_i] = 1$. This is exactly (80). $\qquad \square$

### F.2. Proof of Proposition 4.1

We restate Proposition 4.1 with explicit constants and then prove it.

**Proposition F.4.** *Assume A1, A2, A3 and A4. Let $D_i := \chi^2(\hat{p}^i \| p_{\text{data}})$. Assume that there exist constants $\underline{\eta} > 0$ and $i_0 \geq 0$ such that*

$$\eta_i \geq \underline{\eta} \quad and \quad \varepsilon_{\star,i}^2 \leq \min\left\{1, \frac{\underline{\eta}}{8C_{\text{Low}}}\right\} \qquad \forall i \geq i_0. \tag{81}$$

*where $C_{\text{Low}} \in (0, \infty)$ is defined in Theorem 3.5. Then the following statements hold.*

**(i) Divergent perturbative energy implies divergent accumulated error.** *For all $i \geq i_0$,*

$$\sum_{i \geq i_0} \varepsilon_{\star,i}^2 = +\infty \implies \sum_{i \geq 0} D_i = +\infty$$

*In particular, $(D_i)_{i \geq 0}$ cannot converge to $0$ and be summable.*

**(ii) Uniform perturbation floor implies recurring spikes.** *Assume there exist $i_0 \geq 0$, $\underline{\eta} > 0$, and $\underline{\varepsilon} > 0$ such that*

$$\eta_i \geq \underline{\eta}, \qquad \underline{\varepsilon} \leq \varepsilon_{\star,i}^2 \leq \min\left\{1, \frac{\underline{\eta}}{8C_{\text{Low}}}\right\} \qquad \forall i \geq i_0.$$

*Then,*

$$\limsup_{i \to \infty} D_i \geq \frac{\alpha}{2(1 + (1-\alpha)^2)} \cdot \frac{1}{8}\underline{\eta}\,\underline{\varepsilon}, \tag{82}$$

*and in particular, there exist infinitely many indices $i \geq i_0$ such that*

$$D_i \geq \frac{\alpha}{32(1 + (1-\alpha)^2)}\underline{\eta}\,\underline{\varepsilon}.$$

*Proof.* Lemma F.2 gives,

$$D_{i+1} + (1-\alpha)^2 D_i \geq \frac{\alpha}{2}I_i. \tag{83}$$

Under the conditions of (81), Theorem 3.5 yields for all $i \geq i_0$,

$$I_i := \chi^2(\hat{p}^{i+1} \| q_i) \geq \frac{1}{8}\underline{\eta}\,\varepsilon_{\star,i}^2. \tag{84}$$

Combining (83) and (84) gives, for all $i \geq i_0$,

$$D_{i+1} + (1-\alpha)^2 D_i \geq \frac{\alpha}{16}\underline{\eta}\,\varepsilon_{\star,i}^2, \tag{85}$$

**Proof of (i).** Summing (85) from $i = i_0$ to $n - 1$:

$$\sum_{i=i_0}^{n-1} D_{i+1} \; + \; (1-\alpha)^2 \sum_{i=i_0}^{n-1} D_i \; \geq \; \frac{\alpha}{16} \, \underline{\eta} \sum_{i=i_0}^{n-1} \varepsilon_{\star,i}^2.$$

Reindexing $\sum_{i=i_0}^{n-1} D_{i+1} = \sum_{i=i_0+1}^{n} D_i$, the left-hand side equals

$$\left(1 + (1-\alpha)^2\right) \sum_{i=i_0+1}^{n-1} D_i \; + \; D_n \; + \; (1-\alpha)^2 D_{i_0}.$$

Dropping $D_n \geq 0$ and using $\sum_{i=i_0}^{n-1} D_i \geq \sum_{i=i_0+1}^{n-1} D_i$ yields

$$\left(1 + (1-\alpha)^2\right) \sum_{i=i_0}^{n-1} D_i \; \geq \; \frac{\alpha}{16} \, \underline{\eta} \sum_{i=i_0}^{n-1} \varepsilon_{\star,i}^2 \; - \; (1-\alpha)^2 D_{i_0}.$$

Letting $n \to \infty$, if $\sum_{i \geq i_0} \varepsilon_{\star,i}^2 = +\infty$, then $\sum_{i \geq 0} D_i = +\infty$.

**Proof of (ii).** Assume in addition that $\varepsilon_{\star,i}^2 \geq \underline{\varepsilon}$ for all $i \geq i_0$. Then (85) implies

$$D_{i+1} + (1-\alpha)^2 D_i \; \geq \; \frac{\alpha}{16} \, \underline{\eta} \, \underline{\varepsilon}, \qquad \forall i \geq i_0.$$

Summing from $i = i_0$ to $n - 1$ and repeating the same algebra as above yields

$$\left(1 + (1-\alpha)^2\right) \sum_{i=i_0}^{n-1} D_i \; \geq \; \frac{\alpha}{16} \, \underline{\eta}(n - i_0)\underline{\varepsilon} \; - \; (1-\alpha)^2 D_{i_0}.$$

Dividing by $n - i_0$ gives the averaged lower bound

$$\frac{1}{n - i_0} \sum_{i=i_0}^{n-1} D_i \; \geq \; \frac{\alpha \underline{\eta}}{16(1 + (1-\alpha)^2)} \, \underline{\varepsilon} \; - \; \frac{(1-\alpha)^2 D_{i_0}}{(1 + (1-\alpha)^2)(n - i_0)}.$$

Taking the limit, we have

$$\liminf_{n \to \infty} \frac{1}{n - i_0} \sum_{i=i_0}^{n-1} D_i \; \geq \; \frac{\alpha \underline{\eta}}{16(1 + (1-\alpha)^2)} \, \underline{\varepsilon}.$$

Assume towards contradiction that $\limsup_{i \to \infty} D_i < \frac{\alpha \underline{\eta}}{16(1+(1-\alpha)^2)} \, \underline{\varepsilon}$. Then, by definition of the limsup there exist $\epsilon > 0$ and $N$ such that for all $i \geq N$, $D_i \leq \frac{\alpha \underline{\eta}}{16(1+(1-\alpha)^2)} \, \underline{\varepsilon} - \epsilon$. Averaging yields

$$\limsup_{n \to \infty} \frac{1}{n - i_0} \sum_{i=i_0}^{n-1} D_i \; \leq \; \frac{\alpha \underline{\eta}}{16(1 + (1-\alpha)^2)} \, \underline{\varepsilon} - \epsilon,$$

contradicting the lower bound. Hence

$$\limsup_{i \to \infty} D_i \; \geq \; \frac{\alpha \underline{\eta}}{16(1 + (1-\alpha)^2)} \, \underline{\varepsilon}.$$

$\square$

### F.3. Proof of Theorem 4.2

We first establish a result (Proposition F.5) that bounds $D_{i+1}$ in terms of $D_i$ and $I_i$. We then use the result to prove Theorem F.6, which shows that under the given assumptions and the summability condition $\sum_i \varepsilon_{\star,i}^2 < \infty$, there exists an explicit constant $D_{\max} < \infty$ such that $\sup_{i \geq 0} D_i \leq D_{\max}$. This theorem is then used to prove Theorem 4.2.

We begin by discussing Assumption A5 and its necessity.

**Compatibility on the tail set $\mathcal{G}_i^c$.** Proposition F.5 aims to establish a two-step recursive upper control on $D_i$, for which we need to bound the mixed tail term

$$\mathbb{E}_{q_i}\big[(R_i - 1)^2 \, T_i \, \mathbf{1}_{\mathcal{G}_i^c}\big],$$

where we recall $R_i(\mathbf{x}) := \frac{\hat{p}^{i+1}(\mathbf{x})}{q_i(\mathbf{x})}$, $T_i(\mathbf{x}) := \frac{q_i(\mathbf{x})}{p_{\text{data}}(\mathbf{x})} = \alpha + (1-\alpha)\frac{\hat{p}^i(\mathbf{x})}{p_{\text{data}}(\mathbf{x})}$, and the set $\mathcal{G}_i$ defined in (22). Crucially, this quantity couples two *a priori distinct* ratios: (i) $T_i$, which measures how far the training mixture $q_i$ deviates from $p_{\text{data}}$, and (ii) $R_i$, which measures the sampling/learning error incurred when moving from $q_i$ to $\hat{p}^{i+1}$. On $\mathcal{G}_i^c$ the density ratio $T_i$ can take arbitrarily large values even when $\chi^2(q_i\|p_{\text{data}})$ is finite; therefore, without additional structure, the product $(R_i-1)^2 T_i$ can concentrate precisely where $T_i$ is large, making $\mathbb{E}_{q_i}[(R_i-1)^2 T_i \mathbf{1}_{\mathcal{G}_i^c}]$ uncontrolled. In particular, it is possible that $T_i$ has heavy tails on $\mathcal{G}_i^c$ and $(R_i-1)^2$ is positively aligned with those tails, so that $\mathbb{E}_{q_i}[(R_i-1)^2 T_i \mathbf{1}_{\mathcal{G}_i^c}]$ is large despite small global errors measured by $I_i = \mathbb{E}_{q_i}[(R_i-1)^2]$. We therefore impose the following *compatibility* condition:

**(A5)** For $\delta > 1$ given in A3, and $p' := \frac{1+\delta}{\delta-1}$,

$$\sup_{i \geq i_0} \left\{ \zeta_i^{-1} \mathbb{E}_{q_i}[T_i^{p'} \mathbf{1}_{\mathcal{G}_i^c}] \right\} \leq C_\zeta,$$

which states that the contribution of regions where $q_i$ is much larger than $p_{\text{data}}$ is controlled in an $L^{p'}(q_i)$ sense. This assumption is mild in the regimes of interest: $\mathcal{G}_i$ is constructed adaptively so that $p_{\text{data}}(\mathcal{G}_i^c)$ is small, and the condition only restricts the *tail moment on that small set*, not the global behavior of $T_i$. Finally, some condition of this type is essentially unavoidable: any bound on a mixed product term requires either (a) pointwise control of $T_i$ on $\mathcal{G}_i^c$ (which is false by construction), or (b) an integrability/compatibility constraint preventing $(R_i-1)^2$ from concentrating where $T_i$ is extreme.

**Proposition F.5** (Deterministic two-step upper recursion). *Assume A1-A5, fix $\alpha \in (0,1]$ and recall*

$$D_i := \chi^2(\hat{p}^i\|p_{\text{data}}), \qquad q_i := \alpha p_{\text{data}} + (1-\alpha)\hat{p}^i, \qquad R_i := \frac{\hat{p}^{i+1}}{q_i}.$$

*Fix $\rho \in (0,1)$ and $\zeta_i \in (0,1)$, and define the bulk sets*

$$\mathcal{G}_i := \left\{ x : \left| \frac{\hat{p}^i(x)}{p_{\text{data}}(x)} - 1 \right| \leq \sqrt{\frac{D_i}{\zeta_i}} \right\}, \qquad \mathcal{A}_i(\rho) := \left\{ x : \left| R_i(x) - 1 \right| \leq \rho \right\}, \qquad \Omega_i := \mathcal{G}_i \cap \mathcal{A}_i(\rho).$$

*Let*

$$I_i := \chi^2(\hat{p}^{i+1}\|q_i) = \mathbb{E}_{q_i}\big[(R_i - 1)^2\big], \qquad B_i := \alpha + (1-\alpha)\left(1 + \sqrt{\frac{D_i}{\zeta_i}}\right).$$

*Then for all $i$,*

$$\sqrt{D_{i+1}} \leq (1-\alpha)\sqrt{D_i} + \sqrt{B_i[I_i + \rho^{-(\delta-1)}2^\delta(C_\delta+1)] + \rho^2(1 + (1-\alpha)^2 D_i) + K_\delta \zeta_i^{\frac{\delta-1}{1+\delta}}}, \tag{86}$$

*where $K_\delta = \left(2^\delta(C_\delta+1)\right)^{\frac{2}{1+\delta}} (C_\zeta)^{\frac{\delta-1}{1+\delta}}$ (the constants $\delta, C_\delta$ are from Assumption A3 and $C_\zeta$ is from A5).*

*Proof.* Recall,

$$T_i := \frac{q_i}{p_{\text{data}}}, \qquad A_i := \left(\frac{\hat{p}^{i+1}}{q_i} - 1\right)\frac{q_i}{p_{\text{data}}} = (R_i - 1)\,T_i.$$

Moreover, here, and in the rest of the proof, we use the notation $\|A_i\|_{L^2(p_{\text{data}})} := \left(\mathbb{E}_{p_{\text{data}}}[A_i^2]\right)^{1/2}$, and similarly for other random variables.

One has

$$\frac{\hat{p}^{i+1}}{p_{\text{data}}} - 1 = \left(\frac{\hat{p}^{i+1}}{q_i}\frac{q_i}{p_{\text{data}}}\right) - 1 = \left(\frac{\hat{p}^{i+1}}{q_i} - 1\right)\frac{q_i}{p_{\text{data}}} + \left(\frac{q_i}{p_{\text{data}}} - 1\right) = A_i + T_i - 1. \tag{87}$$

Hence

$$\sqrt{D_{i+1}} = \big\|A_i + T_i - 1\big\|_{L^2(p_{\text{data}})} \leq \|A_i\|_{L^2(p_{\text{data}})} + \|T_i - 1\|_{L^2(p_{\text{data}})} \tag{88}$$

by Minkowski's inequality.

*1. Exact contraction for the refresh term.* By Lemma F.1,

$$\|T_i - 1\|^2_{L^2(p_{\text{data}})} = \int \left(\frac{q_i}{p_{\text{data}}} - 1\right)^2 p_{\text{data}} = \chi^2(q_i \| p_{\text{data}}) = (1 - \alpha)^2 \chi^2(\hat{p}^i \| p_{\text{data}}). \tag{89}$$

Thus,

$$\sqrt{D_{i+1}} \le (1 - \alpha)\sqrt{D_i} + \|A_i\|_{L^2(p_{\text{data}})} \tag{90}$$

*2. Bounding $\|A_i\|^2_{L^2(p_{\text{data}})}$.* By definition,

$$
\begin{aligned}
\|A_i\|^2_{L^2(p_{\text{data}})} &= \int \left(\frac{\hat{p}^{i+1}}{q_i} - 1\right)^2 \frac{q_i}{p_{\text{data}}}^2 p_{\text{data}} = \int \left(\frac{\hat{p}^{i+1}}{q_i} - 1\right)^2 \frac{q_i}{p_{\text{data}}} q_i \\
&= \int_{\Omega_i} \left(\frac{\hat{p}^{i+1}}{q_i} - 1\right)^2 \frac{q_i}{p_{\text{data}}} q_i + \int_{\Omega_i^c} \left(\frac{\hat{p}^{i+1}}{q_i} - 1\right)^2 \frac{q_i}{p_{\text{data}}} q_i \\
&= J_i^{\text{bulk}} + J_i^{\text{tail}}.
\end{aligned}
\tag{91}
$$

We now upper bound $J_i^{\text{bulk}}$ and $J_i^{\text{tail}}$ separately.

*2.a. Bounding the Bulk Term.* On $\mathcal{G}_i$, we have the pointwise bound

$$\frac{\hat{p}^i}{p_{\text{data}}} \le 1 + \sqrt{\frac{D_i}{\zeta_i}} \quad \Longrightarrow \quad T_i = \alpha + (1 - \alpha)\frac{\hat{p}^i}{p_{\text{data}}} \le \alpha + (1 - \alpha)\left(1 + \sqrt{\frac{D_i}{\zeta_i}}\right) = B_i.$$

Therefore $\sup_{\Omega_i} T_i \le \sup_{\mathcal{G}_i} T_i \le B_i$ (since $\Omega_i \subseteq \mathcal{G}_i$) and

$$J_i^{\text{bulk}} \le \left(\sup_{\Omega_i} T_i\right) \int_{\Omega_i} \left(\frac{\hat{p}^{i+1}}{q_i} - 1\right)^2 q_i \le B_i \int \left(\frac{\hat{p}^{i+1}}{q_i} - 1\right)^2 q_i = B_i I_i. \tag{92}$$

*2.b Bounding The Tail term $J_i^{\text{tail}}$.* Since $\Omega_i = \mathcal{A}_i(\rho) \cap \mathcal{G}_i$, we have $\Omega_i^c \subseteq \mathcal{A}_i(\rho)^c \cup (\mathcal{G}_i^c \cap \mathcal{A}_i(\rho))$, hence

$$J_i^{\text{tail}} = \mathbb{E}_{q_i}[(R_i - 1)^2 T_i \mathbf{1}_{\Omega_i^c}] \le \underbrace{\mathbb{E}_{q_i}[(R_i - 1)^2 T_i \mathbf{1}_{\mathcal{G}_i^c \cap \mathcal{A}_i(\rho)}]}_{(*)} + \underbrace{\mathbb{E}_{q_i}[(R_i - 1)^2 T_i \mathbf{1}_{\mathcal{A}_i(\rho)^c}]}_{(**)}. \tag{93}$$

*Bounding (∗).* On $\mathcal{A}_i(\rho)$, $(R_i - 1)^2 \le \rho^2$ and also one has $q_i = T_i p_{\text{data}}$, hence,

$$\mathbb{E}_{q_i}[(R_i - 1)^2 T_i \mathbf{1}_{\mathcal{G}_i^c \cap \mathcal{A}_i(\rho)}] \le \rho^2 \mathbb{E}_{q_i}[T_i \mathbf{1}_{\mathcal{G}_i^c}] \le \rho^2 \mathbb{E}_{p_{\text{data}}}[T_i^2] = \rho^2(1 + \chi^2(q_i \| p_{\text{data}})) = \rho^2(1 + (1 - \alpha)^2 D_i) \tag{94}$$

Hence,

$$\mathbb{E}_{q_i}[(R_i - 1)^2 T_i \mathbf{1}_{\mathcal{G}_i^c \cap \mathcal{A}_i(\rho)}] \le \rho^2(1 + (1 - \alpha)^2 D_i) \tag{95}$$

*Bounding (∗∗).* Splitting $\mathcal{A}_i(\rho)^c$ into $\mathcal{A}_i(\rho)^c \cap \mathcal{G}_i$ and $\mathcal{A}_i(\rho)^c \cap \mathcal{G}_i^c$, we have

$$
\begin{aligned}
\mathbb{E}_{q_i}[(R_i - 1)^2 T_i \mathbf{1}_{\mathcal{A}_i(\rho)^c}] &= \mathbb{E}_{q_i}[(R_i - 1)^2 T_i \mathbf{1}_{\mathcal{A}_i(\rho)^c \cap \mathcal{G}_i}] + \mathbb{E}_{q_i}[(R_i - 1)^2 T_i \mathbf{1}_{\mathcal{A}_i(\rho)^c \cap \mathcal{G}_i^c}] \\
&\le \mathbb{E}_{q_i}[(R_i - 1)^2 T_i \mathbf{1}_{\mathcal{A}_i(\rho)^c \cap \mathcal{G}_i}] + \mathbb{E}_{q_i}[(R_i - 1)^2 T_i \mathbf{1}_{\mathcal{G}_i^c}].
\end{aligned}
\tag{96}
$$

For the first term in (96), notice that since $\mathcal{A}_i(\rho)^c \cap \mathcal{G}_i \subseteq \mathcal{G}_i$ and that on $\mathcal{G}_i$, we have that $\sup_{\mathcal{G}_i} T_i \le B_i$, one can write

$$\mathbb{E}_{q_i}[(R_i - 1)^2 T_i \mathbf{1}_{\mathcal{A}_i(\rho)^c \cap \mathcal{G}_i}] \le B_i \mathbb{E}_{q_i}[(R_i - 1)^2 \mathbf{1}_{\mathcal{A}_i(\rho)^c}]. \tag{97}$$

There thus remains to bound $\mathbb{E}_{q_i}[(R_i - 1)^2 \mathbf{1}_{\mathcal{A}_i(\rho)^c}]$.

On $\{|R_i - 1| > \rho\}$, we have $|R_i - 1|^2 \leq \rho^{-(\delta-1)}|R_i - 1|^{1+\delta}$ and also $|x - 1|^{1+\delta} \leq 2^\delta(x^{1+\delta} + 1)$ for $x \geq 0$. Therefore,

$$(R_i - 1)^2 \mathbf{1}_{\mathcal{A}_i(\rho)^c} \leq \rho^{-(\delta-1)}|R_i - 1|^{1+\delta} \leq \rho^{-(\delta-1)}2^\delta(R_i^{1+\delta} + 1). \tag{98}$$

Taking $\mathbb{E}_{q_i}$ and using Lemma F.3,

$$\mathbb{E}_{q_i}[(R_i - 1)^2 T_i \mathbf{1}_{\mathcal{A}_i(\rho)^c \cap \mathcal{G}_i}] \leq B_i \rho^{-(\delta-1)}2^\delta(C_\delta + 1) \tag{99}$$

For the second term of (96), we use assumption A5, Lemma F.3 with ($\delta > 1$) and Hölder with $p := \frac{1+\delta}{2}, p' := \frac{1+\delta}{\delta-1}$ (and we have $\frac{1}{p} + \frac{1}{p'} = 1$),

$$\mathbb{E}_{q_i}[(R_i - 1)^2 T_i \mathbf{1}_{\mathcal{G}_i^c}] \leq \left(\mathbb{E}_{q_i}[|R_i - 1|^{2p}]\right)^{1/p} \left(\mathbb{E}_{q_i}[T_i^{p'} \mathbf{1}_{\mathcal{G}_i^c}]\right)^{1/p'}$$
$$= \left(\mathbb{E}_{q_i}[|R_i - 1|^{1+\delta}]\right)^{2/(1+\delta)} \left(\mathbb{E}_{q_i}[T_i^{p'} \mathbf{1}_{\mathcal{G}_i^c}]\right)^{(\delta-1)/(1+\delta)}.$$

Using $|x - 1|^{1+\delta} \leq 2^\delta(x^{1+\delta} + 1)$ for $x \geq 0$, we get

$$\mathbb{E}_{q_i}[|R_i - 1|^{1+\delta}] \leq 2^\delta\left(\mathbb{E}_{q_i}[R_i^{1+\delta}] + 1\right) \leq 2^\delta(C_\delta + 1).$$

Moreover, by Assumption A5

$$\mathbb{E}_{q_i}[T_i^{p'} \mathbf{1}_{\mathcal{G}_i^c}] \leq C_\zeta \zeta_i.$$

Therefore,

$$\mathbb{E}_{q_i}[(R_i - 1)^2 T_i \mathbf{1}_{\mathcal{G}_i^c}] \leq \left(2^\delta(C_\delta + 1)\right)^{\frac{2}{1+\delta}} (C_\zeta)^{\frac{\delta-1}{1+\delta}} \zeta_i^{\frac{\delta-1}{1+\delta}} \tag{100}$$

Thus, from (99) and (100) the final bound on $(**)$ is:

$$\mathbb{E}_{q_i}[(R_i - 1)^2 T_i \mathbf{1}_{\mathcal{A}_i(\rho)^c}] \leq B_i \rho^{-(\delta-1)}2^\delta(C_\delta + 1) + \underbrace{\left(2^\delta(C_\delta + 1)\right)^{\frac{2}{1+\delta}} (C_\zeta)^{\frac{\delta-1}{1+\delta}}}_{:=K_\delta} \zeta_i^{\frac{\delta-1}{1+\delta}} \tag{101}$$

Plugging (95) and (101) in (93) yields,

$$J_i^{\text{tail}} \leq \rho^2(1 + (1-\alpha)^2 D_i) + B_i \rho^{-(\delta-1)}2^\delta(C_\delta + 1) + K_\delta \zeta_i^{\frac{\delta-1}{1+\delta}}. \tag{102}$$

Using the tail bound (102) and the bulk bound (92) in (91) yields:

$$\|A_i\|_{L^2(p_{\text{data}})} \leq \sqrt{B_i[I_i + \rho^{-(\delta-1)}2^\delta(C_\delta + 1)] + \rho^2(1 + (1-\alpha)^2 D_i) + K_\delta \zeta_i^{\frac{\delta-1}{1+\delta}}}. \tag{103}$$

Finally, combining (103) and (90) yields,

$$\sqrt{D_{i+1}} \leq (1-\alpha)\sqrt{D_i} + \sqrt{B_i[I_i + \rho^{-(\delta-1)}2^\delta(C_\delta + 1)] + \rho^2(1 + (1-\alpha)^2 D_i) + K_\delta \zeta_i^{\frac{\delta-1}{1+\delta}}} \tag{104}$$

$$\square$$

**Theorem F.6** (Perturbative stability under square-summability of score errors). *Assume A1-A5, and $\varepsilon_{\star,i}^2 \leq \min\left\{1, \frac{\eta_i}{8C}\right\}$ for all $i \geq i_0$ as well as the square-summability condition*

$$\sum_{i=0}^{\infty} \varepsilon_{\star,i}^2 < \infty. \tag{105}$$

*Fix any $\rho \in (0, 1 - \frac{3\alpha}{4})$ and any $\zeta \in (0, 1)$, and set $\zeta_i \equiv \zeta$ in Proposition F.5. Define*

$$b := \frac{1-\alpha}{\sqrt{\zeta}}, \quad C_{\text{loc}} := 2^\delta \rho^{-(\delta-1)}(C_\delta + 1), \quad K_\delta := \left(2^\delta(C_\delta + 1)\right)^{\frac{2}{1+\delta}} C_\zeta^{\frac{\delta-1}{1+\delta}},$$
$$C_{\text{tail}} := K_\delta \zeta^{\frac{\delta-1}{1+\delta}}, \quad C_0 := C_{\text{loc}} + C_{\text{tail}}, \quad K := b(C_{\text{loc}} + 4).$$

*Then there exists an explicit finite constant $D_{\max} < \infty$ such that*

$$\sup_{i \geq 0} D_i \leq D_{\max}.$$

*Moreover, with*

$$\mathcal{E}_2 := \Big( \sum_{i=0}^{\infty} \varepsilon_{\star,i}^2 \Big)^{1/2}, \qquad \beta := 1 - \frac{3\alpha}{4} + \rho \in (0,1),$$

*one may take*

$$D_{\max} = \left( \frac{\rho + \sqrt{C_0} + K/\alpha}{\frac{3\alpha}{4} - \rho} + \frac{\sqrt{C_\varepsilon}}{\sqrt{1 - \beta^2}} \, \mathcal{E}_2 \right)^2. \tag{106}$$

*Finally, the following asymptotic bound holds:*

$$\limsup_{i \to \infty} D_i \leq \Big( \frac{\rho + \sqrt{C_0} + K/\alpha}{\frac{3\alpha}{4} - \rho} \Big)^2. \tag{107}$$

*Proof.* Fix $\rho \in (0, 1 - \frac{3\alpha}{4})$ and $\zeta \in (0,1)$ and take $\zeta_i \equiv \zeta$ in Proposition F.5. Recall that

$$B_i := \alpha + (1 - \alpha)\Big( 1 + \sqrt{D_i/\zeta} \Big), \qquad I_i := \chi^2(\hat{p}^{i+1} \| q_i),$$

and set

$$C_{\mathrm{loc}} := 2^\delta \rho^{-(\delta-1)}(C_\delta + 1), \qquad C_{\mathrm{tail}} := K_\delta \, \zeta^{\frac{\delta-1}{1+\delta}}, \quad K_\delta = \Big( 2^\delta (C_\delta + 1) \Big)^{\frac{2}{1+\delta}} C_\zeta^{\frac{\delta-1}{1+\delta}}.$$

Then for all $i$,

$$\sqrt{D_{i+1}} \leq (1 - \alpha)\sqrt{D_i} + \sqrt{B_i\big(I_i + C_{\mathrm{loc}}\big) + \rho^2\big( 1 + (1-\alpha)^2 D_i \big) + C_{\mathrm{tail}}}. \tag{108}$$

*Step 1: Isolate the global term.* Using $\sqrt{u+v} \leq \sqrt{u} + \sqrt{v}$, we split the square-root in (108) as

$$\sqrt{D_{i+1}} \leq (1 - \alpha)\sqrt{D_i} + \sqrt{B_i(I_i + C_{\mathrm{loc}}) + C_{\mathrm{tail}}} + \rho\sqrt{1 + (1-\alpha)^2 D_i}.$$

Writing $x_i := \sqrt{D_i}$, the last term on the right can be bounded as

$$\rho\sqrt{1 + (1-\alpha)^2 D_i} = \rho\sqrt{1 + (1-\alpha)^2 x_i^2} \leq \rho\big( 1 + (1-\alpha)x_i \big) \leq \rho\big( 1 + x_i \big).$$

Therefore,

$$x_{i+1} \leq (1 - \alpha)x_i + \rho(1 + x_i) + \sqrt{B_i(I_i + C_{\mathrm{loc}}) + C_{\mathrm{tail}}}. \tag{109}$$

*Step 2: Use the perturbative bound on $I_i$.* Under $\varepsilon_{\star,i}^2 \leq 1$, by Theorem E.1 we have $I_i \leq 4\varepsilon_{\star,i}^2$. Hence

$$\sqrt{B_i(I_i + C_{\mathrm{loc}}) + C_{\mathrm{tail}}} \leq \sqrt{B_i\big( C_{\mathrm{loc}} + 4\varepsilon_{\star,i}^2 \big) + C_{\mathrm{tail}}}.$$

Since $\zeta$ is fixed,

$$B_i = 1 + \frac{1 - \alpha}{\sqrt{\zeta}} \, x_i = 1 + b\,x_i, \qquad b := \frac{1 - \alpha}{\sqrt{\zeta}}.$$

Thus, using $\varepsilon_{\star,i}^2 \leq 1$,

$$B_i\big( C_{\mathrm{loc}} + 4\varepsilon_{\star,i}^2 \big) + C_{\mathrm{tail}} = (C_{\mathrm{loc}} + 4\varepsilon_{\star,i}^2) + bx_i(C_{\mathrm{loc}} + 4\varepsilon_{\star,i}^2) + C_{\mathrm{tail}} \leq \underbrace{(C_{\mathrm{loc}} + C_{\mathrm{tail}})}_{=:C_0} + \underbrace{b(C_{\mathrm{loc}} + 4)}_{=:K} \, x_i + 4\varepsilon_{\star,i}^2.$$

Therefore, by $\sqrt{u + v + w} \leq \sqrt{u} + \sqrt{v} + \sqrt{w}$,

$$\sqrt{B_i\big( C_{\mathrm{loc}} + 4\varepsilon_{\star,i}^2 \big) + C_{\mathrm{tail}}} \leq \sqrt{C_0} + \sqrt{K} \, x_i^{1/2} + 2\,\varepsilon_{\star,i}.$$

Plugging into (109) gives

$$x_{i+1} \le (1 - \alpha + \rho)x_i + \rho + \sqrt{C_0} + \sqrt{K}\, x_i^{1/2} + 2\,\varepsilon_{\star,i}. \tag{110}$$

*Step 3: Absorb* $x_i^{1/2}$. Applying Young's inequality $\sqrt{K}\, x^{1/2} \le \frac{\alpha}{4}x + \frac{K}{\alpha}$ for any $x \ge 0$ to (110) yields

$$x_{i+1} \le \Big(1 - \alpha + \rho + \frac{\alpha}{4}\Big)x_i + \Big(\rho + \sqrt{C_0} + \frac{K}{\alpha}\Big) + 2\,\varepsilon_{\star,i}.$$

Define

$$\beta := 1 - \frac{3\alpha}{4} + \rho, \qquad A := \rho + \sqrt{C_0} + \frac{K}{\alpha},$$

Since $\rho < \frac{3\alpha}{4}$,

$$x_{i+1} \le \beta x_i + A + 2\,\varepsilon_{\star,i}. \tag{111}$$

Iterating (111) yields for $n \ge 1$,

$$x_n \le \beta^n x_0 + A \sum_{m=0}^{n-1} \beta^m + 2 \sum_{m=0}^{n-1} \beta^m \varepsilon_{\star,n-1-m}.$$

Since $x_0 = \sqrt{D_0} = 0$, this writes,

$$x_n \le \frac{A}{1 - \beta} + 2 \sum_{m=0}^{n-1} \beta^m \varepsilon_{\star,n-1-m}.$$

For the last term, Cauchy–Schwarz gives

$$\sum_{m=0}^{n-1} \beta^m \varepsilon_{\star,n-1-m} \le \Big( \sum_{m=0}^{\infty} \beta^{2m} \Big)^{1/2} \Big( \sum_{k=0}^{\infty} \varepsilon_{\star,k}^2 \Big)^{1/2} = \frac{1}{\sqrt{1 - \beta^2}}\, \mathcal{E}_2.$$

Since $1 - \beta = \frac{3\alpha}{4} - \rho$, we have

$$\frac{1}{1 - \beta} = \frac{1}{\frac{3\alpha}{4} - \rho} = \frac{4}{3\alpha - 4\rho}.$$

Hence

$$\sup_{n \ge 0} x_n \le x_0 + \frac{A}{1 - \beta} + \frac{2}{\sqrt{1 - \beta^2}}\, \mathcal{E}_2 = x_0 + \frac{4}{3\alpha - 4\rho}\, A + \frac{2}{\sqrt{1 - \beta^2}}\, \mathcal{E}_2.$$

Squaring gives (106) and therefore $\sup_i D_i \le D_{\max} < \infty$.

*The* $\limsup$ *bound* (107). It remains to show that the forcing convolution vanishes:

$$s_n := \sum_{m=0}^{n-1} \beta^m \varepsilon_{\star,n-1-m} \xrightarrow[n \to \infty]{} 0.$$

Since (105) implies $\varepsilon_{\star,i} \to 0$, fix $\epsilon > 0$ and choose $M$ so large that $\sum_{m \ge M} \beta^{2m} \le \epsilon^2$. Then split

$$s_n \le \sum_{m=0}^{M-1} \beta^m \varepsilon_{\star,n-1-m} + \sum_{m \ge M} \beta^m \varepsilon_{\star,n-1-m}.$$

For the first term, $\sum_{m=0}^{M-1} \beta^m \le (1 - \beta)^{-1}$ and $\max_{0 \le m \le M-1} \varepsilon_{\star,n-1-m} \to 0$, so it vanishes as $n \to \infty$. For the second term, Cauchy–Schwarz yields

$$\sum_{m \ge M} \beta^m \varepsilon_{\star,n-1-m} \le \Big( \sum_{m \ge M} \beta^{2m} \Big)^{1/2} \Big( \sum_{k \ge 0} \varepsilon_{\star,k}^2 \Big)^{1/2} \le \epsilon \mathcal{E}_2.$$

Letting $\epsilon \downarrow 0$ gives $s_n \to 0$. Taking $\limsup$ in the iterated form then yields

$$\limsup_{n \to \infty} x_n \le \frac{A}{1 - \beta} = \frac{\rho + \sqrt{C_0} + \frac{K}{\alpha}}{\frac{3\alpha}{4} - \rho},$$

which is (107).

$\square$

### F.3.1. PROOF OF THEOREM 4.2

We prove the following version of Theorem 4.2 with explicit constants.

**Theorem F.7.** *Assume A1-A5 hold for all $i \geq i_0$. Also assume observable errors and the perturbative regime:*

$$\varepsilon_{\star,i}^2 \leq \min\left\{1, \frac{\eta_i}{8C}\right\}, \quad \eta_i > \underline{\eta} > 0, \qquad \text{for all } i \geq i_0,$$

*and the square-summability condition*

$$\sum_{i=0}^{\infty} \varepsilon_{\star,i}^2 < \infty.$$

*Then Theorem F.6 applies, and in particular there exists $D_{\max} < \infty$ such that*

$$\sup_{i \geq i_0} D_i \leq D_{\max}. \tag{112}$$

*Define the discounted perturbation energy*

$$S_N := \sum_{i=i_0}^{N-1} (1-\alpha)^{2(N-1-i)} \varepsilon_{\star,i}^2. \tag{113}$$

*Then there exist constants $0 < c \leq C < \infty$ and $C_{\mathrm{bias}} \in [0, \infty)$, depending only on*

$$(\alpha, \underline{\eta}, \delta, C_\delta, \gamma, K_\gamma, C_\zeta, B_{\max}, D_{\max}),$$

*such that for all $N \geq i_0$,*

$$D_N + C_{\mathrm{bias}} \geq \frac{\alpha\underline{\eta}}{16}\Big(S_N - (1-\alpha)^{2(N-i_0)} D_{i_0}\Big), \quad D_N \leq 3(1-\alpha)^{2(N-i_0)} D_{i_0} + 12\bar{C}_A S_N + C_{\mathrm{bias}}. \tag{114}$$

*where, the bias term can be taken explicitly as*

$$C_{\mathrm{bias}} = \max\left\{ D_{\max}\left(1 + \frac{2(1-\alpha)^2}{1 - (1-\alpha)^2}\right), \frac{3\bar{c}_A}{\alpha^2} \right\}, \tag{115}$$

*where*

$$\bar{c}_A = B_{\max} 2^\delta \rho^{-(\delta-1)}(C_\delta + 1) + \rho^2\big(1 + (1-\alpha)^2 D_{\max}\big) + K_\delta \zeta^{\frac{\delta-1}{1+\delta}},$$

*with*

$$B_{\max} = \alpha + (1-\alpha)\Big(1 + \sqrt{D_{\max}/\zeta}\Big), \qquad K_\delta = \big(2^\delta(C_\delta + 1)\big)^{\frac{2}{1+\delta}} (C_\zeta)^{\frac{\delta-1}{1+\delta}}.$$

*Proof.* We recall that

$$T_i := \frac{q_i}{p_{\mathrm{data}}} = \alpha + (1-\alpha)\frac{\hat{p}^i}{p_{\mathrm{data}}}, \qquad R_i := \frac{\hat{p}^{i+1}}{q_i}, \qquad A_i := (R_i - 1)T_i,$$

Then $\hat{p}^{i+1}/p_{\mathrm{data}} - 1 = A_i + T_i - 1$, hence by Minkowski,

$$\sqrt{D_{i+1}} = \|A_i + T_i - 1\|_{L^2(p_{\mathrm{data}})} \leq \|A_i\|_{L^2(p_{\mathrm{data}})} + \|T_i - 1\|_{L^2(p_{\mathrm{data}})}. \tag{116}$$

By Lemma F.1, $\|T_i - 1\|_{L^2(p_{\mathrm{data}})} = (1-\alpha)\sqrt{D_i}$.

*Step 1: upper and lower control of $\|A_i\|_{L^2(p_{\mathrm{data}})}^2$ by $I_i$ (up to constants).* First note the lower bound: since $T_i \geq \alpha$ pointwise,

$$\|A_i\|_{L^2(p_{\mathrm{data}})}^2 = \mathbb{E}_{q_i}[(R_i - 1)^2 T_i] \geq \alpha\, \mathbb{E}_{q_i}[(R_i - 1)^2] = \alpha I_i. \tag{117}$$

For the upper bound, fix any $\rho \in (0, 1 - \frac{3\alpha}{4})$ and $\zeta \in (0, 1)$ and take $\zeta_i \equiv \zeta$ in Proposition F.5. Its proof (see (103)) yields

$$\|A_i\|_{L^2(p_{\mathrm{data}})}^2 \leq B_i\Big(I_i + 2^\delta \rho^{-(\delta-1)}(C_\delta + 1)\Big) + \rho^2\big(1 + (1-\alpha)^2 D_i\big) + K_\delta \zeta^{\frac{\delta-1}{1+\delta}}, \tag{118}$$

where

$$B_i = \alpha + (1 - \alpha)\left(1 + \sqrt{D_i/\zeta}\right).$$

Under (112), we have the uniform bound

$$B_i \leq \alpha + (1 - \alpha)\left(1 + \sqrt{D_{\max}/\zeta}\right) =: B_{\max} < \infty, \qquad i \geq i_0,$$

and also $\rho^2(1 + (1-\alpha)^2 D_i) \leq \rho^2(1 + (1-\alpha)^2 D_{\max})$. Therefore, for all $i \geq i_0$,

$$\|A_i\|^2_{L^2(p_{\text{data}})} \leq \bar{C}_A I_i + \bar{c}_A, \tag{119}$$

with

$$\bar{C}_A := B_{\max}, \qquad \bar{c}_A := B_{\max} 2^\delta \rho^{-(\delta-1)}(C_\delta + 1) + \rho^2\left(1 + (1-\alpha)^2 D_{\max}\right) + K_\delta \zeta^{\frac{\delta-1}{1+\delta}}.$$

*Step 2: Replace $I_i$ by $\varepsilon^2_{\star,i}$.* By Theorem 3.5 in the perturbative regime $\varepsilon^2_{\star,i} \leq \min\left\{1, \frac{\eta_i}{8C}\right\}$,

$$\frac{1}{8}\underline{\eta}\,\varepsilon^2_{\star,i} \leq I_i \leq 4\,\varepsilon^2_{\star,i}, \qquad i \geq i_0.$$

Combining with (117)–(119) gives, for all $i \geq i_0$,

$$\frac{\alpha}{8}\underline{\eta}\,\varepsilon^2_{\star,i} \leq \|A_i\|^2_{L^2(p_{\text{data}})} \leq 4\bar{C}_A\,\varepsilon^2_{\star,i} + \bar{c}_A. \tag{120}$$

*Step 3: Discounted upper bound (with bias).* From (116), (120), and recalling $\|T_i - 1\|_{L^2(p_{\text{data}})} = (1-\alpha)\sqrt{D_i}$,

$$\sqrt{D_{i+1}} \leq (1-\alpha)\sqrt{D_i} + 2\sqrt{\bar{C}_A}\,\varepsilon_{\star,i} + \sqrt{\bar{c}_A}.$$

Iterating for $N \geq i_0$ yields

$$\sqrt{D_N} \leq (1-\alpha)^{N-i_0}\sqrt{D_{i_0}} + 2\sqrt{\bar{C}_A}\sum_{i=i_0}^{N-1}(1-\alpha)^{N-1-i}\varepsilon_{\star,i} + \sqrt{\bar{c}_A}\sum_{i=i_0}^{N-1}(1-\alpha)^{N-1-i}.$$

The geometric sum is bounded by $\alpha^{-1}$, and by Cauchy-Schwarz,

$$\sum_{i=i_0}^{N-1}(1-\alpha)^{N-1-i}\varepsilon_{\star,i} \leq S_N^{1/2},$$

where $S_N$ is defined in (113). Hence,

$$\sqrt{D_N} \leq (1-\alpha)^{N-i_0}\sqrt{D_{i_0}} + 2\sqrt{\bar{C}_A}\,S_N^{1/2} + \frac{\sqrt{\bar{c}_A}}{\alpha}.$$

Squaring and using $(u + v + w)^2 \leq 3(u^2 + v^2 + w^2)$ gives

$$D_N \leq 3(1-\alpha)^{2(N-i_0)}D_{i_0} + 12\bar{C}_A\,S_N + \frac{3\bar{c}_A}{\alpha^2}.$$

Now, using that $\frac{3\bar{c}_A}{\alpha^2} \leq \max\{D_{\max}\left(1 + \frac{2(1-\alpha)^2}{1-(1-\alpha)^2}\right), \frac{3\bar{c}_A}{\alpha^2}\}$, we obtain the upper bound in (114),

$$D_N \leq 3(1-\alpha)^{2(N-i_0)}D_{i_0} + 12\bar{C}_A\,S_N + \max\{D_{\max}\left(1 + \frac{2(1-\alpha)^2}{1-(1-\alpha)^2}\right), \frac{3\bar{c}_A}{\alpha^2}\}.$$

*Discounted Lower Bound.* We establish the lower bound via induction on $N$.

**Claim:** For all $N \geq i_0 + 1$, the following inequality holds:

$$D_N + (1-\alpha)^{2(N-i_0)} D_{i_0} + 2 \sum_{k=i_0+1}^{N-1} (1-\alpha)^{2(N-k)} D_k \geq \frac{\alpha\eta}{16} S_N. \tag{121}$$

*Base Case ($N = i_0 + 1$):* For $N = i_0 + 1$, the sum is empty and $S_{i_0+1} = \varepsilon_{\star,i_0}^2$. Lemma F.2 alongside Theorem 3.5 yields:

$$D_{i_0+1} + (1-\alpha)^2 D_{i_0} \geq \frac{\alpha}{2} I_{i_0} \geq \frac{\alpha\eta}{16} \varepsilon_{\star,i_0}^2 = \frac{\alpha\eta}{16} S_{i_0+1},$$

which proves the base case.

*Inductive Step:* Assume the claim holds for some $N \geq i_0 + 1$. Multiply the assumed inequality (121) by $(1-\alpha)^2$:

$$(1-\alpha)^2 D_N + (1-\alpha)^{2(N+1-i_0)} D_{i_0} + 2 \sum_{k=i_0+1}^{N-1} (1-\alpha)^{2(N+1-k)} D_k \geq \frac{\alpha\eta}{16}(1-\alpha)^2 S_N. \tag{122}$$

From Lemma F.2, the single-step recurrence at generation $N$ gives:

$$D_{N+1} + (1-\alpha)^2 D_N \geq \frac{\alpha\eta}{16} \varepsilon_{\star,N}^2. \tag{123}$$

Adding (122) and (123) together yields:

$$D_{N+1} + 2(1-\alpha)^2 D_N + (1-\alpha)^{2(N+1-i_0)} D_{i_0} + 2 \sum_{k=i_0+1}^{N-1} (1-\alpha)^{2(N+1-k)} D_k \geq \frac{\alpha\eta}{16}(\varepsilon_{\star,N}^2 + (1-\alpha)^2 S_N).$$

By definition, $\varepsilon_{\star,N}^2 + (1-\alpha)^2 S_N = S_{N+1}$. Moreover, the term $2(1-\alpha)^2 D_N$ perfectly absorbs into the summation, extending its upper limit to $N$:

$$D_{N+1} + (1-\alpha)^{2(N+1-i_0)} D_{i_0} + 2 \sum_{k=i_0+1}^{N} (1-\alpha)^{2(N+1-k)} D_k \geq \frac{\alpha\eta}{16} S_{N+1}.$$

This completes the induction.

*Isolating $D_N$ via Uniform Boundedness:* To isolate $D_N$, we must bound the accumulated history on the left side of (121). From (112), we know that $\sup_{i \geq i_0} D_i \leq D_{\max}$. Consequently, the summation is bounded by a geometric series:

$$2 \sum_{k=i_0+1}^{N-1} (1-\alpha)^{2(N-k)} D_k \leq 2D_{\max} \sum_{j=1}^{\infty} (1-\alpha)^{2j} = 2D_{\max} \frac{(1-\alpha)^2}{1-(1-\alpha)^2}.$$

We can rewrite the left side of (121) to match the target format by extracting $\frac{\alpha\eta}{16}(1-\alpha)^{2(N-i_0)} D_{i_0}$:

$$D_N + (1 - \frac{\alpha\eta}{16})(1-\alpha)^{2(N-i_0)} D_{i_0} + 2 \sum_{k=i_0+1}^{N-1} (1-\alpha)^{2(N-k)} D_k + \frac{\alpha\eta}{16}(1-\alpha)^{2(N-i_0)} D_{i_0} \geq \frac{\alpha\eta}{16} S_N.$$

Substituting the geometric upper bound and bounding $(1 - \frac{\alpha\eta}{16})(1-\alpha)^{2(N-i_0)} D_{i_0} \leq D_{\max}$ yields:

$$D_N + D_{\max}\left(1 + \frac{2(1-\alpha)^2}{1-(1-\alpha)^2}\right) \geq \frac{\alpha\eta}{16}\left(S_N - (1-\alpha)^{2(N-i_0)} D_{i_0}\right).$$

Taking $C_{\text{bias}} = \max\{D_{\max}\left(1 + \frac{2(1-\alpha)^2}{1-(1-\alpha)^2}\right), \frac{3\bar{c}_A}{\alpha^2}\}$ yields the left-hand inequality of (114):

$$D_N + C_{\text{bias}} \geq \frac{\alpha\eta}{16}\left(S_N - (1-\alpha)^{2(N-i_0)} D_{i_0}\right).$$

$\square$

# G. Experimental Validation

We validate our theoretical framework by implementing the recursive training described in Section 1 on three different datasets: a 10-dimensional Gaussian mixture model (GMM), where ground-truth quantities can be computed analytically or estimated with high precision, and for score networks trained on image datasets (Fashion-MNIST (Xiao et al., 2017) and CIFAR-10 ((Krizhevsky, 2009)). We verify both the intra-generational bounds (Propositions 3.1 and 3.3) and the global accumulation formula (Theorem 4.2) on these settings.

### G.1. First Experimental Setup: 10-dimensional Gaussian Mixture

**Ground-truth distribution.** We consider a mixture of five isotropic Gaussians in $\mathbb{R}^{10}$:

$$p_{\text{data}}(\mathbf{x}) = \frac{1}{5} \sum_{k=1}^{5} \mathcal{N}(\mathbf{x}; \boldsymbol{\mu}_k, \sigma^2 \mathbf{I}_{10}), \tag{124}$$

where the means $\{\boldsymbol{\mu}_k\}_{k=1}^{5}$ are placed at the origin $(0,0)$ and at $((-4,-4),(-4,4),(4,-4),(4,4))$ and $\sigma = 0.6$. This configuration creates a well-separated mixture with analytically tractable score functions.

**Exact Gaussian closure for GMM data.** Contrary to realistic data distributions (e.g. images), the Gaussian initialisation for $p_{\text{data}}$ in this experiment is motivated by the close-form expressions it yields:

- *Explicit score expression.* The true score of $p_{\text{data}}$ can be written explicitly as:

$$\nabla_{\mathbf{x}} \log p_{\text{data}}(\mathbf{x}) = \sum_{k=1}^{5} \underbrace{\left( \frac{\mathcal{N}(\mathbf{x}; \boldsymbol{\mu}_k, \sigma^2 \mathbf{I}_{10})}{\sum_{j=1}^{5} \mathcal{N}(\mathbf{x}; \boldsymbol{\mu}_j, \sigma^2 \mathbf{I}_{10})} \right)}_{\text{Posterior probability } \pi_k(\mathbf{x})} \left( -\frac{\mathbf{x} - \boldsymbol{\mu}_k}{\sigma^2} \right) \tag{125}$$

- *Marginals explicit expression.* Because the forward diffusion is linear with additive Gaussian noise, each mixture component remains Gaussian at every diffusion time, and all time marginals are therefore Gaussian mixtures with explicitly known parameters. Moreover, both the ideal and learned reverse-time SDEs preserve Gaussianity at the component level. Consequently, for every generation $i$, the generated distribution $\hat{p}^i$ and the training mixture $q_i = \alpha p_{\text{data}} + (1 - \alpha)\hat{p}^i$ are exactly Gaussian mixtures. This yields closed-form expressions for the score, drift mismatch, and quadratic variation, ensuring that all theoretical quantities appearing in the bounds are well-defined and can be evaluated with high numerical precision.

**Score estimation via kernel density.** For the Gaussian mixture, rather than training neural networks, we use soft kernel density estimators (KDE) (Silverman, 1986) with bandwidth $h = \sigma = 0.6$ to approximate the score function from samples. Given $N$ training samples $\{x_j\}_{j=1}^{N}$, the estimated score at diffusion time $t$ is:

$$\hat{\mathbf{s}}_t(\mathbf{x}) = \frac{\sum_{j=1}^{N} w_j(\mathbf{x}, t) \cdot (a_t \mathbf{x}_j - \mathbf{x})}{\sigma_t^2}, \quad \text{where} \quad w_j(\mathbf{x}, t) \propto \exp\left( -\frac{\|\mathbf{x} - a_t \mathbf{x}_j\|_2^2}{2\sigma_t^2} \right), \tag{126}$$

with $a_t = e^{-t/2}$, $\sigma_t^2 = a_t^2 h^2 + (1 - a_t^2)$ and the constant of proportionality chosen to ensure $\sum_{j=1}^{N} w_j(\mathbf{x}, t) = 1$. This provides a smooth, differentiable score estimate that captures the structure of the training distribution while introducing controlled approximation error.

**Recursive training protocol.** We implement the iterative retraining procedure described in Section 1:

1. **Generation 0:** Sample $N = 100{,}000$ points from $p_{\text{data}}$ and construct the initial score estimator $\hat{s}_0$.

2. **Generation $i \to i+1$:** Form the training mixture $q_i = \alpha\, p_{\text{data}} + (1 - \alpha)\, \hat{p}^i$ by combining $\lfloor \alpha N \rfloor$ fresh samples from $p_{\text{data}}$ with $(N - \lfloor \alpha N \rfloor)$ synthetic samples from $\hat{p}^i$.

3. **Score update:** Construct the new score estimator $\hat{\mathbf{s}}_{i+1}$ from the training mixture $q_i$.

4. **Sampling:** Generate $\hat{p}^{i+1}$ by running the reverse diffusion SDE with drift $\hat{\mathbf{s}}_{i+1}$ for 500 Euler-Maruyama steps from $T = 4$ to $t_0 = 0.02$.

We repeat this process for 20 generations across three fresh data fractions $\alpha \in \{0.1, 0.5, 0.9\}$, with 10 independent runs per configuration to estimate variance.

**Divergence estimation.** We estimate the $\chi^2$ divergence $\chi^2(\hat{p}^i \| p_{\text{data}})$ using a binary classifier trained to distinguish samples from $\hat{p}^i$ versus $p_{\text{data}}$. Specifically, we train a 3-layer MLP with SiLU activations for 30 epochs on balanced datasets, then estimate:

$$\chi^2(\hat{p}^i \| p_{\text{data}}) \approx \mathbb{E}_{\mathbf{x} \sim p_{\text{data}}} \left[ \left( \frac{\hat{p}^i(\mathbf{x})}{p_{\text{data}}(\mathbf{x})} - 1 \right)^2 \right] = \mathbb{E}_{\mathbf{x} \sim p_{\text{data}}} \left[ (e^{\hat{r}(\mathbf{x})} - 1)^2 \right], \tag{127}$$

where $\hat{r}(\mathbf{x})$ is the learned log-density ratio. A similar procedure is used to estimate $I_i = \chi^2(\hat{p}^{i+1} \| q_i)$ and $\text{KL}(\hat{p}^{i+1} \| q_i)$.

**Score error computation.** For the 10-dimensional Gaussian mixture experiment, the score error $\mathbf{e}_i(\mathbf{x}, t) = \hat{\mathbf{s}}_{\theta_i}(\mathbf{x}, t) - \mathbf{s}^\star_{q_i}(\mathbf{x}, t)$ is computed as follows. The learned score $\hat{\mathbf{s}}_{\theta_i}$ is obtained via a soft kernel density estimator (KDE) with bandwidth $h = 0.6$ fitted to the training samples from $q_i$. The target score $\mathbf{s}^\star_{q_i}$ of the mixture $q_i = \alpha\, p_{\text{data}} + (1 - \alpha)\, \hat{p}^i$ is computed analytically using the identity

$$\mathbf{s}^\star_{q_i}(\mathbf{x}, t) = \frac{\alpha\, p_{\text{data},t}(\mathbf{x})\, \mathbf{s}^\star_{\text{data}}(\mathbf{x}, t) + (1 - \alpha)\, \hat{p}^i_t(\mathbf{x})\, \hat{\mathbf{s}}^i(\mathbf{x}, t)}{\alpha\, p_{\text{data},t}(\mathbf{x}) + (1 - \alpha)\, \hat{p}^i_t(\mathbf{x})},$$

where $\mathbf{s}^\star_{\text{data}}$ is the exact score of the Gaussian mixture (available in closed form), and $\hat{\mathbf{s}}^i$ is the KDE-approximated score of $\hat{p}^i$ using 50,000 samples from the previous generation.

**Score error and score energy.** For each generation $i$, we track the score error $\mathbf{e}(\mathbf{x}, t)$ at time $t \in [t_0, T]$ and for sample $\mathbf{x}$ as the difference between the learned model and the true model at this generation given by,

$$\mathbf{s}^\star_{q_i}(\mathbf{x}, t) = \frac{\alpha\, p_{\text{data},t}(\mathbf{x})\, \mathbf{s}^\star_{\text{data}}(\mathbf{x}, t) + (1 - \alpha)\, \hat{p}^i_t(\mathbf{x})\, \hat{\mathbf{s}}^i(\mathbf{x}, t)}{\alpha\, p_{\text{data},t}(\mathbf{x}) + (1 - \alpha)\, \hat{p}^i_t(\mathbf{x})}.$$

In addition, we track the integrated score error along sampling trajectories:

$$\hat{\varepsilon}_i^2 = \mathbb{E} \left[ \int_0^T \|\hat{\mathbf{s}}_{i+1}(\hat{\mathbf{Y}}^i_t, t) - \mathbf{s}_{q_i}(\hat{\mathbf{Y}}^i_t, t)\|_2^2 \, dt \right], \tag{128}$$

where the expectation is over trajectories $\{\hat{\mathbf{Y}}^i_t\}$ sampled using the *learned* score $\hat{\mathbf{s}}_{i+1}$. For the lower bound, we also compute $\varepsilon^2_{\star,i}$ taking expectation along trajectories sampled using the *target* score $\mathbf{s}_{q_i}$, approximated via the mixture formula.

**Observability coefficient.** We estimate, for all generations $i \in \{0, ..., 19\}$, the observability of errors coefficient $\hat{\eta}_i = \text{Var}(\mathbb{E}[M_i \mid \mathbf{Y}^{\star,i}_{t_0}])/\text{Var}(M_i)$, using the following method:

1. Run $N = 50,000$ reverse trajectories using the target score $\mathbf{s}_{q_i}$. For each trajectory $j = 1, \ldots, N$, we simulate the ideal reverse process $(\mathbf{Y}^{\star,(j)}_s)_{s \in [t_0, T]}$ using the target score $\mathbf{s}^\star_{q_i}$ via the Euler–Maruyama discretization with step size $\Delta t = T/n_{\text{steps}}$:

$$\mathbf{Y}^{\star,(j)}_{s-\Delta t} = \mathbf{Y}^{\star,(j)}_s + \left[ -\tfrac{1}{2}\mathbf{Y}^{\star,(j)}_s - \mathbf{s}^\star_{q_i}(\mathbf{Y}^{\star,(j)}_s, s) \right]\Delta t + \sqrt{\Delta t}\, \mathbf{z}^{(j)}_s,$$

where $\mathbf{z}^{(j)}_s \sim \mathcal{N}(0, \mathbf{I}_d)$ are independent standard Gaussian increments. At each step, we compute the score error $\mathbf{e}_i(\mathbf{Y}^{\star,(j)}_s, s) = \hat{\mathbf{s}}_{\theta_i}(\mathbf{Y}^{\star,(j)}_s, s) - \mathbf{s}^\star_{q_i}(\mathbf{Y}^{\star,(j)}_s, s)$ and accumulate the discretized martingale:

$$M_i^{(j)} \approx - \sum_{k=1}^{n_{\text{steps}}} \mathbf{e}_i(\mathbf{Y}^{\star,(j)}_{t_k}, t_k) \cdot \mathbf{z}^{(j)}_{t_k} \sqrt{\Delta t},$$

which approximates the Itô integral $M_i = -\int_{t_0}^T \mathbf{e}_i(\mathbf{Y}_s, s) \cdot d\bar{\mathbf{B}}_s$. The terminal samples $\mathbf{Y}^{(j)}_{t_0}$ and martingale values $M_i^{(j)}$ are then used to estimate the observability coefficient as described above.

2. Train a regression MLP to predict $M_i$ from $\mathbf{Y}_{t_0}^{\star,i}$.

3. Compute $\hat{\eta}_i = \mathrm{Var}(\hat{f}(\mathbf{Y}_{t_0}^{\star,i}))/\mathrm{Var}(M_i)$, where $\hat{f}$ is the learned regressor.

This provides an empirical estimate of how much score error information is retained in the terminal distribution.

**Heatmap for Memory Structure.** To empirically validate the theoretical memory structure in Theorem 4.2, we measure how score errors at each generation propagate to affect the quality of the final distribution. More specifically, to isolate the contribution of generation $k$'s score error to $D_n$, we run two parallel experiments:

- *Baseline*: All generations use their learned scores $\hat{\mathbf{s}}_1, \ldots, \hat{\mathbf{s}}_n$, yielding divergence $D_n$ (computed as described above).

- *Ablated at $i$*: Generation $i$ uses the *true* score $\mathbf{s}_{q_i}$ (here chosen to be the well trained (on real data) generation 0 model) instead of $\hat{\mathbf{s}}_i$, while all other generations use their learned scores, yielding divergence $D_n^{(-i)}$.

The difference measures how much generation $i$'s imperfect score worsened the final distribution:

$$\mathrm{Contrib}[i, n] = D_n - D_n^{(-i)}. \tag{129}$$

The motivation behind this approach is to understand "How much better would $\hat{p}^n$ approximate $p_{\mathrm{data}}$ if generation $i$ had used a perfect score? A large contribution indicates that errors at generation $i$ significantly degrade the final output, even many generations later.

### G.2. Results

**Intra-generational bounds.** Figures 3 and 4 validate both Propositions 3.1 and 3.3. The upper bound $\mathrm{KL}(\hat{p}^{i+1}\|q_i) \leq \frac{1}{2}\hat{\varepsilon}_i^2$ holds consistently, with the gap reflecting higher-order terms neglected in the Girsanov analysis. The lower bound $\chi^2(\hat{p}^{i+1}\|q_i) \geq \frac{1}{8}\eta_i \varepsilon_{\star,i}^2$ is also satisfied, with the tightness depending on the observability structure.

**Memory structure.** Figure 5 visualizes the contribution matrix, showing how past score errors propagate to current divergence. The effective memory horizon scales inversely with $\alpha$: at $\alpha = 0.1$, score errors from 10+ generations ago still contribute significantly, while at $\alpha = 0.9$, only the most recent generation matters.

**Global error accumulation.** Figure 6 shows the evolution of $D_i = \chi^2(\hat{p}^i\|p_{\mathrm{data}})$ across generations. The empirical trajectories closely follow the theoretical prediction $D_n \approx \sum_{k=0}^{N-1}(1-\alpha)^{2(N-1-k)}\varepsilon_{\star,k}^2$, confirming the discounted-sum structure of error accumulation. Past generation 4, we can see our control holds, validating our framework.

**Observability.** Figure 7 shows that $\hat{\eta}_i$ stabilizes to a positive constant after initial transients, with higher $\alpha$ yielding larger observability. This confirms that score errors on distributions closer to $p_{\mathrm{data}}$ project more strongly onto distinguishable directions, tightening the lower bound. Figure 8 provides qualitative confirmation: at low $\alpha$, the learned distribution visibly disperses over generations, while high $\alpha$ maintains distributional fidelity throughout the recursive process.

### G.3. Second Experimental Setup: Fashion-MNIST and CIFAR-10

Having established the theoretical framework in the controlled GMM setting, we turn to higher-dimensional image datasets, Fashion-MNIST and CIFAR-10, to verify that the core structural predictions continue to hold. extend to high-dimensional image distributions with neural network score estimation.

**Focused validation strategy.** In the high-dimensional image setting ($d = 784$), precise estimation of divergences such as $\chi^2(\hat{p}^i\|p_{\mathrm{data}})$ becomes statistically challenging: density ratio estimation in high dimensions suffers from the curse of dimensionality, and classifier-based divergence estimates exhibit high variance. Rather than attempting to verify all theoretical quantities with potentially unreliable estimates, we focus on two robust signatures that remain interpretable at scale:

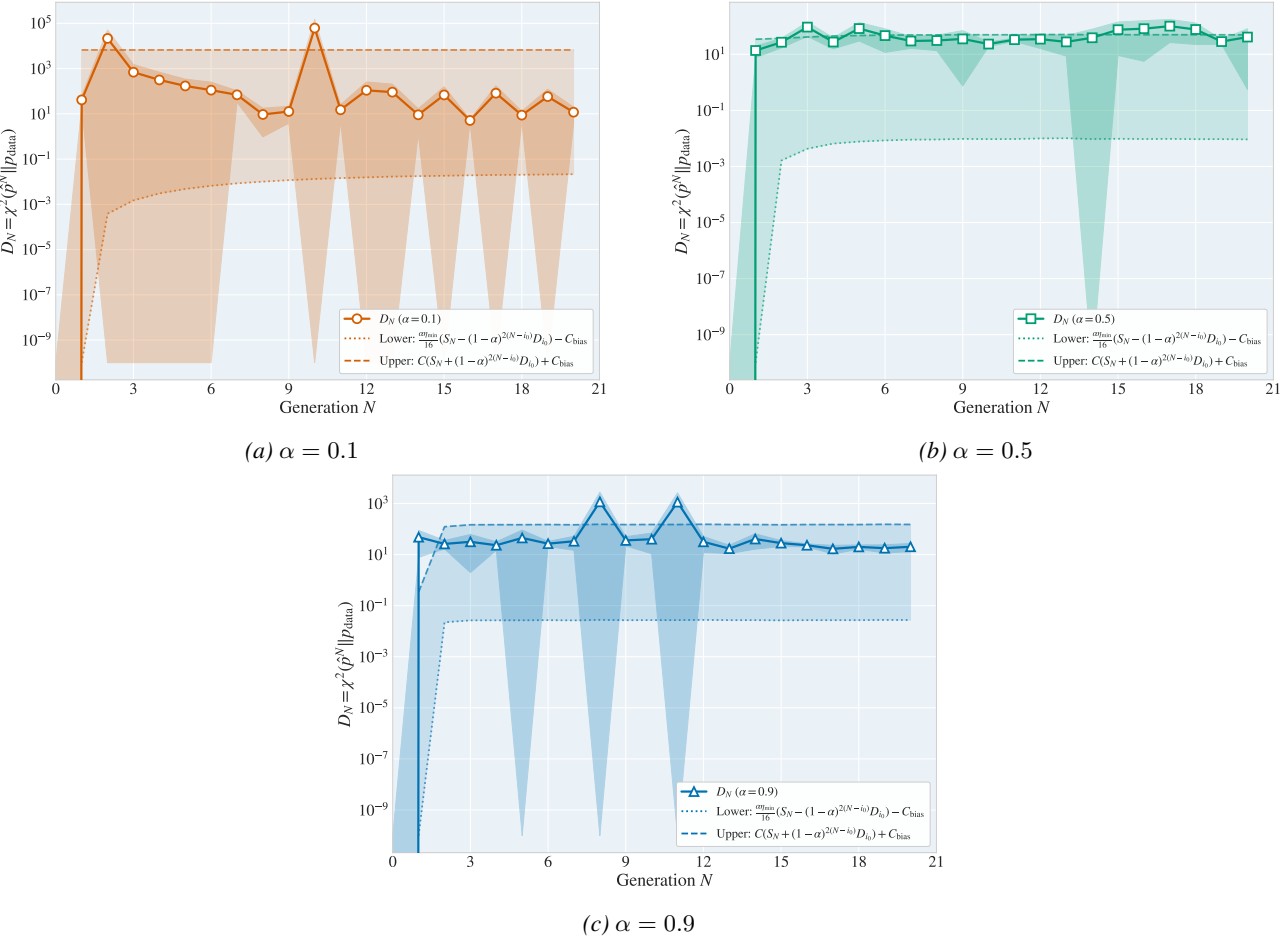

*(a)* $\alpha = 0.1$

*(b)* $\alpha = 0.5$

*(c)* $\alpha = 0.9$

*Figure 6.* **Accumulation of inter-generational divergence in the 10D GMM experiment.** Solid lines show the empirical divergence $D_N = \chi^2(\hat{p}^N \| p_{\text{data}})$ averaged over runs, with shaded regions indicating one standard deviation, for values of $\alpha \in \{0.1, 0.5, 0.9\}$. In each figure, the dotted curve corresponds to the *theoretical lower bound* $\frac{\alpha \eta_{\min}}{16(1+(1-\alpha)^2)} S_N$, derived in Theorem F.7, where $S_N = \sum_{k=0}^{N-1}(1-\alpha)^{2(N-1-k)}\varepsilon_{\star,k}^2$ is the predicted accumulated error energy. While the lower bound is easy to compute and corresponds to the theoretical lower bound derived in Theorem F.7, the upper bound, more difficult to compute directly (as it depends on theoretical constants) is obtained by an affine regression of $S_N + (1-\alpha)^{2N}D_0$ onto the observed divergence. As such, the upper bound is not a theoretical bound and is shown only to illustrate that the functional dependence predicted by the theory accurately captures the observed growth across generations.

1. **Memory contribution structure.** The heatmap visualization of the $(1-\alpha)^{2(N-k)}\varepsilon_{\star,k}^2$ contributions in (23) depends only on the *structure* of error propagation, not on absolute divergence magnitudes. The qualitative pattern— a narrow diagonal band for large $\alpha$, and a wide triangular region for small $\alpha$—is dimension-independent and directly tests the discounted-sum accumulation formula.

2. **Observability coefficient.** The ratio $\eta_i = \text{Var}(\mathbb{E}[M_i \mid \mathbf{Y}_{t_0}])/\text{Var}(M_i)$ is a normalized quantity in $[0, 1]$ that measures the *fraction* of score error information retained in terminal samples. Unlike raw divergences, $\eta_i$ does not scale with dimension and provides a stable diagnostic across settings.

The memory heatmaps test the *structural* prediction of Theorem 4.2: that current divergence decomposes as a geometrically-weighted sum of past innovations, with decay rate $(1-\alpha)^2$. If this structure holds empirically, then the theoretical framework captures the essential dynamics of recursive training, even if absolute divergence values are difficult to estimate precisely.

The observability coefficient tests whether our visibility of errors discussion (Section 3.2) remains valid in a fully data-driven experiments: that only gradient-aligned score errors affect the terminal distribution. A stable, non-zero $\eta_i$ confirms that our lower bound provides meaningful information in the neural network setting.

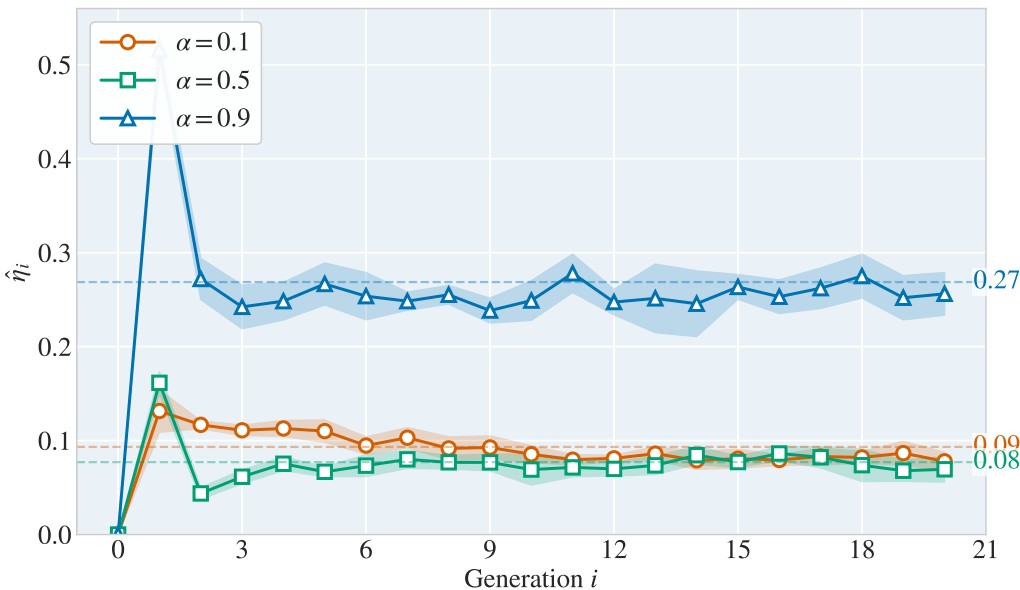

*Figure 7.* **Observability coefficient $\hat{\eta}_i$ across generations in the 10-dimensional Gaussian Mixture setting.** The observability coefficient measures the fraction of score error energy that leaves a detectable imprint on the terminal distribution: $\hat{\eta}_i = \mathrm{Var}(\widehat{\mathbb{E}}[M_i \mid \mathbf{Y}_{t_0}^{\star,i}])/\mathrm{Var}(M_i)$, where $M_i = -\int_0^T \mathbf{e}_i \cdot \mathrm{d}\bar{\mathbf{B}}_t$ is the error martingale. Higher $\alpha$ (more fresh data) yields larger $\hat{\eta}_i$, indicating that score errors are more strongly coupled to observable features when training distributions are closer to $p_{\mathrm{data}}$. At low $\alpha$, the training mixture $q_i$ drifts further from the data manifold, causing errors to accumulate in directions that are less distinguishable at the terminal time. Dashed horizontal lines indicate asymptotic means; shaded regions show $\pm 1$ standard deviation across runs. The non-zero values confirm that the lower bound $\chi^2(\hat{p}^{i+1} \| q_i) \geq \frac{1}{8} \eta_i \varepsilon_{\star,i}^2$ provides meaningful information about intra-generational error.

**Experimental setup.** We train diffusion models on $28 \times 28$ grayscale images from Fashion-MNIST ($N = 50{,}000$ training samples). The score network is a U-Net architecture with:

- Residual blocks with group normalization and SiLU activations

- Sinusoidal time embeddings projected through a 2-layer MLP

- Dropout regularization ($p = 0.1$) to prevent overfitting

- Exponential moving average (EMA) of weights with decay 0.99

**The $\varepsilon$-prediction objective.** In the standard DDPM formulation (Ho et al., 2020), the forward diffusion process adds Gaussian noise to a data sample $\mathbf{x}_0 \sim p_{\mathrm{data}}$ according to a variance schedule $\{\bar{\alpha}_t\}_{t=1}^T$. The noisy sample at time $t$ is given by

$$\mathbf{x}_t = \sqrt{\bar{\alpha}_t}\,\mathbf{x}_0 + \sqrt{1 - \bar{\alpha}_t}\,\mathbf{z}, \quad \text{where } \mathbf{z} \sim \mathcal{N}(0, \mathbf{I}_d).$$

Here, $\mathbf{z}$ is the *noise variable*—a standard Gaussian vector that represents the stochastic component added during the forward process. The neural network $\varepsilon_\theta(\mathbf{x}_t, t)$ is trained to predict this noise from the corrupted sample $\mathbf{x}_t$, yielding the denoising objective:

$$\mathcal{L}(\theta) = \mathbb{E}_{t \sim \mathrm{Unif}(t_0, T),\, \mathbf{x}_0 \sim p_{\mathrm{data}},\, \mathbf{z} \sim \mathcal{N}(0, \mathbf{I}_d)} \left[ \| \varepsilon_\theta(\mathbf{x}_t, t) - \mathbf{z} \|_2^2 \right].$$

This is equivalent to score matching, since the score of the noisy distribution satisfies $\nabla_{\mathbf{x}_t} \log p_t(\mathbf{x}_t) = -\mathbf{z}/\sqrt{1 - \bar{\alpha}_t}$, and thus $\varepsilon_\theta$ implicitly parameterizes the score function.

**Recursive training protocol.** We implement the iterative retraining procedure with fresh data fractions $\alpha \in \{0.1, 0.5, 0.9\}$:

1. **Generation 0:** Train the base model on real Fashion-MNIST data for 200 epochs with AdamW optimizer (learning rate $2 \times 10^{-4}$, weight decay $10^{-4}$) and cosine learning rate annealing.

| Gen 0 | Gen 10 | Gen 15 | Gen 25 |
|---|---|---|---|

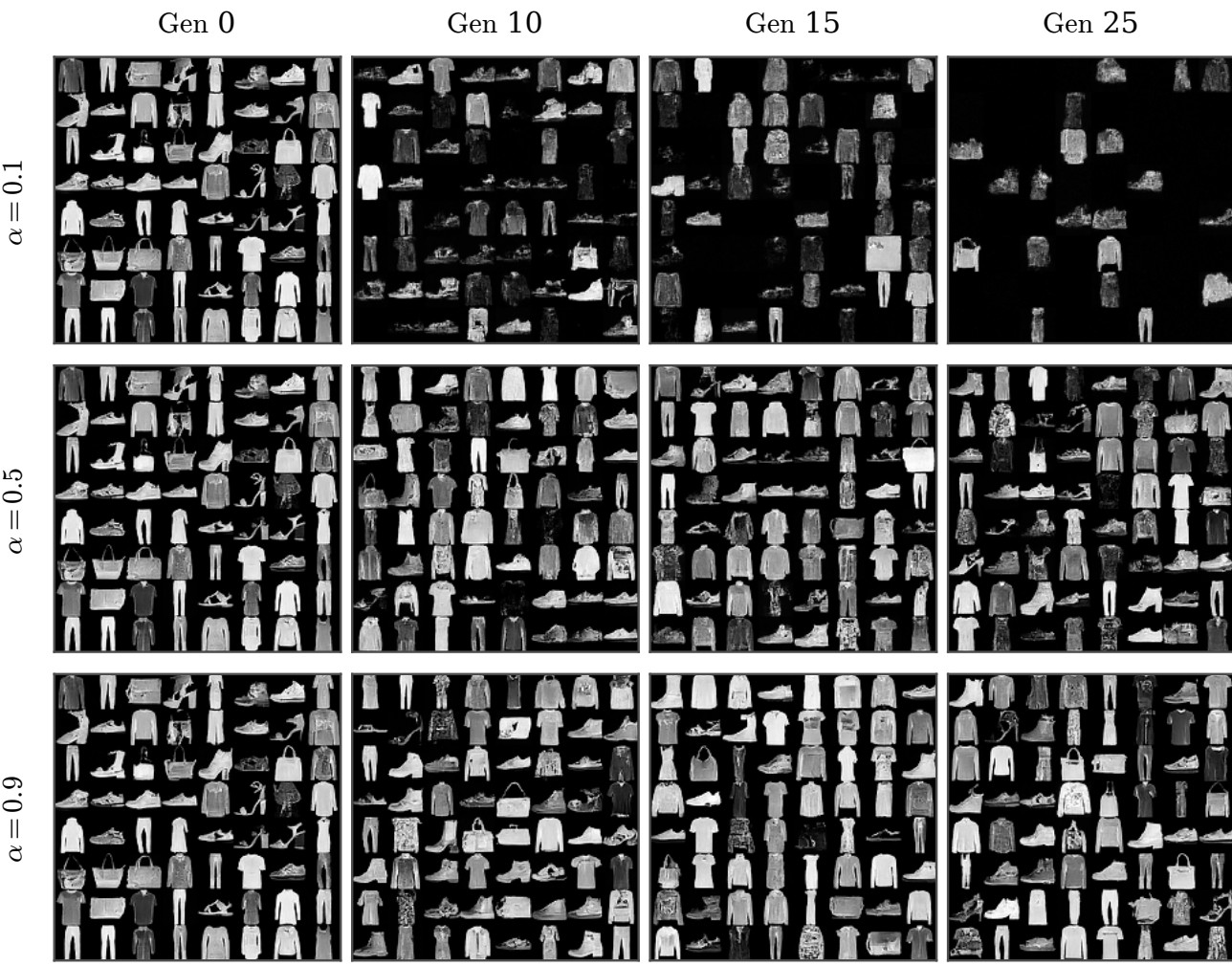

*Figure 8.* Random samples at generations 0 (left), 10, 15 and generation 25 (right) following the above recursive pipeline (right) for three fresh-data mixing rates $\alpha \in \{0.1, 0.5, 0.9\}$ (from top ($\alpha = 0.1$) to bottom ($\alpha = 0.9$)). Smaller values of $\alpha$ (more self-generated data) yield rapid degradation—loss of sharpness and diversity—while larger $\alpha$ stabilizes training and preserves sample quality over generations.

2. **Generation** $i \rightarrow i + 1$**:** Construct the training mixture $q_i = \alpha\, p_{\text{data}} + (1 - \alpha)\, \hat{p}^i$ by combining $\lfloor \alpha N \rfloor$ real images with $(1 - \lfloor \alpha N \rfloor)$ synthetic samples generated via DDPM sampling (Ho et al., 2020) from the current model.

3. **Fine-tuning:** Train the new model on $q_i$ for 100 epochs, with early stopping if validation loss plateaus for 50 epochs.

4. **Sampling:** Generate synthetic data using 1000-step DDPM sampling with the EMA model weights.

We run 10 generations per $\alpha$ value, tracking all theoretical quantities throughout the process.

**Metric computation.** Computing the quantities from our theoretical framework requires careful adaptation to the high-dimensional image setting.

- **Global divergence** $D_i = \chi^2(\hat{p}^i \| p_{\text{data}})$**:** In the CIFAR-10 (resp. Fashion-MNIST) experiments we evaluate distributional drift using a proxy $\chi^2$-divergence computed in a learned feature space rather than pixel space. Concretely, for each generation $i$ we draw $M$ synthetic samples $\mathbf{x}_{\text{gen}}^{(m)} \sim \hat{p}^i$ from the EMA-smoothed DDPM sampler and $M$ real samples $\mathbf{x}_{\text{real}}^{(m)} \sim p_{\text{data}}$ from the CIFAR-10 (resp. Fashion-MNIST) training set (we use $M = 1000$ in our implementation). We pass images through a fixed auxiliary classifier $c(\cdot)$ trained once on CIFAR-10 (resp. Fashion-MNIST), and extract penultimate-layer embeddings $f(\mathbf{x}) \in \mathbb{R}^d$. Let $\{f_{\text{gen}}^{(m)}\}_{m=1}^M$ and $\{f_{\text{real}}^{(m)}\}_{m=1}^M$ denote generated and real features. We then

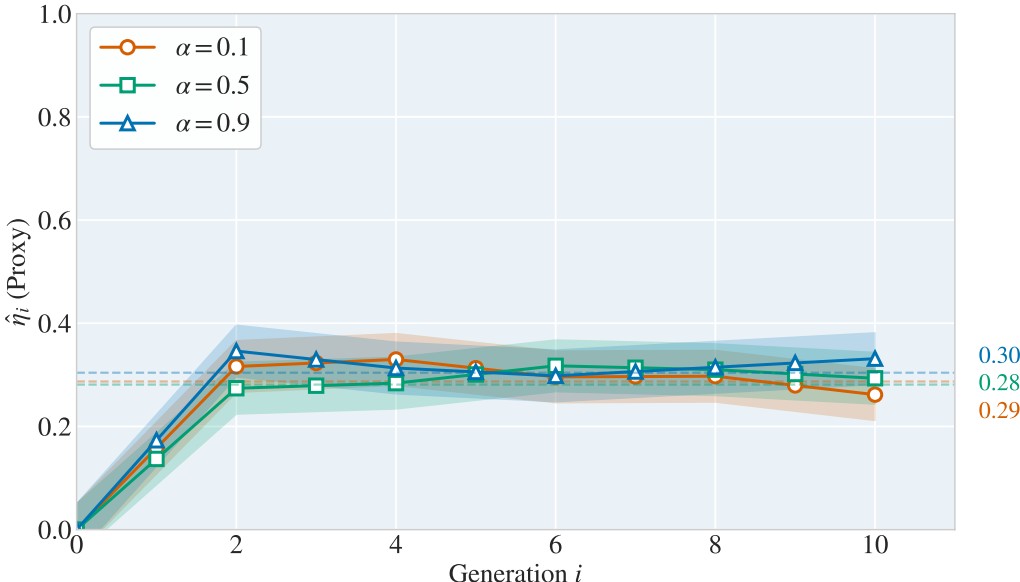

*Figure 9.* **Observability coefficient on CIFAR-10.** We estimate $\eta_i = \text{Var}(\widehat{\mathbb{E}}[M_i \mid \mathbf{Y}_{t_0}])/\text{Var}(M_i)$ by regressing the error martingale on terminal sample features (described in p.35). After an initial transient (generation 0 uses a placeholder value of 1), all three $\alpha$ values converge to similar observability levels ($\eta \approx 0.25$–$0.35$), indicating that approximately 30% of score error energy propagates to detectable distributional changes. Unlike the GMM setting where higher $\alpha$ yielded higher $\eta$, the CIFAR-10 curves overlap substantially, suggesting that observability in high-dimensional image spaces is governed more by the data manifold geometry than by the training mixture composition. The non-zero values confirm that the lower bound $\chi^2 \geq \frac{1}{8}\eta\varepsilon_\star^2$ provides meaningful information in the neural network setting.

fit Gaussian densities in feature space, $p_f = \mathcal{N}(\boldsymbol{\mu}_p, \Sigma_p)$ using $\{f_{\text{gen}}^{(m)}\}$ and $q_f = \mathcal{N}(\boldsymbol{\mu}_q, \Sigma_q)$ using $\{f_{\text{real}}^{(m)}\}$, with empirical mean and covariance (regularized as $\Sigma \leftarrow \Sigma + \epsilon I$ for numerical stability, taking a small $\epsilon$). The divergence proxy is the Monte Carlo estimate of $\chi^2(p_f \| q_f)$ under $q_f$:

$$\chi^2_{\text{feat}}(\hat{p}^i \| p_{\text{data}}) \ := \ \frac{1}{M} \sum_{m=1}^{M} \left( \frac{p_f(f_{\text{real}}^{(m)})}{q_f(f_{\text{real}}^{(m)})} - 1 \right)^2 \qquad f_{\text{real}}^{(m)} \sim q_f,$$

where the density ratio is evaluated via Gaussian log-likelihoods using a Cholesky factorization of $\Sigma_p$ and $\Sigma_q$. This metric captures how far the generated distribution has drifted from the real data distribution along discriminative directions encoded by the classifier features.

- **Score error energy $\hat{\varepsilon}_i^2$ and $\varepsilon_{\star,i}^2$:** We subsample 50 timesteps uniformly from $[0, T]$ and accumulate $\|\varepsilon_{\theta_{i+1}}(\mathbf{x}_t, t) - \varepsilon_{\theta_i}(\mathbf{x}_t, t)\|_2^2$ along sampling trajectories, where $\varepsilon_{\theta_i}(\mathbf{x}_t, t)$ denotes the learned $i$-th model according to the $\varepsilon$-prediction mechanism described above. For $\hat{\varepsilon}_i^2$, trajectories are sampled using the new model $\theta_{i+1}$; for $\varepsilon_{\star,i}^2$, trajectories use the previous model $\theta_i$.

- **Observability coefficient $\eta_i$:** Exactly as in the 10-dimensional Gaussian multivariate mixture model, we run paired reverse trajectories using the previous model's score, recording terminal samples $\mathbf{Y}_{t_0}$ and the discretized error martingale $M_i \approx -\sum_t \mathbf{e}_i(\mathbf{Y}_{t_0}, t) \cdot \Delta B_t$. A regression network is trained to predict $M_i$ from classifier features of $\mathbf{Y}_{t_0}$, yielding $\hat{\eta}_i = \text{Var}(\hat{f}(\mathbf{Y}_{t_0}))/\text{Var}(M_i)$.

**Heatmap.** Once again, the procedure here is exactly the same as the one described in the 10-dimensional Gaussian multivariate mixture model. Theorem 4.2 predicts that score errors propagate with geometric decay:

$$\text{Contrib}[i, n] \ \approx \ (1-\alpha)^{2(N-1-i)} \varepsilon_i^2, \tag{130}$$

where $\varepsilon_i^2 = \int_0^T \|\hat{\mathbf{s}}_k - \mathbf{s}_{q_k}\|_2^2 \, \mathrm{d}t$ is the path-space score error at generation $k$. The factor $(1-\alpha)^{2(N-1-i)}$ captures the mitigation induced by fresh data injection: when $\alpha$ is large, errors decay rapidly; when $\alpha$ is small, errors persist across many

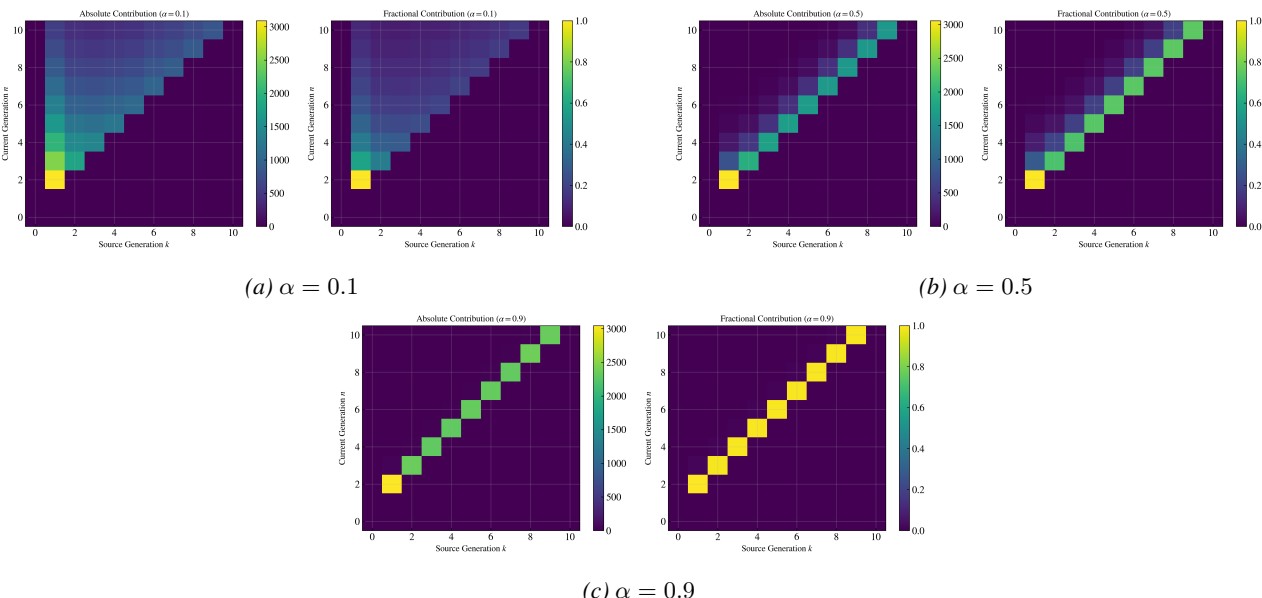

(a) $\alpha = 0.1$

(b) $\alpha = 0.5$

(c) $\alpha = 0.9$

*Figure 10.* **Memory structure of error accumulation on Fashion-MNIST.** Each panel shows absolute (left) and fractional (right) contributions of generation-$k$ errors to generation-$N$ divergence, computed as $(1 - \alpha)^{2(N-1-k)} \varepsilon_{\star,k}^2$. The structural predictions of Theorem 4.2 are clearly validated: **(a)** At $\alpha = 0.1$, errors persist across the full 10-generation horizon, with early generations (particularly generation 1) contributing substantially even at generation 10—the wide triangular structure indicates long memory and susceptibility to collapse. **(b)** At $\alpha = 0.5$, intermediate memory decay creates a band approximately 3–4 generations wide, balancing error persistence with forgetting. **(c)** At $\alpha = 0.9$, the sharp diagonal structure confirms that only the most recent generation contributes meaningfully—past errors are rapidly forgotten, preventing accumulation. These patterns match the GMM results (Figure 5) despite the $78\times$ increase in dimension, confirming that memory structure is a universal feature of recursive diffusion training independent of data complexity.

generations. As illustrated in Figure 10 the conclusions of this experiment match the theory, suggesting it also holds in higher dimensions.

