# OpenReview forum: "Quantifying Error Propagation and Model Collapse in Diffusion Models"
_ICML.cc/2026/Conference — ICML 2026 regular_

### Official Review · Reviewer_emzm · 2026-02-25

**Soundness:** 3
**Presentation:** 3
**Significance:** 2
**Originality:** 3
**Overall Recommendation:** 3
**Confidence:** 2

**Summary:**

The paper investigates recursive training of generative diffusion models on mixtures of real and synthetic data. For this setting, the authors claim novel bounds on the accumulated divergence between generated and target distributions. Based on their analysis, the authors claim that recursive training leads to two distinct regimes where the divergence provably cannot converge to zero or where it is stable and can be analytically decomposed. Empirically, the authors also study recursive training for synthetic datasets and some analysis is also present for image data.

**Compliance With Llm Reviewing Policy:**

Affirmed.

**Final Justification:**

My main concern was around practical implications of this work but I appreciate that the contribution is theoretical here, so this is not easy to resolve.

I do not specialize in theory and haven't gone through the derivations rigorously. Therefore, I am not comfortable with providing a strong recommendation and remain neutral.

**Key Questions For Authors:**

1. Starting in line 135, the authors comment that the stable regime for the divergence may require up to exponentially many samples in the dimension $d$. Given that diffusion models operate on high dimensional data such as images and videos, is there reason to believe that the stable regime analyzed here reflects such cases? I appreciate that the intrinsic dimension of such data is often smaller than $d$ (line 192), but it is not clear to me whether this property holds in recursive training.

**I acknowledge that I may not be within the paper’s target audience and have adjusted my review confidence accordingly.**

**Limitations:**

Given that this work is mainly theoretical, claims are qualified via technical assumptions. Still, some discussion on real world applicability (see Weaknesses and Questions above) would be appreciated. I encourage the authors to include an explicit Limitations section in the final version of the paper.

**Strengths And Weaknesses:**

Strengths
-

1. The problem of recursive training is important and relevant for modern generative modeling.
2. Though the paper is largely theoretical and mathematically dense, the central setting and contributions are clearly defined so that they may be appreciated by a broader audience.
3. The authors have included a detailed Appendix, with derivations and experimental setup.


Weaknesses
-

1. Despite the topic being important, the implications of the theory are unclear. The fact that more synthetic data leads to faster degradation is intuitively understood and is not surprising.

2. Quantiative analysis is focused on toy settings such as MoG that do not reflect real applications. Empirically, the derived bounds also appear to be loose, e.g., in Figure 3.

---

> ### Author Rebuttal · Authors · 2026-03-28
>
> We thank the reviewer for their time and constructive feedback.
>
> **Q1: Implications of theory are unclear**
>
> We agree that the qualitative statement "more synthetic data can worsen recursive retraining" is intuitive. Our contribution is to provide a quantitative mechanism that deepens this intuition by describing how degradation propagates once score errors are present. Specifically, our theory goes beyond the qualitative intuition by quantifying:
>
> 1. Which errors translate to marginal drift: Theorem 3.4 shows that the one-generation divergence is equivalent up to constants to the pathwise score-error energy. This is not obvious because pathwise errors need not be visible at the endpoint; our observability coefficient makes precise when such errors do or do not translate into marginal drift.
>
> 2. How fast old errors are forgotten: Theorem 4.2 shows that score errors from $m$ generations ago are suppressed by exactly $(1-\alpha)^{2m}$, giving a precise geometric forgetting rate as a function of the fresh-data fraction $\alpha$ (Fig. 5, 10).
>
> In short, our work identifies the score errors as the effective quantity propagated across generations, and quantifies the role of $\alpha$ as well as how these errors are damped over generations.
>
> **Q2: Toy setting does not reflect real applications and looseness of bounds**
>
> The mixture-of-Gaussians serves as a controlled environment where key theoretical quantities in our analysis -- pathwise score-error energies, observability coefficient, and accumulated divergences -- can be estimated reliably to allow a meaningful comparison with our bounds. We agree this setting is not fully representative of modern applications, which is why we also include image-data experiments in Appendix G.3. On Fashion-MNIST (Fig. 10), we observe the geometric memory structure of Theorem 4.2: the influence of generation-$k$ error on later generations decays at a rate consistent with $(1-\alpha)^{2(N-k)}$ across different values of $\alpha$. On CIFAR-10 (Fig. 9), the observability coefficient is strictly positive and stable under neural network score learning. We view these experiments not as a full numerical validation of every constant in the bounds, but as evidence that the structural predictions of the theory remain visible beyond the analytically tractable MoG case.
>
> The standard upper bound in Figure 3 is expected to be loose: the upper bound controls terminal-marginal divergence using path-space KL that accounts for all score errors made along the reverse diffusion, even those that do not affect endpoint marginals. The lower bound is intrinsically harder because it requires filtering only errors that affect marginals, as captured by our observability coefficient $\eta_i$. To the best of our knowledge, our work is the first lower bound on the error of diffusion models, and opens avenues for further improvements. We will clarify in the revision that the main empirical takeaway is the validation of the predicted functional dependence and regime structure, rather than tight numerical saturation of the bounds.
>
> **Key Question: Discussion on high-dimensional data**
>
> We agree that the small score error assumption is a limitation and that small intrinsic dimension may not hold under recursive training. Our contribution is to characterize the propagation dynamics conditional on being in the small score-error regime. Once per-generation score errors are sufficiently small, Theorem 4.2 characterizes exactly how they accumulate as a geometrically discounted sum governed by $\alpha$.
>
> Our Fashion-MNIST results (Fig. 10) provide indirect evidence of the validity of the small error assumption. If recursive training was driving the system outside this regime, one would expect the geometric memory structure to deviate from the theoretical prediction. Its persistence over 10 generations is at least indirect empirical evidence that the regime of Theorem 4.2 remains approximately valid in practice. We acknowledge that establishing this formally remains an open question and will add this to the Limitations section of the revised manuscript.

---

> > ### Author Rebuttal · Reviewer_emzm · 2026-04-03
> >
> > Thank you for your responses.
> >
> > My main concern was around practical implications of this work but I appreciate that the contribution is theoretical here, so this is not easy to resolve.
> >
> > I do not specialize in theory and haven't gone through the derivations rigorously. Therefore, I am not comfortable with providing a strong recommendation and remain neutral.

---

### Official Review · Reviewer_AU8H · 2026-03-10

**Soundness:** 4
**Presentation:** 4
**Significance:** 3
**Originality:** 3
**Overall Recommendation:** 5
**Confidence:** 4

**Summary:**

This theory-focused paper analyzes the "model collapse" phenomenon in score-based diffusion models trained recursively on data distributions containing a positive fraction of synthetic data from last round (generation) of training.

The authors rigorously quantify the drift of the learned distribution at the $i$th generation from the true data distribution using the $\chi^2$ metric. They derive an expression for the accumulated divergence of the learned model after $N$ generations, clearly showing how both the synthetic data error term and the proportion of true data $\alpha$ contribute to error accumulation across generations. The theoretical results are supported by clear and convincing numerical experiments.

**Compliance With Llm Reviewing Policy:**

Affirmed.

**Final Justification:**

My concerns are fully addressed.

**Key Questions For Authors:**

1. The requirement that the observability coefficients $\eta_i$ remain uniformly bounded away from zero across generations is necessary for Theorem 4.2 to be meaningful but seems somewhat strong in practice. Although extensive numerical evidence supports this assumption, providing a simple sufficient condition or an example of a basic model where this holds would be helpful for readers.

2. Assumptions A3 and A4 appear to be standard necessary conditions for the analysis. It would be beneficial to include additional discussion about the types of settings in which these assumptions are likely to hold.

3. Reference [1] (Cui, Pehlevan, Lu, NeurIPS 2025) also theoretically studies a simplified model of generative diffusion and the emergence of mode collapse. Including a mention of this work and clarifying its relation to the current paper would help readers better understand the state of knowledge on this topic.

I have also found the following typos:
- On page 7, Section 4: "[...] if $\sum_i \varepsilon_i^2 = \infty$, then we show that $D_i$ *and* cannot converge to zero[...]"
- On page 7, Section 4.2: "[...] $G_i^c$ consists of the regions where *where* [...]"


[1]: Cui, Hugo, Cengiz Pehlevan, and Yue M. Lu. "A solvable model of learning generative diffusion: theory and insights." The Thirty-ninth Annual Conference on Neural Information Processing Systems.

**Limitations:**

Yes.  The limit of smallness of score error is stated in the last section .

**Strengths And Weaknesses:**

### Strengths
The paper is very well written and presented, with clear and rigorous analysis. The results are informative and represent a new and important contribution. The plots and experiments effectively support the theoretical findings.

### Weaknesses
- The assumption that $\eta_i \ge \underline{\eta}$ uniformly over all $i$ to establish Theorem 4.2 is somewhat strong.
- The analysis focuses on the small-error regime ($\varepsilon_{*,i}$ uniformly small), which limits its applicability to scenarios with potentially large synthetic errors.
- While technical assumptions A3 and A4 are reasonable, providing additional intuition behind them would improve clarity for readers.

### Soundness

The theoretical model, which uses a mixture distribution with a fixed level $\alpha$ at each generation to represent recursive self-training, is reasonable and well motivated. The chosen discrepancy measures are appropriate and meaningful. All theoretical results are rigorously stated and thoroughly justified. The uniform lower bound assumption on $\eta_i$ is somewhat strong and required for the results to hold, but numerical experiments on both toy and real datasets indicate that this assumption likely holds in practice. Overall, the theoretical claims are strongly supported by convincing numerical evidence.

### Presentation

The paper is clearly written and well presented. The results, contributions, and positioning relative to prior literature are clearly stated. It however appears that reference [1] and the results therein are relevant to the current work. Including a discussion of this reference would further improve the presentation and contextualization.

### Significance

A thorough understanding of error propagation and the model collapse phenomenon in generative models is crucial in modern machine learning. This paper provides a clear and important contribution to understanding these phenomena specifically for score-based diffusion models. The insights offered not only advance theoretical knowledge but also have the potential to guide practical applications. Overall, this is a significant and noteworthy contribution.


### Originality

Despite a growing body of empirical evidence for this phenomenon, a rigorous theoretical model for the propagation of errors across generations of recursively trained score-based diffusion models has been missing. This work makes an important step towards closing that gap. Although none of the tools used to establish the bounds are fundamentally novel, the analysis is highly non-trivial and employs several clever techniques that could inform the study of related problems in the future.

---

> ### Author Rebuttal · Authors · 2026-03-28
>
> We thank the reviewer for their time and constructive feedback.
>
> **Q1: Example of a simple example of a basic model where observability uniform boundedness holds**
>
> Thanks for the suggestion. We will add the following simple multidimensional example. For generation $i$, consider the toy linear-Gaussian reverse model $\mathrm{d}\mathbf Y_{i,s}^\star = -\beta \mathbf Y_{i,s}^\star \mathrm{d}s + \mathrm{d}\mathbf B_s$ and a linear state-dependent score error of the form $\mathbf e_i(\mathbf x,s)=a_i(s) C_i \mathbf x$, where $C_i=C_i^\top \succ 0$, and $a_i(s)\ge 0$ is not identically zero. In this case, $M\_T^i = -\int_{t\_0}^T a_i(s)(C\_i\mathbf{Y}\_s^{i,\star})\cdot \mathrm{d}\mathbf{B}\_s$. Using Ito's formula on the quadratic form $\mathbf Y_s^{\star\top} C_i \mathbf Y_s^\star$, one has that $\mathbb E[M_T^i \mid \mathbf Y_{t_0}^{i,\star}=\mathbf y] = \mathbf y^\top K_i \mathbf y + c_i$ for some matrix $K_i \succ 0$. Hence $\operatorname{Var}\big(\mathbb E[M_T^i\mid \mathbf Y_{t_0}^{i,\star}]\big)>0,$
> which implies $\eta_i>0$. The conditions $C_i = C_i^\top \succ 0$ and $a_i(s) \ge 0$ rules out sign-changing cancellations of the state-dependent score error along the reverse dynamics. This provides a simple sufficient example showing that, in a basic linear-Gaussian model with state-dependent score error, the observability coefficient is strictly positive.
>
> **Q2: Discussion of settings in which A3 and A4 hold**
>
> We will add a short discussion giving a simple setting under which (A3) and (A4) are natural.
>
> For (A3), a sufficient condition is when score errors are uniformly bounded, i.e. $\|\mathbf{e}\_i(\mathbf{x},t)\|\_2 \le c$ for all $(\mathbf{x},t) \in \mathbb{R}^d \times [t_0,T]$. In that case, the quadratic variation $\langle M^i \rangle_T$ (defined in equation (14), page 4) is deterministically bounded by $c^2(T-t_0)$. Since $Z\_T^i = M\_T^i - \frac{1}{2}\langle M^i\rangle\_T$, standard exponential-martingale estimates imply that $\mathbb{E}_{\mathbb{P}\_i^\star}[\exp((1+\delta)Z\_T^i)] \le C\_\delta < \infty$, with $C\_\delta$ depending only on $c$, $\delta$ and $T-t\_0$, yielding (A3).
>
> (A4) is a complementary concentration assumption on the quadratic variation $\langle M^i\rangle\_T$ relative to its mean $\varepsilon\_{\star,i}^2$: it is not implied by boundedness alone, but is the additional concentration naturally satisfied in such bounded-error regimes where $\langle M^i\rangle\_T$ has finite moments of all orders and remains of the same order as its mean along typical reverse trajectories. It rules out situations in which the pathwise score-error energy is carried by rare trajectories with unusually large spikes.
>
> We will also clarify in the revision that truncation at $t_0>0$ is important here. By working on $[t_0,T]$, (i)  the analysis avoids the singular regime near $t=0$, where the score and drift mismatch can become poorly behaved and (ii) restricts the analysis to smoothed marginals for which these integrability and moment conditions are more naturally satisfied in practice.
>
> **Q3: Discussion of reference [1]**
>
> Thank you for pointing us to Cui, Pehlevan, and Lu [1], which is indeed relevant. We will discuss its relation to our work  in the revision. Our understanding is that [1] provide a tight characterization of the training dynamics and density generated for the following setting: a two-layer DAE-parametrized diffusion model trained with online SGD to learn specific target densities (high-dimensional Gaussian mixtures). Their analysis captures the architectural biases and mode collapse phenomenon for this class of densities, and they observe that their framework can be iterated to study model collapse across successive generations. Our work is complementary in three ways. First, our bounds are architecture-agnostic: they apply to any score network and any target distribution satisfying our assumptions. Second, we focus on the error accumulation over generations (Theorem 4.2) and characterizes how fast and how far the recursion drifts from $p\_\mathrm{data}$ in a pipeline mixing fresh real data with synthetic data. [1] focuses on what shape the generated density takes. Finally, we operate with fixed dimension whereas [1] work in a high-dimensional asymptotic regime. The two perspectives are complementary and open interesting extensions: [1] gives a precise geometric picture of what collapse looks like for a tractable model, while we provide bounds on the rate and magnitude of collapse for general models. We agree this comparison will help readers better situate our contribution and will update the manuscript accordingly.

---

> > ### Author Rebuttal · Reviewer_AU8H · 2026-04-03
> >
> > Thanks for the response.

---

### Official Review · Reviewer_H5ae · 2026-03-15

**Soundness:** 3
**Presentation:** 3
**Significance:** 2
**Originality:** 3
**Overall Recommendation:** 5
**Confidence:** 4

**Summary:**

This paper studies recursive retraining of diffusion models, where at each generation the new model is trained on a mixture of fresh real data and synthetic samples from the previous model. On the theory side, the authors establish upper and lower bounds linking one-step generation error to pathwise score-estimation error, introducing an observability coefficient $ \eta_i $ for the lower bound, and then derive a discounted accumulation formula across generations showing how the fresh-data fraction $ \alpha $ controls long-run drift.

**Compliance With Llm Reviewing Policy:**

Affirmed.

**Final Justification:**

My concerns are fully addressed.

**Key Questions For Authors:**

1. Line 208: "rather than" is repeated twice
2. Do you think if the finite time horizon $ T $ and initial distribution mismatch error will affect your lower bound result?

**Limitations:**

yes

**Strengths And Weaknesses:**

The paper studies a timely and important problem. Recursive training and model collapse are central concerns in modern generative AI. Technical contributions are strong. The upper bound from pathwise score error to generation error is standard, while the lower bound via the observability coefficient is more interesting and helps clarify when score error actually leaves an error accumulation in the generated distribution.

I have the following concerns:
1. I am a little bit surprised that a lower bound require more assumptions than upper bounds. Could you provide intuition why lower bounds only hold for your proposed restricted problem instances?
2. In practice, we use a neural network to learn score functions. Besides small score matching error, the neural network may have implicit regularization and inductive bias. Do you think it may help mitigate error accumulation in re-training?
3. While the experiments are helpful, they mainly serve to validate the theory rather than establish broad practical guidance. Although acceptable for a theory paper, could you provide numerical experiments on real dataset instead of Gaussian mixture models?

---

> ### Author Rebuttal · Authors · 2026-03-28
>
> We thank the reviewer for their time and constructive feedback.
>
> **Q1: Intuition behind assumptions on lower bound and upper bound**
>
> For each generation $i$, we wish to bound the intra-generation divergence between $\hat{p}^i$ and $q_i$.
> The upper bound is easier because path-space KL divergence upper bounds the terminal-marginal KL by data processing (with our notation, at generation $i$, $\mathrm{KL}(\hat{p}^i \| q_i) \le \mathrm{KL}(\hat{\mathbb{P}}_i \| \mathbb{P}^\star_i)$), and the path-space divergence is directly controlled by the score-error energy through Girsanov’s theorem. By contrast, the lower bound is harder because only the component of the pathwise error that remains visible at the endpoint matters: some errors may cancel along the reverse dynamics and may not affect the terminal marginal. In fact, one can have zero discrepancy between the marginals while the path-space discrepancy is nonzero. The observability coefficient, introduced in Eq. (17), is precisely designed to quantify the fraction of pathwise error that remains visible at the level of marginals.
>
> **Q2: Effect of Implicit regularization and inductive bias**
>
> We agree that implicit regularization and inductive bias might help mitigate error accumulation in re-training. While they do not appear explicitly, their effect is reflected in our framework. Indeed, these architectural and optimization choices influence the size and evolution of the score errors $(\varepsilon\_{\star,i}^2)\_i$ and our theory characterizes how the score errors are propagated or forgotten across generations. By Theorem 3.4 (p. 6),  $\chi^2(\hat p^{i+1}\|q_i)\asymp \varepsilon_{\star,i}^2$, so better regularization and implicit bias directly reduces the one-generation discrepancy. By, Theorem 4.2, the accumulated divergence is governed by $\sum_{i=i_0}^{N}(1-\alpha)^{2(N-i)}\varepsilon_{\star,i}^2$, so the benefit compounds across generations: if $(\varepsilon_{\star, i})_i$ is made sufficiently small or decaying, geometric forgetting dominates accumulation.
>
> Furthermore, our framework reveals that the structure of errors matters independently of their energy. The observability coefficient $\eta_i$ identifies that not all components of score error actually drive divergence.  This suggests that regularization techniques that shape the structure of score errors toward less observable directions could be doubly effective. We leave a systematic investigation of this to future work.
>
> **Q3: Experiments on real datasets beyond Gaussian mixture models**
>
> To partially address this concern, we included real images experiments on Fashion-MNIST and CIFAR-10 (Appendix G.3) to test whether the main mechanisms predicted by our theory remain visible beyond the analytically tractable GMM setting. Specifically, on CIFAR-10 we validate that the observability coefficient remains positive in practice (Fig. 9), which supports the relevance of our lower-bound in a fully data-driven neural-network setting. On Fashion-MNIST we validate the memory structure predicted by Theorem 4.2 (Fig. 10): the contribution of past generations appear to decay geometrically, and the decay rate depends on $\alpha$ as predicted by the theory.
>
> **Key Question 2: Effect of finite time horizon and initialization error**
>
> Yes, a finite horizon $T$ introduces an additional bias term, because the learned reverse process is initialized from a Gaussian law rather than from $q_{i,T}$, the terminal marginal of the forward diffusion. For the OU forward diffusion, this initialization mismatch decays exponentially fast in $T$. By contrast, the score-error energy entering our lower bound, $\varepsilon\_{\star,i}^2 := \mathbb E\_{\mathbb{P}\_i^\star}[ \int\_{t_0}^T \|\mathbf{e}\_i(\mathbf{Y}\_s^{i,\star},s)\|_2^2\,\mathrm{d}s]$, is not expected to grow exponentially with $T$; under a mild uniform-in-time second-moment bound on the score error, it grows at most linearly in $T$. Hence, for sufficiently large $T$, the initialization mismatch is negligible compared with the score-error term that drives the lower-bound mechanism. We omitted it in order to isolate score-error propagation across generations, but we will add a discussion clarifying this point in the revision.

---

> > ### Author Rebuttal · Reviewer_H5ae · 2026-04-04
> >
> > Thank you for your response. I like this rebuttal.

---

### Official Review · Reviewer_T23q · 2026-03-16

**Soundness:** 4
**Presentation:** 4
**Significance:** 3
**Originality:** 3
**Overall Recommendation:** 4
**Confidence:** 3

**Summary:**

The paper theoretically study the model collapse phenomenon. The paper split the total model collapse error into two parts: short-term intra-generation divergence and long-term accumulate divergence. The intra-generation divergence is proven to be bounded by the score proximation error integrated along the sampling trajectory with a constant scale. An the long-term accumulate error is the accumulate divergence of the intra-generation divergence with an expoential decay. The paper also has experiments to support their main theorem.

**Compliance With Llm Reviewing Policy:**

Affirmed.

**Key Questions For Authors:**

1. As shown in Figure 3 (left) and Figure 4, the experiments seem to disalign with the theory when $i = 1$. Is there any discussion for this?

**Limitations:**

Yes

**Strengths And Weaknesses:**

## Strength
1. The paper is well-motivated. Model collapse is an important topic in generative AI. Providing a formal theoretical analysis for it is valuable for the field.
2. The paper is technical solid and novel. The paper provides novel two-sided equivalence between score error energy and distribution divergence, and it also been demonstrated by experiments.


## Weakness

1. The paper lacks key related works discussion, which might weaken their contribution:
   a. The paper states in line 99 - 101 that "this is the first lower bound for diffusion models quantifying the discrepancy between the learned distribution and the target". But there are papers previously doing this, such as [1]. What's more, [1] taken into the network architecture into consideration. While the paper proves a architecture-agnoistic error bound, it would be more convincing to extend experiments in Figure 3 for different model parameters.
   b. There is previous work [2] that theoretically study the model collapse, they also taken network architecture into consideration. The paper needs to detailedly discussed with [2] and differentiate themself.

2. As empirically observed in [3] and theoretically showed in [2], there is a phase transition for $\alpha$: when $\alpha$ larger than a threshold, the synthetic data will boost performance instead of hurt the performance. However, this phase transition is not shown in this paper.

[1] Li, Puheng, Zhong Li, Huishuai Zhang, and Jiang Bian. "On the generalization properties of diffusion models." Advances in Neural Information Processing Systems 36 (2023): 2097-2127.

[2] Fu, Shi, Sen Zhang, Yingjie Wang, Xinmei Tian, and Dacheng Tao. "Towards theoretical understandings of self-consuming generative models." arXiv preprint arXiv:2402.11778 (2024).

[3] Alemohammad, Sina, Josue Casco-Rodriguez, Lorenzo Luzi, Ahmed Imtiaz Humayun, Hossein Babaei, Daniel LeJeune, Ali Siahkoohi, and Richard Baraniuk. "Self-consuming generative models go mad." In The Twelfth International Conference on Learning Representations. 2023.

---

> ### Author Rebuttal · Authors · 2026-03-28
>
> We thank the reviewer for their time and constructive feedback.
>
> **Q1: Related Work**
>
> Thanks for pointing out [1] and [2], which we examined carefully and will discuss in our revised manuscript. To the best of our reading, neither proves a lower bound on the discrepancy between the learned diffusion distribution $\hat p$ and its training target $p_{\mathrm{data}}$. Both [1] and [2] study specific one-hidden-layer random-feature models and obtain upper bounds that depend on the sample size and model capacity. By contrast, we give an architecture-agnostic lower bound and two-sided bounds linking one-generation distribution discrepancy to: i) pathwise score-error energy and ii) an observability coefficient, and propagate this bound across generations. We view these works as complementary to ours in that analysis in [1,2] could be used to obtain precise upper bounds on the score-error energy in terms of training sample size and model capacity for one-hidden-layer random features models.
>
> We agree that [2] is relevant in the setting of model collapse for diffusion models and will discuss it in the revised version. However, our settings differs: they consider a model-specific and finite-sample setting while we focus on a population-level recursive retraining. Both our work and [2] address error accumulation under recursive training, but the technical viewpoint and guarantees are complementary: [2] provides architecture-dependent upper bounds, while our contribution is a two-sided bound characterization in terms of score-error energy that yields an explicit discounted accumulation law across generations. This last result is consistent with and formalizes the tradeoff discussed in [2, Remark 4.7].
>
> **Q2: Phase Transition**
>
> [2] studies recursive training with fixed real data $n$ and increasing synthetic samples $m$, creating two competing effects: larger $m$ improves score estimation, but reduces the effective real data fraction $\frac{n}{n+m}$, exacerbating distribution shift from $p\_\mathrm{data}$. [2] shows that as $\frac{m}{n}$ increases, the accumulated divergence first grows then decays to $0$, formalizing the empirical findings of [3] for one-hidden-layer random features models.
>
> Our framework is complementary: we treat the score-error sequence $(\varepsilon\_{\star, i}^2)\_i$ as given and characterize how it propagates, with $\alpha$ playing the role of $\frac{n}{n+m}$. In our results, $\alpha$ only controls the geometric forgetting rate $(1-\alpha)^{2(N-i)}$ (Theorem 4.2): if $\sum_i \varepsilon_{\star, i}^2 < \infty$, it remains bounded with geometric forgetting. This provides a propagation-level interpretation of the phase transition in [2, 3]. In their setting, increasing $m$ affects both the forgetting rate (through $\alpha = \frac{n}{n+m}$) and the score errors (through sample size). The decreasing branch corresponds to the regime where the larger sample size makes $(\varepsilon\_{\star, i}^2)\_i$ summable, yielding stability in our Theorem 4.2; the increasing branch corresponds to the regime where distribution shift dominates, keeping $\varepsilon_{\star, i}^2$ bounded away from zero and triggering the persistent divergence of Proposition 4.1. Thus, our summability threshold can be viewed as the propagation-level counterpart of their phase transition. Making this correspondence fully precise would require integrating [2]'s estimation analysis with our propagation framework, which we view as an interesting direction for future work.
>
> **Key Question 1: Misalignment for $i=1$**
>
> This is a useful observation, but does not contradict the theory because our bounds are stated under (A1)--(A5), which are only required to hold after some threshold generation $i_0$, allowing an initial transient regime in which the recursive dynamics may not yet satisfy the stability / small-error regimes. Empirically, this transient lasts roughly one step in our setup; after that, the bounds are consistent with theory. In the revised version, we will make this discussion explicit.

---

> > ### Author Rebuttal · Reviewer_T23q · 2026-04-04
> >
> > The author have addressed my concern. I will keep my score.

---

### Decision · Program_Chairs · 2026-04-30

**Decision:**

Accept (regular)

**Comment:**

This paper provides a solid theoretical analysis of the model collapse phenomenon, presenting both upper and lower bounds that are nearly tight. In the rebuttal, the authors addressed concerns regarding related work and experimental validation. Most reviewers agree that this is a strong paper worthy of acceptance, while one reviewer, who is not a specialist in theory, raised concerns about the practical relevance of the developed theory. I believe that practical concerns should not prevent the publication of a strong theoretical paper.